# Inhibition of IL-11 signalling extends mammalian healthspan and lifespan

Anissa A. Widjaja[1,12,13 ✉], Wei-Wen Lim[1,2,12], Sivakumar Viswanathan[1], Sonia Chothani[1], Ben Corden[2,3], Cibi Mary Dasan[1], Joyce Wei Ting Goh[1], Radiance Lim[1], Brijesh K. Singh[1], Jessie Tan[2], Chee Jian Pua[2], Sze Yun Lim[1], Eleonora Adami[4], Sebastian Schafer[1], Benjamin L. George[1], Mark Sweeney[5], Chen Xie[2], Madhulika Tripathi[1], Natalie A. Sims[6,7], Norbert Hübner[4,8,9], Enrico Petretto[1,10], Dominic J. Withers[5,11], Lena Ho[1], Jesus Gil[5,11], David Carling[5,11] & Stuart A. Cook[1,2,5,13 ✉]

For healthspan and lifespan, ERK, AMPK and mTORC1 represent critical pathways and inflammation is a centrally important hallmark[1–7]. Here we examined whether IL-11, a pro-inflammatory cytokine of the IL-6 family, has a negative effect on age-associated disease and lifespan. As mice age, IL-11 is upregulated across cell types and tissues to regulate an ERK–AMPK–mTORC1 axis to modulate cellular, tissue- and organismal-level ageing pathologies. Deletion of *Il11* or *Il11ra1* protects against metabolic decline, multi-morbidity and frailty in old age. Administration of anti-IL-11 to 75-week-old mice for 25 weeks improves metabolism and muscle function, and reduces ageing biomarkers and frailty across sexes. In lifespan studies, genetic deletion of *Il11* extended the lives of mice of both sexes, by 24.9% on average. Treatment with anti-IL-11 from 75 weeks of age until death extends the median lifespan of male mice by 22.5% and of female mice by 25%. Together, these results demonstrate a role for the pro-inflammatory factor IL-11 in mammalian healthspan and lifespan. We suggest that anti-IL-11 therapy, which is currently in early-stage clinical trials for fibrotic lung disease, may provide a translational opportunity to determine the effects of IL-11 inhibition on ageing pathologies in older people.

The major signalling mechanisms that regulate lifespan across species include ERK, STK11 (also known as LKB1), AMPK, mTORC1 and IGF1–insulin modules[1–3]. These pathways are collectively perturbed in old age to activate hallmarks of ageing, which include mitochondrial dysfunction, inflammation and cellular senescence[1]. In aged organisms, the AMPK–mTORC1 axis is uniquely important for metabolic health, with notable effects in adipose tissue[8,9], and therapeutic inhibition of mTOR extends lifespan in mice[10,11].

Ageing studies to date have focused largely on lifespan extension, particularly in yeast, worms and fruit flies, but lifespan extension does not necessarily reflect longer healthspan[12–14]. There is a need for integrated studies to determine the effects of interventions on both healthspan and lifespan. Laboratory mice are particularly suited for such experiments, as ageing pathologies that are important for human wellbeing and function are apparent and lifespan studies are well established in mice[1,15].

The importance of chronic sterile inflammation for ageing pathologies is increasingly recognized and inflammation itself is a central hallmark of ageing[7,16,17]. In simplified terms, ageing is associated with a dysfunctional adaptive immune system that is characterized by immunosenescence and thymic involution along with inappropriate activation of innate immune genes such as IL-6[7,16,18,19]. The pro-inflammatory signalling factors NF-κB and JAK–STAT3 are specifically implicated in ageing and JAK inhibitors can alleviate age-related dysfunction[2,20,21].

We proposed that IL-11, a pro-inflammatory and pro-fibrotic member of the IL-6 family[22], may promote age-associated pathologies and reduce lifespan. This premise was founded on studies showing that IL-11 can activate ERK–mTORC1 and/or JAK–STAT3[22–25] (Fig. 1a), the observation that IL-11 is upregulated in older people[26] and the fact that IL-11 is increasingly recognized to have a role in senescence, a hallmark of ageing[27]. Here, using a range of genetic and pharmacological approaches, we tested the hypothesis that IL-11 signalling has a negative effect on healthspan and lifespan in mice.

## IL-11 is upregulated with age

We determined IL-11 expression in the liver, visceral gonadal white adipose tissue (vWAT) and skeletal muscle (gastrocnemius) in

[1]Cardiovascular and Metabolic Disorders Program, Duke–National University of Singapore Medical School, Singapore, Singapore. [2]National Heart Research Institute Singapore, National Heart Centre Singapore, Singapore, Singapore. [3]Barts Heart Centre, Barts Health NHS Trust, London, UK. [4]Cardiovascular and Metabolic Sciences, Max Delbrück Center for Molecular Medicine in the Helmholtz Association (MDC), Berlin, Germany. [5]MRC Laboratory of Medical Sciences, London, UK. [6]Bone Biology and Disease Unit, St Vincent's Institute of Medical Research, Melbourne, Victoria, Australia. [7]Department of Medicine, St Vincent's Hospital, The University of Melbourne, Melbourne, Victoria, Australia. [8]DZHK (German Centre for Cardiovascular Research), Partner Site Berlin, Berlin, Germany. [9]Charité–Universitätsmedizin, Berlin, Germany. [10]Institute for Big Data and Artificial Intelligence in Medicine, School of Science, China Pharmaceutical University, Nanjing, China. [11]Institute of Clinical Sciences, Faculty of Medicine, Imperial College, London, UK. [12]These authors contributed equally: Anissa A. Widjaja, Wei-Wen Lim. [13]These authors jointly supervised this work: Anissa A. Widjaja, Stuart A. Cook. ✉e-mail: anissa.widjaja@duke-nus.edu.sg; stuart.cook@duke-nus.edu.sg

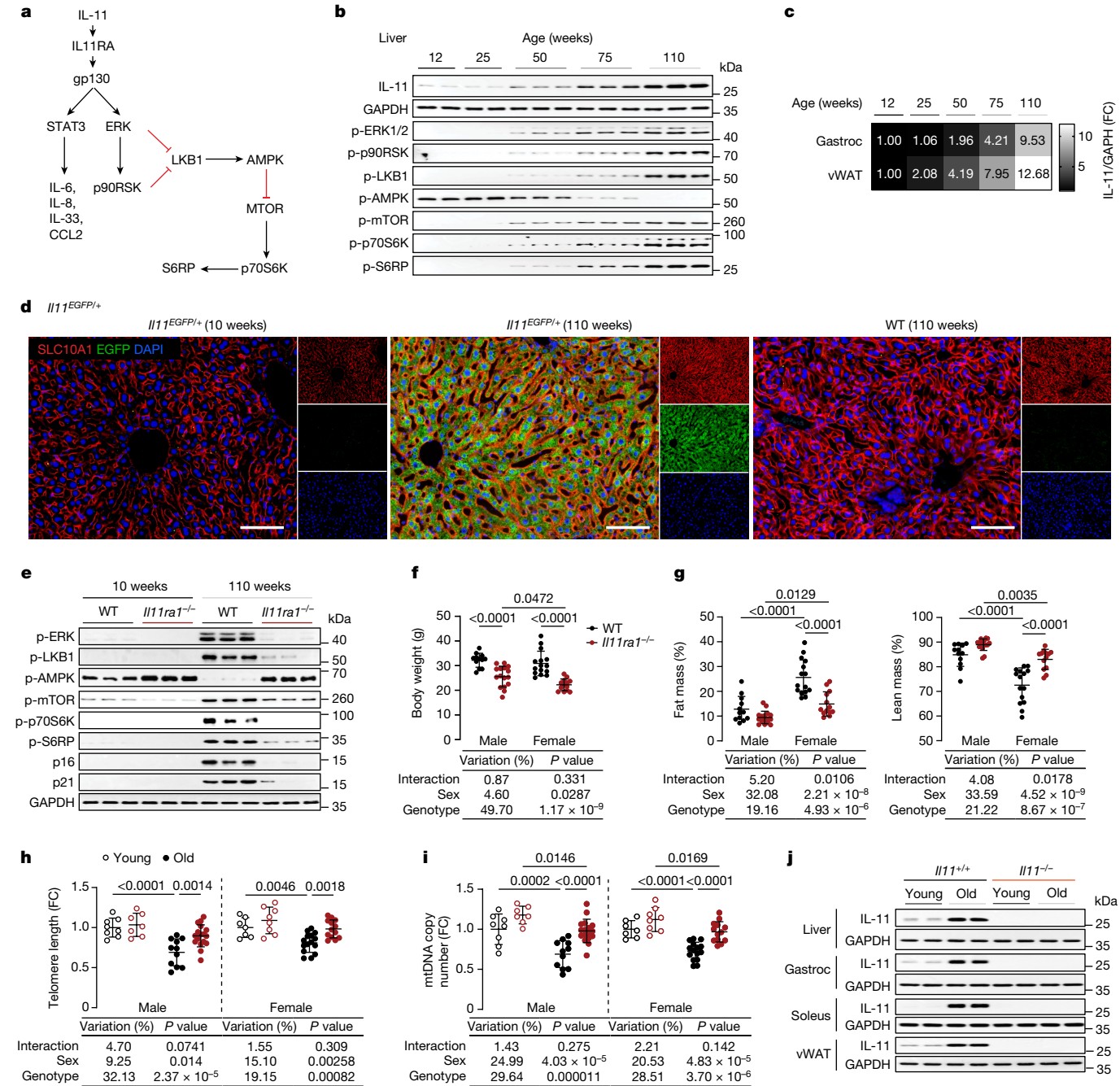

**Fig. 1 | The IL-11–ERK–mTORC1 signalling module is upregulated in ageing and associated with senescence and metabolic decline. a**, Signalling pathway by which IL-11 induces canonical STAT3 activation and non-canonical ERK activation, LKB1–AMPK inactivation and mTOR activation. **b**, Western blots of the indicated liver phosphoproteins from male mice aged 12–110 weeks (n = 5 per group); total (phosphorylated plus unphosphorylated) proteins are shown in Extended Data Fig. 1a. **c**, Heat map showing densitometry of IL-11 protein expression normalized to GAPDH (immunoblots are shown in Extended Data Fig. 1c) in gastrocnemius (gastroc) and vWAT from 12- to 110-week-old male mice (n = 5 per group). **d**, Representative immunofluorescence images (scale bars, 100 μm) of liver EGFP and SLC10A1 expression from 10 and 110-week-old *Il11-EGFP* mice and age-matched wild-type (WT) controls (n = 3 per group). Scale bars, 100 μm. **e**, Western blot of liver extracts from 10- and 110-week-old

male wild-type and *Il11ra1⁻/⁻* mice (n = 5 per group); total proteins are shown in Extended Data Fig. 2a. **f,g**, Body weight (**f**) and percentages of fat and lean mass (male wild-type, n = 12; male *Il11ra1⁻/⁻*, n = 16; female wild-type, n = 15; female *Il11ra1⁻/⁻*, n = 13). **h,i**, Telomere length (**h**) and mtDNA copy number (**i**) in liver from young (10-week-old) and old (110-week-old) male and female wild-type and *Il11ra1⁻/⁻* mice (young male wild-type, n = 8; young male *Il11ra1⁻/⁻*, n = 7; old male wild-type, n = 11; old male *Il11ra1⁻/⁻*, n = 17; young female wild-type, n = 7; young female *Il11ra1⁻/⁻*, n = 8; old female wild-type, n = 15; old female *Il11ra1⁻/⁻*, n = 13). FC, fold change. **j**, IL-11 and GAPDH immunoblots from the indicated organs of 12- and 105-week-old male wild-type and *Il11⁻/⁻* mice (liver and soleus, n = 4 per group; vWAT and gastrocnemius, n = 6 per group). **f–i**, Data are mean ± s.d.; the table below each panel shows summary statistics from two-way ANOVA with Sidak's correction. For gel source data, see Supplementary Fig. 1.

mice over a time course of ageing, which revealed progressive up-regulation of IL-11 in all tissues (Fig. 1b,c and Extended Data Fig. 1a,b). With age, there was progressive activation of ERK–p90RSK, inactivation of LKB1–AMPK and mTOR–p70S6K activation, which comprise

an IL-11 signalling module, in liver and muscle (Fig. 1b and Extended Data Fig. 1c). IL-11 up-regulation was confirmed in livers, vWAT and skeletal muscle of two-year-old male and female mice (Extended Data Fig. 1d).

To identify the cell types that express *Il11* in old mice, we queried the Tabula Muris Senis[28], which was uninformative, likely as *Il11* is expressed at low levels and IL-11 is largely translationally regulated[25]. To better identify IL-11-expressing cells in tissues from old mice, we performed immunohistochemistry in tissues from aged *Il11-EGFP* reporter mice[29]. In two-year-old *Il11-EGFP* mice, IL-11 was apparent in parenchymal cells (hepatocytes in liver, adipocytes in vWAT and myocytes in skeletal muscle) and also seen in stromal, epithelial and endothelial cells across tissues and in nerves in skeletal muscle (Fig. 1d and Extended Data Fig. 1e–g). Thus, IL-11 is expressed in diverse cell types in different tissues with ageing.

## IL-11 is associated with senescence

To explore further the relationship between IL-11 up-regulation and ERK–mTOR activation in tissues of old mice, we studied ten-week-old and two-year-old *Il11ra1*[-/-] mice and wild-type littermate controls[30,31]. On immunoblots of liver, vWAT and gastrocnemius extracts, old wild-type mice exhibited increased phosphorylated (p)-ERK and p-p90RSK, increased p-LKB1 leading to reduced p-AMPK, and increased levels of p-mTOR, p-p70S6K and p-S6RP (Fig. 1e and Extended Data Fig. 2a–c). Levels of the canonical senescence markers p16[Inka4] and p21[Waf1/Cip1] were increased in tissues of old wild-type mice. By contrast, the phosphorylation status of these various kinases and S6RP, and levels of p16 and p21 were similar between old *Il11ra1*[-/-] mice and young wild-type mice (Fig. 1e and Extended Data Fig. 2a–c).

## Deletion of *Il11ra1* improves metabolism

Compared with wild-type littermate controls, two-year-old *Il11ra1*[-/-] mice had lower body weights (Fig. 1f), and female *Il11ra1*[-/-] mice had decreased fat mass and increased lean mass (Fig. 1g). *Il11ra1*[-/-] mice of both sexes had slightly higher core body temperatures than wild-type controls (Extended Data Fig. 2d). Old *Il11ra1*[-/-] mice of both sexes had lower indexed vWAT mass (vWAT weight normalized to body weight) and increased indexed gastrocnemius mass (Extended Data Fig. 2e). Liver indices were similar between genotypes, whereas liver triglyceride levels were lower in *Il11ra1*[-/-] mice (Extended Data Fig. 2e,f).

Serum cholesterol and triglyceride levels were higher in old wild-type mice than in *Il11ra1*[-/-] littermates (Extended Data Fig. 2g,h). Livers of old *Il11ra1*[-/-] mice of both sexes exhibited reduced expression of pro-inflammatory (*Ccl2*, *Ccl5*, *Tnf* and *Il1b*) and fatty acid synthesis (*Acc*, *Fasn* and *Srebp1c*) genes (Extended Data Fig. 2i,j). Serum alanine transaminase (ALT) and aspartate aminotransferase (AST) levels, markers of hepatocyte damage, were increased in old wild-type mice but not in old *Il11ra1*[-/-] mice (Extended Data Fig. 2k,l).

We assessed telomere lengths and mitochondria DNA (mtDNA) copy numbers—biomarkers associated with biological age[1,32]—in liver and gastrocnemius and found significant preservation of these phenotypes in tissues of old *Il11ra1*[-/-] mice (Fig. 1h,i and Extended Data Fig. 2m,n).

## IL-11 induces senescence in human cells

IL-11 is linked with senescence—we therefore explored the direct effects of IL-11 on senescence in human cell types corresponding to those found to express IL-11 in aged mice (Fig. 1d and Extended Data Fig. 1e–g). Stimulation of human fibroblasts or hepatocytes with IL-11 activated ERK–mTOR, increased levels of p16 and p21, and reduced expression of PCNA and cyclin D1, which was prevented by U0126 or rapamycin (Extended Data Fig. 3a,b). Supernatants of IL-11-stimulated fibroblasts had increased amounts of senescence-associated secretory phenotype factors, and this increase was inhibited by U0126 or rapamycin (Extended Data Fig. 3c,d). Deeper profiling of mTORC1-dependent senescence-associated secretory phenotype factors (IL-6, IL-8, LIF, VEGFA, HGF, CCL2, CXCL1, CXCL5, CXCL6 and CCL20[33]) in supernatants

of hepatocytes revealed significant IL-11-stimulated ERK- and mTORC1-dependent regulation of the majority of these proteins (Extended Data Fig. 3e–g).

Accumulation of senescent cells contributes to ageing pathologies. We modelled replicative senescence by serially passaging human fibroblasts in the presence of a neutralizing IL11RA antibody (X209) or an IgG control[29,34]. We observed passage-dependent phosphorylation of ERK–mTORC1, NF-κB and STAT3 along with increased amounts of senescence markers—these effects were IL-11-dependent with evidence of both autocrine and paracrine effects (Extended Data Fig. 4a–d). Telomere lengths and mtDNA copy numbers were similar between early passage (passage 4 (P4)) cells and X209-treated late passage (P14) cells, whereas these phenotypes were reduced in IgG-treated P14 cells (Extended Data Fig. 4e,f). Basal metabolic respiration was impaired in IgG-treated P14 fibroblasts, whereas X209-treated P14 cells were similar to P4 fibroblasts (Extended Data Fig. 4g,i).

## Deletion of *Il11* in female mice

To support the data generated in *Il11ra1*-deleted mice on a mixed C57BL6/129 genetic background[30] and to more deeply dissect age-related effects, we studied young (3-month-old) and aged (2-year-old) female mice with deletion of *Il11 (Il11*[-/-]) on a C57BL6/J background[31].

Immunoblots confirmed IL-11 up-regulation across tissues in old age in this additional strain (Fig. 1m). Old female *Il11*[-/-] mice had lower body weights and fat mass and preserved lean mass (Fig. 2a–c). The frailty score[15] of old female *Il11*[-/-] mice was lower than that of old wild-type mice and their body temperatures were mildly increased (Fig. 2d and Extended Data Fig. 5a). Lower frailty scores were largely driven by improvements in tremor, loss of fur colour, gait disorders and vestibular disturbance (Supplementary Table 1). Muscle strength was higher in both young and old *Il11*[-/-] mice (a phenomenon that was observed for some other phenotypes) compared with age-matched controls (Fig. 2e and Extended Data Fig. 5b).

Chronic inhibition of mTORC1 with rapamycin can cause glucose intolerance owing to indirect inhibition of mTORC2[35]. It was therefore important to more fully assess the effects of IL-11 inhibition on liver function, metabolism and glucose utilization in old mice. As wild-type mice aged, there were increases in serum AST, ALT, cholesterol and triglycerides, which were collectively mitigated in old *Il11*[-/-] mice (Fig. 2f and Extended Data Fig. 5c,d). Glucose tolerance test (GTT) and insulin tolerance test (ITT) profiles of old *Il11*[-/-] mice were similar to those of young wild-type mice, whereas GTTs and ITTs of old wild-type mice showed impairment (Fig. 2g and Extended Data Fig. 5e,f). Indexed skeletal muscle mass was greater in both young and old *Il11*[-/-] mice compared with the equivalent wild-type mice (Extended Data Fig. 5g).

Compared with old wild-type controls, old *Il11*[-/-] mice had reduced liver mass indices and liver triglyceride content (Extended Data Fig. 5h,i). Indexed vWAT and inguinal subcutaneous white adipose tissue (scWAT) masses were reduced in old *Il11*[-/-] mice, whereas brown adipose tissue (BAT) mass was unchanged (Fig. 2h and Extended Data Fig. 5j). To examine the potential role of de novo lipogenesis in aged tissue, we profiled the expression of fatty acid synthesis genes in vWAT and found their expression to be increased with age in old wild-type mice but not in old *Il11*[-/-] mice (Fig. 2i). Similar to *Il11ra1*[-/-] mice (Extended Data Fig. 2b,c), activation of the ERK–mTORC1 axis and up-regulation of senescence markers were mitigated in vWAT and gastrocnemius of old *Il11*[-/-] mice (Fig. 2j and Extended Data Fig. 5k,l). Pro-inflammatory gene expression was increased in vWAT of old wild-type mice but not in that of old *Il11*[-/-] mice (Extended Data Fig. 5m).

Telomere lengths and mtDNA content in liver, skeletal muscle and vWAT were reduced in old wild-type mice and these effects were attenuated by *Il11* deletion (Fig. 2k,l). Serum IL-6 levels were increased in old wild-type mice but not in old *Il11*[-/-] mice (Extended Data Fig. 2n).

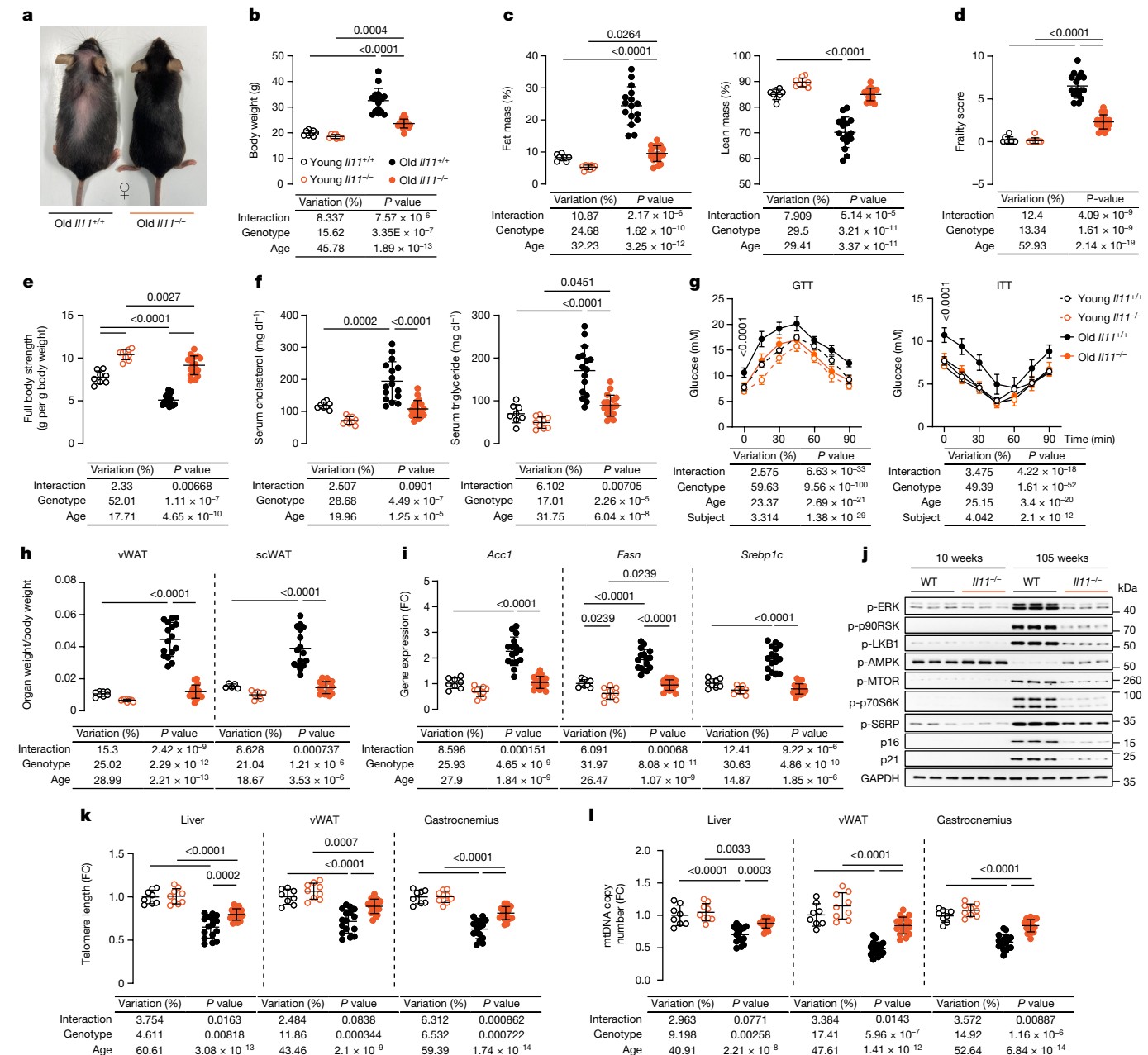

**Fig. 2 | Female *Il11*-deleted mice are protected from age-associated obesity, frailty, and metabolic decline. a**, Representative photograph of 105-week-old female wild-type and *Il11*⁻/⁻ mice. **b–g**, Body weight (**b**), percentage of fat and lean mass (normalized to body weight), frailty score (**d**), full body grip strength (**e**), serum cholesterol and triglycerides (**f**), and GTT and ITT (**g**) of young (12-week-old) and old (105-week-old) female wild-type and *Il11*⁻/⁻ mice. **h–j**, Indexed vWAT and scWAT weight (**h**), relative vWAT mRNA expression level of *Acc1*, *Fasn* and *Srebp1c* (**i**), and western blot showing activation status of ERK1/2, p90RSK, LKB1, AMPK, mTOR, p70S6K and S6RP and protein expression

levels of p16, p21 and GAPDH (**j**). *n* = 6 per group. Western blots for the respective total proteins are shown in Extended Data Fig. 5k. **k,l**, Telomere length and mtDNA copy number (**l**) in vWAT from young and old female wild-type and *Il11*⁻/⁻ mice. **b–i,k–l**, Data are mean ± s.d. (young wild-type, *n* = 8; young *Il11*⁻/⁻, *n* = 9; old wild-type, *n* = 16; old *Il11*⁻/⁻, *n* = 18; except for **h** (scWAT): young wild-type, *n* = 5; young *Il11*⁻/⁻, *n* = 7; old wild-type and *Il11*⁻/⁻, *n* = 16). Two-way ANOVA with Sidak's correction (**b–f,h,i,k,l**); two-way repeated measures ANOVA with Sidak's correction (**h**). For gel source data, see Supplementary Fig. 1.

## Deletion of *Il11* in male mice

Detrimental changes in body habitus, body weight, fat mass, lean mass and frailty scores (driven by tremor, coat condition and fur loss) were mitigated and body temperatures were mildly increased in old male *Il11*⁻/⁻ mice (Extended Data Fig. 6a–e and Supplementary Table 2). Muscle strength was higher in both young and old *Il11*⁻/⁻ mice compared with age-matched wild-type controls, as seen for female mice (Extended Data Fig. 6f). GTT and ITT profiles of old *Il11*⁻/⁻ mice were similar to those of

young mice, whereas those of old wild-type controls were impaired (Extended Data Fig. 6g).

Metabolic cage analysis revealed that the respiratory exchange ratio (RER) was overall higher in old *Il11*⁻/⁻ mice compared with old wild-type mice (Extended Data Fig. 6h). After a period of starvation, refeeding resulted in a greater increase of RER in old *Il11*⁻/⁻ mice, consistent with better metabolic flexibility[36] (Extended Data Fig. 6h). Whereas old *Il11*⁻/⁻ mice were leaner and weighed less than wild-type controls, they consumed more food and had similar levels of locomotor activity

(Extended Data Fig. 6h). Calorie-losing enteropathy of $Il11^{-/-}$ mice was excluded by bomb calorimetry (Extended Data Fig. 6i).

Sarcopenia was apparent in the muscle of old wild-type mice, and this effect was less pronounced in old $Il11^{-/-}$ mice, which exhibited greater indexed muscle mass even when young, as seen for female mice (Extended Data Fig. 6j,k). Indexed liver weights were similar between mice of both genotypes (Extended Data Fig. 6l). As for female mice, perhaps the most notable beneficial difference associated with loss of $Il11$ function in old male mice was seen for vWAT mass, whereas BAT mass was similar across genotypes and ages (Extended Data Fig. 6m). An incidental finding of enlarged seminal vesicles, an idiosyncratic age-specific phenomenon in old male mice[37], was more common in old wild-type mice compared with old $Il11^{-/-}$ mice (wild-type: 11 out of 15, $Il11^{-/-}$: 1 out of 14; $P = 0.0003$) (Supplementary Table 2).

## Anti-IL-11 therapy in old male mice

Studies of germline genetic loss of function on healthspan and/or lifespan should be complemented by interventions in late life only to identify phenotypes that are carried through from younger animals and for translational relevance. This is pertinent to the current study, in which some beneficial effects of $Il11$ loss of function (such as increased muscle mass and strength) are apparent even in young $Il11$-deleted mice (Fig. 2 and Extended Data Fig. 6). To achieve this goal, we administered a neutralizing IL-11 antibody (X203) or IgG control to aged mice and studied healthspan indices[29,38] (Fig. 3a).

Compared with controls, mice receiving X203 from 75 to 100 weeks of age progressively lost body weight that was defined by a reduction in indexed fat mass (Fig. 3b,c). Impaired glucose metabolism was apparent across experimental groups at study start that was improved in mice receiving X203, whereas IgG had no effect (Fig. 3d). Frailty scores were mildly increased across experimental groups at study initiation (Fig. 3e). Over the study period, mice receiving no treatment or IgG exhibited frailty progression (for example, tremor and gait disorder), whereas those on X203 did not (Fig. 3e and Supplementary Table 3). Muscle strengths of 100-week-old mice receiving anti-IL-11 were higher than aged-matched controls receiving IgG or untreated, and also higher than those of 75-week-old mice at the start of the experiment (Fig. 3f and Extended Data Fig. 7a).

After six weeks of antibody administration, mice were studied in metabolic cages. The RER of mice receiving X203 was higher than that of IgG-treated mice but lower than that of a cohort of young mice (Fig. 3g and Extended Data Fig. 7b), suggesting that X203 slows age-associated metabolic inflexibility. Administration of X203 was associated with higher core temperatures and increased food intake, whereas locomotor activity levels and faecal caloric densities were similar between study groups (Fig. 3h and Extended Data Fig. 7b,c).

Mice left untreated or given IgG had increased serum cholesterol, triglycerides and IL-6 by study end, which were collectively lowered by X203 therapy to below the levels at 75 weeks of age (Extended Data Fig. 7d,e). Over the course of the experiment, markers of liver damage, hepatic triglyceride content and indexed liver mass increased in untreated and IgG control mice, whereas these phenotypes were either improved or reduced in mice receiving X203 (Fig. 3i–k and Extended Data Fig. 7f).

There was a reduction in indexed vWAT and liver mass and an increase in indexed muscle mass in 100-week-old mice receiving X203, compared with both age-matched controls and 75-week-old mice (Fig. 3k and Extended Data Fig. 7g). Mice receiving X203 had diminished scWAT and an increase in BAT (Extended Data Fig. 7h). The age-specific phenotype of enlarged seminal vesicles in male mice[37] was again diminished by IL-11 loss of function (IgG: 8 out of 13, X203: 2 out of 12; $P = 0.022$) (Supplementary Table 3).

Fibrosis is a canonical feature of ageing and a hallmark of senescence, and IL-11 is known to be pro-fibrotic in human cells and in young adult mice[22,39]. We quantified fibrosis in aged vWAT, skeletal muscle and livers of old mice across experimental groups, which showed reversal of tissue fibrosis across organs of mice receiving X203 (Fig. 3l).

Compared with 75-week-old mice, vWAT from mice receiving IgG for 25 weeks had increased activation of the IL-11–mTORC1 axis and higher expression of senescence markers (Fig. 3m and Extended Data Fig. 7i). By contrast, mice receiving X203 had reduced ERK–mTOR activity and decreased expression of p21 and p16 (Fig. 3m and Extended Data Fig. 7i). One-hundred-week-old untreated and IgG-treated mice had telomere attrition and a reduction in mtDNA copy number, which were not seen in X203-treated mice (Extended Data Fig. 7j,k).

## Anti-IL-11 therapy in old female mice

We also examined the effects of anti-IL-11 therapy on ageing pathologies in old female mice (Extended Data Fig. 8a). Old female mice receiving X203 lost body weight, whereas those administered with IgG gained weight (Extended Data Fig. 8b). At the end of the study period, mice on X203 had lower fat mass, higher lean mass and better GTTs and ITTs than at the outset, whereas the opposite effect was observed in mice on IgG (Extended Data Fig. 8c,d). Frailty scores were similar between study groups at the start of the experiment and these scores progressed in mice receiving IgG but not in mice receiving X203 (Extended Data Fig. 8e and Supplementary Table 4). Muscle strengths were greater than starting levels in female mice receiving X203 and core body temperatures were mildly increased (Extended Data Fig. 8f–h).

## Anti-IL-11 restores white adipose beiging

To further dissect molecular mechanisms, we performed bulk RNA sequencing (RNA-seq) of vWAT, gastrocnemius and liver from IgG-treated or anti-IL-11-treated 100-week-old mice (Supplementary Table 5). Across tissues, mice receiving anti-IL-11 had the most significant gene set enrichment scores for hallmarks of oxidative phosphorylation and metabolism, whereas scores for markers of inflammation, EMT and JAK–STAT3 signalling were reduced (Fig. 4a).

In aged vWAT, genes associated with senescence (*Cdkn2a*, *Tnf*, *Il10*, *Il1b*, *Bst1*, *Irg1*, *Parp14*, *Itgax* and *Itgam*) as identified by The Tabula Muris Senis[28] were upregulated, an effect that was mitigated by anti-IL-11 therapy (Fig. 4b and Extended Data Fig. 9a). A similar, although less pronounced inhibition of senescence markers was seen in muscles and livers of mice receiving anti-IL-11.

More detailed study of the vWAT transcriptome revealed that the gene that was most upregulated by anti-IL-11 genome-wide was *Ucp1*, which is important for the development of thermogenic 'beige' adipocytes in white adipose tissue (WAT) deposits[40,41] (Fig. 4c and Supplementary Table 5). On closer inspection, we found up-regulation of a larger beiging programme (*Acot2*, *Cidea*, *Cox4i1*, *Cox8b*, *Dio2*, *Elovl3*, *Eva1a*, *Fabp3*, *Ppargc1a*, *Ppargc1b*, *Ppara* and *Prdm16*) in vWAT of mice receiving anti-IL-11 (Fig. 4d). Age-dependent up-regulation of UCP1 and PGC1α in male mice receiving anti-IL-11 was validated at the protein level (Fig. 4e)

To support our findings, we showed age-related suppression of UCP1 expression in vWAT of female control mice, which was mitigated in female mice lacking *Il11* and in mice of both sexes lacking *Il11ra1* (Fig. 4f and Extended Data Fig. 9b). A targeted assessment of mitochondrial gene expression in vWAT revealed significant increases in terms associated with mitochondrial biogenesis and function in mice receiving anti-IL-11 (Extended Data Fig. 9c,d).

In mice receiving X203, there was strong up-regulation of *Clstn3b*, a newly identified mammal-specific product of the 3′ end of the *Clstn3* locus that promotes WAT triglyceride metabolism in partnership with *S100b*[42,43], which was also upregulated (Fig. 4g and Extended Data Fig. 9e). There was limited down-regulation of *Ucp1* in BAT with age in wild-type mice, and *Ucp1* expression was mildly increased in BAT of $Il11^{-/-}$ mice but not in mice receiving anti-IL-11 (Extended Data Fig. 9f,g).

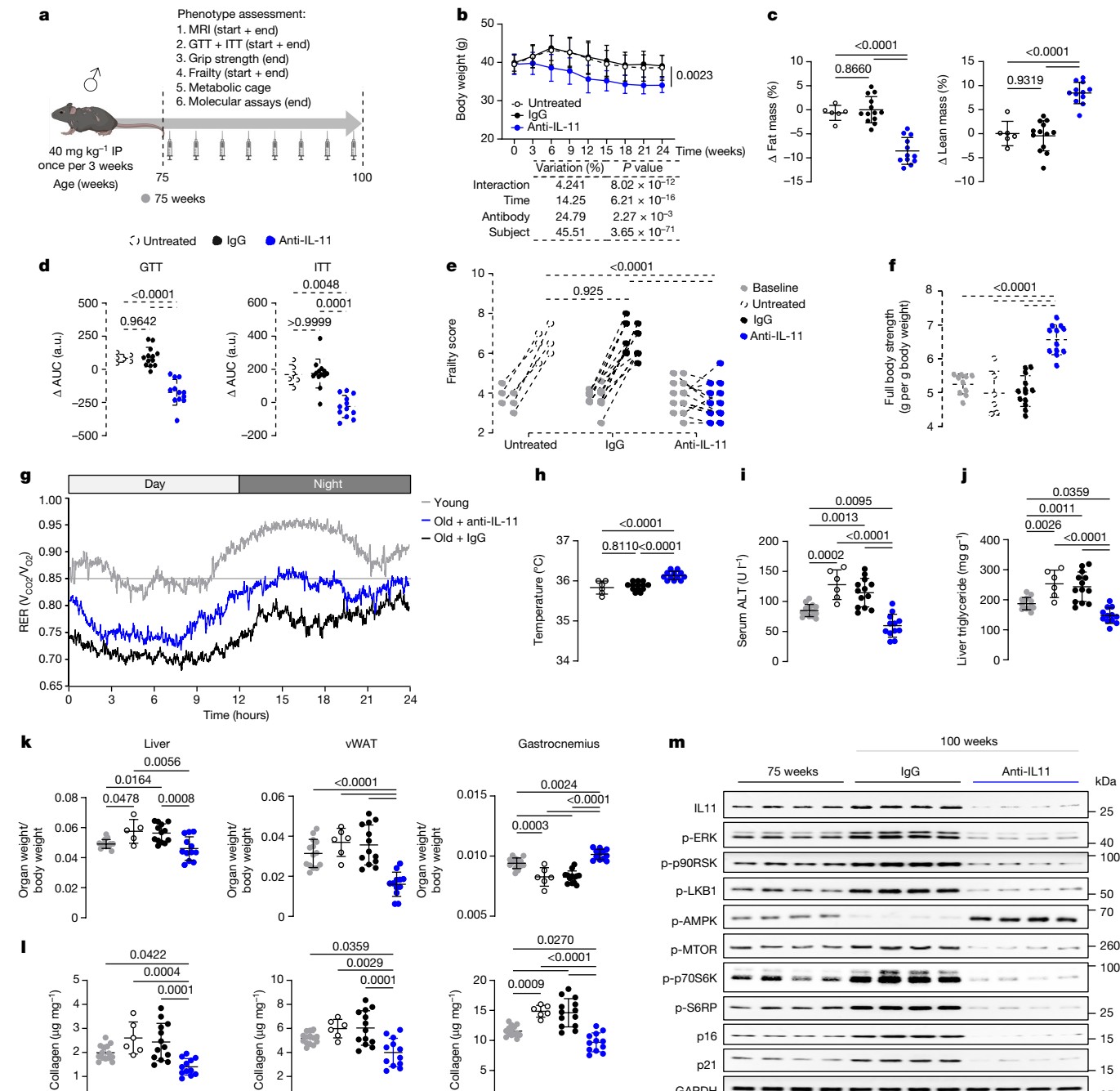

**Fig. 3 | Therapeutic inhibition of IL-11 reduces age-associated metabolic dysfunction, pathogenic signalling and sarcopenia in male mice.** **a**, Schematic of anti-IL-11 (X203) therapeutic dosing experiment in old male mice for experiments shown in **b**–**m**. Mice were either aged naturally (untreated) or given either X203 or an IgG control antibody (40 mg kg⁻¹, every 3 weeks) starting from 75 weeks of age for a duration of 25 weeks. Created with BioRender.com. **b**, Body weights across time. **c**,**d**, Changes (Δ) in fat and lean mass percentage (**c**) and area under the curve (AUC) of GTT and ITT (**d**) (values at endpoint (100-week-old) − values at starting point (75-week-old)). a.u., arbitrary units. **e**, Frailty scores at start and endpoint. Data are shown as values recorded at starting and endpoint. **f**, Full body grip strength. **g**, RER in young (14-week-old) and IgG or X203-treated old (81-week-old) mice, 6 weeks after IgG or X203 administration was started (n = 10 per group). **h**–**j**, Body temperatures (**h**), serum ALT (**i**) and liver triglycerides (**j**). **k**,**l**, Indexed weights of (**k**) and total collagen content (by hydroxyproline assay) in (**l**) liver, gastrocnemius and vWAT. **m**, Western blot showing activation status of ERK1/2, p90RSK, LKB1, AMPK, mTOR, p70S6K, S6RP and protein expression levels of IL-11, p16, p21 and GAPDH in vWAT (n = 6 per group). Western blots of total protein are presented in Extended Data Fig. 7i. **b**–**d**,**f**,**h**–**l**, Data are mean ± s.d. 75-week-old control: n = 10 (**f**), n = 14 (**i**–**l**); untreated 100-week-old: n = 6 (except for **k** (liver), n = 5); IgG-treated 100-week-old: n = 13; X203-treated 100-week-old: n = 12. Two-way repeated measures ANOVA with Sidak's correction (**b**); one-way ANOVA with Tukey's correction (**c**,**d** (GTT), **e**,**f**,**h**–**l**); one-way ANOVA with Kruskal–Wallis correction (**d** (ITT)). For gel source data, see Supplementary Fig. 1.

Pro-inflammatory gene expression was higher in vWAT of mice receiving IgG compared with those receiving X203 (Fig. 4b), mirroring findings seen in livers of *Il11ra1⁻/⁻* mice and vWAT of *Il11⁻/⁻* mice (Extended Data Figs. 2i and 5m). Further analysis of young and old *Il11ra1⁻/⁻* and wild-type mice confirmed age-dependent pro-inflammatory gene expression in the vWAT of wild-type mice that was decreased in mice of *Il11ra1⁻/⁻* genotype across sexes (Extended Data Fig. 9h).

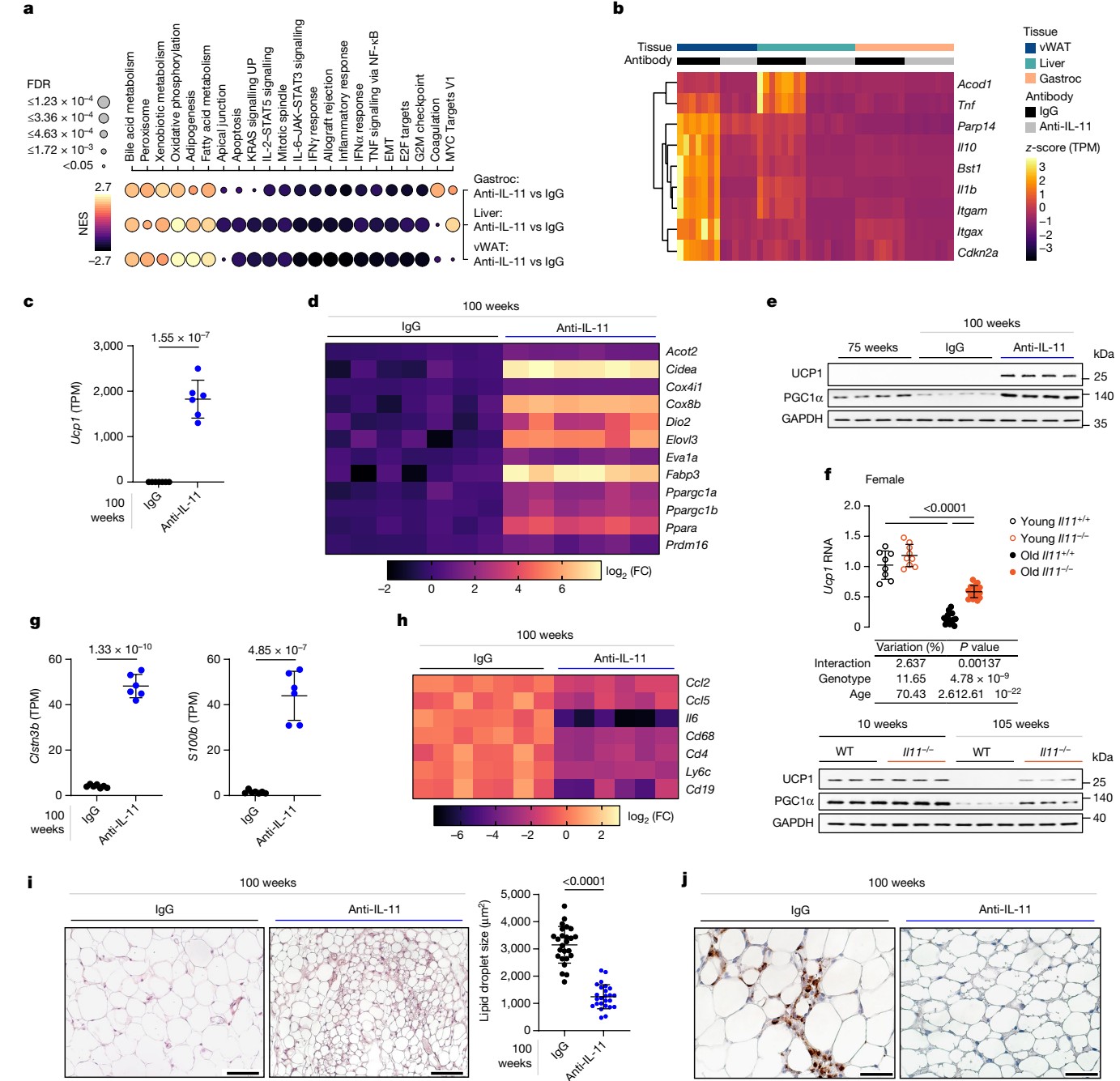

**Fig. 4 | Anti-IL-11 reduces vWAT inflammation and reactivates an age-repressed thermogenic programme. a–e,g–j,** Data for therapeutic experiments in old male mice as shown in Fig. 3a. **a,** Bubble map showing hallmark gene set enrichment analysis for differentially expressed genes in the vWAT, liver and gastrocnemius of mice receiving anti-IL-11 therapy compared with IgG. Colour represents normalized enrichment score (NES); black represents negative NES, indicating down-regulation of the gene set; yellow represents positive NES, suggesting up-regulation. Dot size indicates significance (the larger the dot, the smaller the adjusted *P* value). EMT, epithelial– mesenchymal transition. **b,** Heat map of row-wise scaled transcripts per million (TPM) values of senescence genes in vWAT, liver, gastrocnemius. **c,** Abundance of *Ucp1* reads in vWAT. **d,** log₂-transformed fold change heat map of beiging genes in vWAT from IgG- or anti-IL-11-treated 100-week-old mice, based on RNA-seq. **e,** Western blot of UCP1, PGC1α and GAPDH expression in vWAT (*n* = 6

per group). **f,** Relative expression levels of *Ucp1* mRNA (young wild-type, *n* = 8; young *Il11*⁻/⁻, *n* = 9; old wild-type, *n* = 16; old *Il11*⁻/⁻, *n* = 18) as well as UCP1 and PGC1α protein expression (*n* = 6 per group) in vWAT isolated from young and old female wild-type and *Il11*⁻/⁻ mice. **g,h,** Abundance of *Clstn3b* and *S100b* reads (**g**) and log₂-transformed fold change heat map of pro-inflammatory markers (from RNA-seq) (**h**) in vWAT. **i,** Haematoxylin and eosin-stained vWAT (scale bars, 100 μm) and quantification of lipid droplet size (mean of lipid droplet area, *n* = 25 (5 fields per mouse from 5 mice per group)). **j,** Immunohistochemistry staining of CD68 in vWAT (scale bars, 50 μm). **a–d,f–h,** Liver and gastrocnemius (*n* = 8 per group), vWAT IgG, *n* = 7; vWAT anti-IL-11, *n* = 6. **c,f,g,i,** Data are mean ± s.d. Two-tailed Student's *t*-test (**c,g,i**); two-way ANOVA with Sidak's correction (**f**). For gel source data, see Supplementary Fig. 1. Scale bars: 100 μm (**i**), 50 μm (**j**).

Stromal inflammation is associated with immune cell infiltration, and we found that the immune cell surface marker genes *Cd68*, *Cd4*, *Ly6C* and *Cd19* were downregulated in the vWAT of mice receiving X203

(Fig. 4h). Histology studies revealed that vWAT of X203-treated mice exhibited an average 2.5-fold reduction in lipid droplet area, increased beige adipocyte foci and fewer resident CD68⁺ macrophages (Fig. 4i,j).

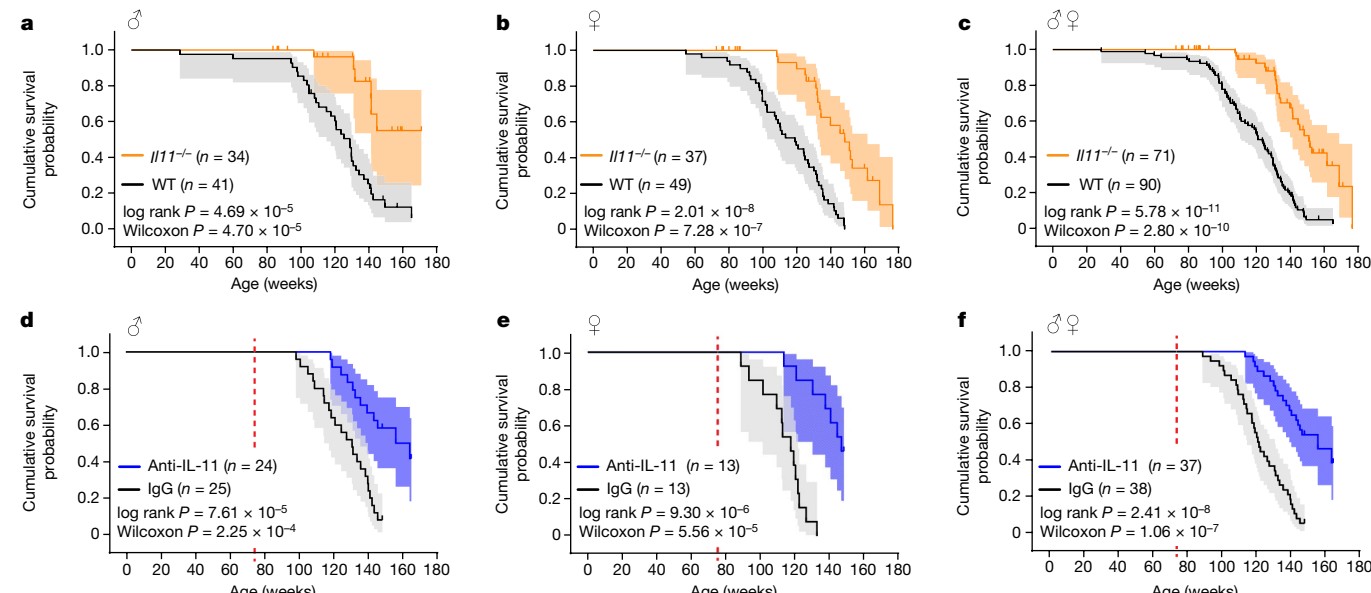

**Fig. 5 | Genetic or pharmacologic inhibition of IL-11 extends life expectancy of male and female mice. a–c**, Kaplan–Meier survival curves (shading represents 95% confidence interval) showing the cumulative survival probabilities for male (**a**), female (**b**) and sex-pooled (**c**) wild-type and *Il11*[−/−] mice. **d–f**, Kaplan–Meier survival curves showing the cumulative survival probabilities for male (**d**), female (**e**) and sex-pooled (**f**) mice, comparing those receiving monthly administration of IgG or X203 (40 mg kg[−1], intraperitoneal injection), starting from 75 weeks of age (red dotted line). Statistical significance (two-tailed *P* value) was assessed by means of the log-rank (Mantel–Cox) and Wilcoxon test for survival curve comparisons.

## Inhibition of IL-11 extends lifespan

In parallel to the healthspan experiments, we carried out lifespan studies in male and female *Il11*[−/−] mice and wild-type littermate controls, which we observed until they were found dead or euthanized when moribund (Fig. 5a–c). Pooled analysis showed that *Il11*[−/−] mice had significantly longer lifespans than wild-type controls (median lifespan: wild-type, 120.9 weeks; *Il11*[−/−], 151 weeks). Sex-specific analyses revealed significant lifespan extension in female *Il11*[−/−] (median lifespan: wild-type, 118.9 weeks; *Il11*[−/−], 148.3 weeks) as well as male *Il11*[−/−] (median lifespan: wild-type, 128.7 weeks; *Il11*[−/−], to be determined) mice.

To progress our studies to a more translationally relevant approach, we examined the effects on lifespan following IL-11 inhibition in late life using 75-week-old male and female mice assigned to receive monthly injections of either anti-IL-11 or IgG until death (Fig. 5d–f and Extended Data Fig. 10). Pooled analysis showed that mice receiving anti-IL-11 have significantly longer lifespans (median lifespan: IgG, 120.9 weeks; X203, 155.6 weeks). Sex-specific analyses showed a significant extension of lifespan in females (median lifespan: IgG, 117.1 weeks; X203, 146.4 weeks), which was also apparent for males (median lifespan: wild-type, 130.3 weeks; X203, 159.6 weeks).

Cancers are a common cause of death in old mice[44], and gross autopsy data revealed fewer macroscopic tumours in mice with *Il11* deletion (pooled sexes: wild-type, 49 out of 84 mice had tumours; *Il11*[−/−], 3 out of 25 mice had tumours; *P* < 0.0001) or on anti-IL-11 therapy (pooled sexes: IgG, 22 out of 36 mice had tumours; X203, 3 out of 19 mice had tumours; *P* = 0.0013) (Supplementary Tables 6 and 7).

## Discussion

IL-11 is progressively upregulated across tissues with age, probably as an alarmin-type response to age-related pathogenic factors that include cytokines, proteotoxic stress, oxidative species and DNA damage, among others[22]. We propose that the pleiotropic benefits seen with inhibition of IL-11 reflect its modulation of multiple ageing pathways (such as ERK, AMPK, mTOR and JAK–STAT3), as seen using polypharmacy in flies[2,6]. IL-11 has not been extensively studied and was not previously

thought to be important for ageing, however SNPs at the *IL11* locus are associated with osteoarthritis[45] and menopause[46], and IL-11 is linked with senescence and diseases that are common in older people[22,27].

The metabolic effects seen with inhibition of IL-11 in old mice phenocopy those of young mice with WAT-specific deletion of Raptor[41]. Although we did not study mice at thermoneutrality, we surmise that inhibition of IL-11 prevents mTORC1 activation in fat, affording age-repressed WAT beiging that can be particularly prominent in mice[40,41]. We highlight that although we excluded food intake and enteric or locomotor-related energy expenditure and showed WAT beiging across genetic and therapeutic models, we did not pinpoint the specific physiology leading to weight loss with IL-11 inhibition.

Beyond metabolism, inhibition of IL-11 improved deterministic features of ageing that are common among vertebrates (such as frailty and sarcopenia), showing generic anti-ageing benefits at the organismal level. Intriguingly, some of the beneficial effects of germline *Il11ra1* or *Il11* deletion, notably in muscle and fat, were apparent even in young mice, perhaps suggesting primacy of metabolic benefits. We did not discern cell-type specificity but infer that tissue-localized IL-11 activity is important, given its known autocrine and paracrine activities[22].

Inhibition of IL-11 increased lifespan in both male and female mice. The magnitude of lifespan extension remains to be fully determined but current data suggest that anti-IL-11 therapy given in late life increases median lifespan by more than 20% in both sexes. In these experiments, anti-IL-11 was injected in mice from 75 weeks of age (human equivalent to approximately 55 years of age) and it remains to be seen whether administration to older mice has similar effects and/or if short term anti-IL-11 therapy is effective for lifespan extension, as seen for rapamycin. Mouse mortality in old age is often cancer-related[44] and our end-of-life autopsy data support the notion that inhibition of IL-11 significantly reduces age-related cancers. Of note, IL-11 is important for tumorigenesis and tumour immune evasion and clinical trials of anti-IL-11 in combination with immunotherapy to treat cancer are planned[22].

Chronic sterile inflammation is an important hallmark of ageing that is intimately linked with senescence and implicated in the pathogenesis of age-related frailty, metabolic dysfunction and multi-morbidity[7,16,17,39]. Studies of invertebrates have shown that innate immune signalling,

notably Jak–Stat signalling in fly adipose tissue, can adversely affect metabolism and lifespan[47,48]. We show here that a pro-inflammatory cytokine can affect affect age-related decline and lifespan in a mammal. The relative contributions of canonical (JAK–STAT3) and non-canonical (MEK–ERK) IL-11 signalling, alone or in combination, for ageing phenotypes remain to be determined.

Inhibition of ERK or mTOR or activation of AMPK by trametinib, rapamycin or metformin, respectively, increase lifespan in model organisms and such drugs are advocated by some for use in humans. However, these agents have on- and off-target toxicities along with variable, and sometimes detrimental, effects on healthspan and inflammation[12,13,35,49]. Our data suggest that anti-IL-11 therapy, which has a reassuring safety profile and is currently in early-stage clinical trials for fibroinflammatory diseases, is a potentially translatable approach for extending human healthspan and lifespan[22].

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

## Methods

### Antibodies

**Commercial antibodies.** Adiponectin (AdipoQ, 21613-1-AP, Proteintech), p-AMPK Thr172 (2535, clone 40H9, CST), AMPK (5832, clone D63G4, CST), CD31 (ab222783, clone EPR17260-263, abcam), CD68 (ab125212, abcam), cyclin D1 (55506, clone E3P5S, CST), p-ERK1/2 Thr202/Tyr204 (4370, clone D13.14.4E, CST), ERK1/2 (4695, clone 137F5, CST), GAPDH (2118, clone 14C10, CST), FHL1 (10991-1-AP, Proteintech), GFP (ab290 and ab6673, abcam), p-LKB1 Ser428 (3482, clone C67A3, CST), LKB1 (3047, clone D60C5, CST), p-mTOR Ser2448 (2971, CST), mTOR (2972,CST), p-NF-κB p65 Ser536 (3033, clone 93H1, CST), NF-κB p65 (8242, clone D14E12, CST), p16 (human, ab108349, clone EPR1473, abcam), p16 (mouse, ab232402, clone EPR20418, abcam), p21 (human, ab109520, clone EPR362, abcam), p21 (mouse, ab188224, clone EPR18021, abcam), p-p70S6K Thr389 (9234, clone 108D2, CST), p70S6K (2708, clone 49D7, CST), p-p90RSK Ser380 (11989, clone D3H11, CST), p90RSK (9355, clone 32D7, CST), PDGFRα (AF1062, R&D systems), p-S6 ribosomal protein Ser235/236 (4858, clone D57.2.2E, CST), S6 ribosomal protein (2217, clone 5G10, CST), PCNA (13110, clone D3H8P, CST), PGC1α (ab191838, abcam), SLC10A1 (MBS177905, MyBioSource), SM22α (ab14106, abcam), p-STAT3 Tyr705 (4113, clone M9C6, CST), STAT3 (4904, clone 79D7, CST), UCP1 (72298, clone E9Z2V, CST), anti-rabbit horseradish peroxidase (HRP) (7074, CST), anti-mouse HRP (7076, CST), anti-rabbit Alexa Fluor 488 (ab150077, abcam), anti-goat Alexa Fluor 488 (ab150129, abcam) and anti-rabbit Alexa Fluor 555 (ab150074, abcam). All commercially available antibodies have been validated by their manufacturer as indicated in their respective datasheet and/or website.

**Custom-made antibodies.** IgG (clone 11E10), anti-IL-11 (clone X203 for western blot and neutralizing studies), anti-IL11RA (clone X209 for neutralizing study) were manufactured by Genovac. The suitability of IgG (11E10) as a control antibody was validated previously[29]. X203 was validated for neutralization of human and mouse IL-11[29,38] and for western blot[38,50]. X209 was validated previously for neutralization of human and mouse IL11RA[38] and for western blot[38].

### Recombinant proteins

Recombinant human IL-11 (hIL11, Z03108, Genscript).

### Chemicals

Bovine serum albumin (BSA, A7906, Sigma), 16% formaldehyde (w/v), methanol-free (28908, Thermo Fisher Scientific), DAPI (D1306, Thermo Fisher Scientific), DMSO (D2650, Sigma), rapamycin, (9904, CST), Triton X-100 (T8787, Sigma), Tween-20 (170-6531, Bio-Rad) and U0126 (9903, CST),

### Ethics statements

All experimental protocols involving human subjects (commercial primary human cell lines) were performed in accordance with the ICH Guidelines for Good Clinical Practice. All participants provided written informed consent and ethical approvals have been obtained by the relevant parties as written in the datasheets provided by ScienCell from which primary human cardiac fibroblasts and primary human hepatocytes were commercially sourced.

Animal studies were carried out in compliance with the recommendations in the Guidelines on the Care and Use of Animals for Scientific Purposes of the National Advisory Committee for Laboratory Animal Research (NACLAR). All experimental procedures were approved (SHS/2019/1481 and SHS/2019/1483) and conducted in accordance with the SingHealth Institutional Animal Care and Use Committee (IACUC). Certified veterinarians were responsible for all animal experiment procedures according to the laws governing animal research in Singapore.

### Cell culture

Cells were grown and maintained at 37 °C and 5% $CO_2$. The growth medium was renewed every 2–3 days and cells were passaged at 80% confluence, using standard trypsinization techniques. All experiments were carried out at P3, unless otherwise specified. Cells were serum-starved overnight in basal media prior to stimulation with different treatment conditions (in the absence or presence of antibodies or inhibitors) and durations, as outlined in the main text or figure legends. All commercial cell lines were characterized by the company based on their morphology and by using immunofluorescence for cell-specific markers, as detailed in the respective product datasheet and certificate of analysis. Potential biological contaminants for HIV-1, HBV, HCV, mycoplasma, bacteria, yeast and fungi were confirmed negative as outlined in the certificate of analysis.

**Primary human cardiac fibroblasts.** Primary human cardiac fibroblasts (HCFs) (52-year-old male, 6330, lot 9580, ScienCell) were authenticated by their fibroblast morphology and phenotype, characterized by immunofluorescence staining for fibronectin and vimentin. Cell were grown and maintained in complete fibroblasts medium-2 (2331, ScienCell) supplemented with 5% foetal bovine serum (FBS, 0500, ScienCell), 1% fibroblasts growth supplement-2 (FGS-2, 2382, ScienCell) and 1% penicillin-streptomycin (P/S, 0513, ScienCell). For replicative senescence study, primary HCFs were serially passaged (from P4 to P14) in the absence or presence of a neutralizing IL11RA antibody (X209) or an IgG isotype control (11E10).

**Primary human hepatocytes.** Primary human hepatocytes were isolated from a 22-week-old foetus (5200, lot 34967, ScienCell) and authenticated by their hepatocyte morphology and phenotype, characterized by positive immunofluorescence for cytokeratin-18 and western blot for albumin. Following recovery from the initial thaw cycle, hepatocytes were seeded at a density $4 \times 10^5$ cells per well of a collagen-coated 6-well plate and maintained in hepatocyte medium (5201, ScienCell) which contains 2% FBS and 1% penicillin-streptomycin. Hepatocytes were then used directly for downstream experiments within 48 h of seeding.

### Olink proximity extension assay

Human hepatocytes were seeded at a density of $2.5 \times 10^5$ cells per well into 6-well plates. The culture supernatants were collected following stimulation with IL-11 (0, 6 and 24 h) and were sent to Olink Proteomics for proximity extension assays using the 92-protein inflammation panel. Zero-hour time points refer to time-matched, unstimulated controls that were cultured and collected in parallel with the other stated time points. In this experiment, IL-11 was added at different times to stimulate cells; for instance, at 15:00 on day 1 for the 24-h time point and at 09:00 on day 2 for the 6-h time point. Supernatants from the unstimulated control, 6 and 24-h time points were collected at the same time. The protein concentrations were expressed as normalized protein expression (NPX; $\log_2$ scale) and those proteins with concentrations below the limit of detection were excluded from analysis.

### Operetta high throughput phenotyping assay

HCFs (P4, P7, P10 and P14) were seeded in 96-well black CellCarrier plates (PerkinElmer) at a density of $6 \times 10^3$ cells per well either untreated or in the presence of IgG or X209. After reaching ~80% confluence, cells were fixed in 4% formaldehyde, and permeabilized with 0.1% Triton X-100. Non-specific sites were blocked with blocking solutions (0.5% BSA and 0.1% Tween-20 in PBS). Cells were incubated overnight (4 °C) with primary antibodies (p16 and p21) at a dilution of 1:500, followed by incubation with the appropriate Alexa Fluor 488 secondary antibodies (1:1,000, 1 h, room temperature). Cells were then counterstained with 1 µg ml$^{-1}$ DAPI in blocking solution. Antibodies and DAPI were diluted in blocking solutions. Each condition was imaged from duplicated wells

and a minimum of seven fields per well using Operetta high-content imaging system 1483 (PerkinElmer). The measurement of p16 and p21 fluorescence intensity per area (normalized to the number of cells) was performed with Columbus 2.9 (PerkinElmer).

## Seahorse assay

Primary HCFs were seeded into the Seahorse XF 96-well Cell Culture Microplate ($40 \times 10^3$ cells per well) and serum-starved overnight prior to stimulations. Seahorse measurements were performed on Seahorse XFe96 Extracellular Flux Analyzer (Agilent). XF Cell Mito Stress Test kit (103015-100, Agilent) and Seahorse XF Mito Fuel Flex Test kit (103260-100, Agilent) were used according to the manufacturer's protocol to measure the mitochondrial oxygen consumption rate and the percentage of fatty acid oxidation, respectively as described previously[51]. Seahorse Wave Desktop software (Ver 2.6.3) was used for report generation and data analysis.

## Animal models

All mice were housed at 21–24 °C with 40–70% humidity on a 12-h light/dark cycle and provided food and water ad libitum. Our mouse colonies hold specific pathogen free (SPF) status and undergo quarterly and annual tests for common pathogens. The room housing our animals is positive for murine norovirus and *Helicobacter*, and these particular pathogens are deemed acceptable within our SPF facility. Sample sizes were determined based on the authors' experience with the preliminary studies and by referencing a healthspan/lifespan study in mice[52] to detect a 20% change in phenotype between treatment groups or genotypes with 80% power ($\alpha = 0.05$). Sample sizes for experiments involving *Il11ra1*$^{-/-}$ and *Il11*$^{-/-}$ mice (and their respective wild-type mice) varied depending on animal availability. Mice were randomly allocated to experimental groups on the day of the treatment except for *Il11ra1*$^{-/-}$ and *Il11*$^{-/-}$ in which randomization was not applicable. Treatments or genotypes were not disclosed to investigators generating quantitative readouts during data collection but were revealed during the analysis. The mouse strains used in our study are described below.

**Il11ra1-deleted mice.** Male and female *Il11ra1*$^{+/+}$ (wild-type) and *Il11ra1*$^{-/-}$ mice[25] (B6.129S1-Il11ra$^{tm1Wehi}$/J, The Jackson Laboratory) were euthanized at 110 weeks of age for blood and tissue collection; 10–12 weeks old male and female mice of the respective genotypes were used as controls.

**Il11-deleted mice.** Male and female mice lacking functional alleles for *Il11* (*Il11*$^{-/-}$), which were generated and characterized previously[31,50], and their wild-type counterparts were euthanized at 10–12 weeks of age (young controls) and 104–108 weeks of age (old mice).

**Il11-EGFP reporter mice.** Young (10-week-old) and old (100-week-old) transgenic mice (C57BL/6 J background) with *EGFP* knocked into the *Il11* gene (*Il11-EGFP* mice, Cyagen Biosciences)[29] were euthanized for immunofluorescence staining studies of liver, gastrocnemius and vWAT. Old wild-type littermates were used as aged negative controls.

**In vivo administration of anti-IL-11.** Male and female C57BL/6 J mice (Jackson Laboratory) were randomized prior to receiving either no treatment, anti-IL-11 (X203) or IgG (11E10). X203 or 11E10 (40 mg kg$^{-1}$, every 3 weeks) were administered by intraperitoneal injection, starting from 75 weeks of age for a duration of 25 weeks; mice were then euthanized at 100 weeks of age.

## Lifespan studies

Lifespan studies involved two distinct experimental groups (male and female) (1) C57BL/6 J mice (Jackson Laboratory) aged 75 weeks that received monthly injections of either anti-IL-11 (X203) or IgG (11E10) at a dosage of 40 mg kg$^{-1}$; and (2) wild-type and *Il11*$^{-/-}$ mice. Mice were inspected daily and medicated for non-life-threatening conditions by

an experienced veterinarian, as needed. The principal experimental endpoint was age of death, which was recorded when mice were found dead or at the time of euthanasia if they were deemed severely moribund (or unlikely to survive longer than 48 h) at the time of inspection, according to previously described criteria[10]. Mice with gross tumours, when present, were monitored for tumour progression and euthanized when tumours developed >1.5 cm in size (at largest dimension) or when tumours become ulcerated, infected or interfered with mobility as permitted by the local SingHealth IACUC. These limits were not exceeded in any of the experiments. Gross examination was conducted at autopsy following natural death or euthanasia of mice to observe and document the presence of visible tumours in the larger body organs and to record any noticeable gross appearances.

## GTT and ITT

Mice were fasted for 6 h prior to baseline blood glucose measurement. For GTT, mice were injected intraperitoneally with 20% glucose at 2 mg per g lean mass. For ITT, mice were injected intraperitoneally with recombinant human insulin at 1.2 mU per g body weight. Both glucose and insulin were diluted in sterile DPBS. Blood glucose concentrations were then measured at 15, 30, 60, 75 and 90 min after glucose or insulin administration for GTT or ITT, respectively. Blood was collected via tail snip and Accu-Chek blood glucometer was used for blood glucose measurements.

## Echo MRI

Mouse body composition (total body fat and lean mass measurements) was performed 1 day prior to GTT/ITT or euthanasia by Echo MRI analysis using 4in1 Composition Analyzer for live small animals (Echo Medical Systems).

## Frailty scoring

Frailty scoring was performed, with observers blinded to treatment, at the start of the experiment or 1–2 days prior to euthanasia using a 27-point frailty scoring system[15]. Body temperatures were recorded by rectal thermometry using Kimo Thermocouple Thermometer (TK110, Kimo).

## Grip strength assessment

A digital grip strength meter (BIO-GS3, BIOSEB) was used to measure full body (4 limbs) and forelimb (both forepaws) grip strengths, as per the manufacturer's instruction. Mice were allowed to rest for at least 1 h between the two tests. The average of 3 readings of maximal average force exerted by each mouse on the grip strength meter was used for analysis.

## Measurement of whole-body metabolic parameters

Whole-body metabolic parameters for IgG and X203-treated (antibodycohort), and wild-type and IL-11-knockout (KO cohort) mice were assessed by open-circuit indirect calorimetry. Animals were single-housed in the PhenoMaster automated home-cage system (TSE Systems) at a temperature of 22°C and in a humidity-controlled environment with a 12-h light/dark cycle. Parameters including oxygen consumption ($VO_2$), carbon dioxide production ($VCO_2$), food intake, and locomotor activity were measured simultaneously at 1-min time intervals. RER was calculated using the $VCO_2/VO_2$ ratio. Locomotor activity was divided into horizontal plane locomotor activities, defined as the total number of infrared beam breaks in the $x$ and $y$ axis (counts). Mice were monitored for 5 consecutive overnight periods including an acclimatization period during the first light/dark cycle (day 0–1), which was not used for analysis. For both antibody and KO cohorts, the control (IgG or wild-type) group ($n = 10$) and intervention (X203 or IL-11-knockout) group ($n = 10$) were divided equally into two consecutive monitoring sessions. Baseline RER comparison was made using measurements from the second light/dark cycle (day 1–2). Animals were

given ad libitum access to food and water except during test phases introduced after day 2 where food access was restricted to assess the resting metabolic rate (measured at thermoneutrality (28°C)) and adaptation to fasting (12 h).

## Bomb calorimetry

To measure energy content in mouse stool, bomb calorimetry was performed by the core service at Department of Food Science and Technology, National University of Singapore. All faecal samples were collected and stored at −80 °C prior to measurement. Approximately 0.9 g of faecal samples were placed into a combustion bag in which a cotton thread and benzoic acid tablets were used as combustion aid. The gross calorie content was then determined using the IKA C5003 Control bomb calorimeter on the isoperibolic mode, with C5001 cooling system and oxygen gas supplied. Data was derived as a single point reading ($n = 1$). The average relative error ranges from 0.07%–0.59%.

## Colorimetric and enzyme-linked immunosorbent assays (ELISA)

The levels of ALT, AST, cholesterol and IL-6 in mouse serum were measured using Alanine Transaminase Activity Assay Kit (ab105134, abcam), Aspartate Aminotransferase Activity Assay Kit (ab105135, abcam), Cholesterol Assay Kit (ab65390, Abcam), and Mouse IL-6 Quantikine ELISA Kit (M6000B, R&D Systems), respectively. The levels of triglyceride in mouse livers and serum were measured using Triglyceride Assay Kit (ab65336, Abcam). Total collagen content in mouse livers, gastrocnemius, and vWAT were measured using Quickzyme Total Collagen assay kit (QZBtotco15, Quickzyme Biosciences). The levels of IL-6, IL-8 and IL-11 in equal volumes of cell culture media collected from experiments with primary human cells were quantified using Human IL-8/CXCL8 Quantikine ELISA Kit (D8000C, R&D Systems), Human IL-6 Quantikine ELISA Kit (D6050, R&D Systems), Human IL-11 Quantikine ELISA kit (D1100, R&D Systems). All ELISA and colorimetric assays were performed according to the manufacturer's protocol. Triglyceride Assay Kit (ab65336, Abcam),

## Immunoblotting

Western blots were carried out on total protein extracts from liver, gastrocnemius, and vWAT tissues, which were homogenized in RIPA Lysis and Extraction Buffer (89901, Thermo Fisher Scientific) containing protease and phosphatase inhibitors (A32965 and A32957, Thermo Fisher Scientific). Protein lysates were separated by SDS–PAGE, transferred to PVDF membranes, blocked for 1 h with 3% BSA, and incubated overnight with primary antibodies (1:1,000 in TBST). This study was conducted over six years, and western blots were performed on many tissues, the smallest of which provided limited protein for blotting. To conserve antibody usage and maximize data output, membranes were often cut at the appropriate molecular weight markers and probed with different antibodies. In all instances, equal loading of protein lysates per membrane was ensured. Protein bands were visualized using SuperSignal West Femto Maximum Sensitivity Substrate detection system (34096, Thermo Fisher Scientific) with the appropriate HRP secondary antibodies (1:1,000 in TBST). Raw uncropped blots are provided in Supplementary Fig. 1 and semi-quantitative densitometry analyses are provided in Supplementary Fig. 2.

## Quantitative PCR with reverse transcription

Total RNA was extracted from cells or snap-frozen tissues using TRIzol Reagent (15596026, Thermo Fisher Scientific) and RNeasy Mini Kit (74104, Qiagen). PCR amplifications were performed using iScript cDNA Synthesis Kit (1708891, Bio-Rad). Gene expression analysis was performed with QuantiNova SYBR Green PCR Kit (208056, Qiagen) technology using StepOnePlus (Applied Biosystem). Expression data were normalized to *GAPDH* mRNA expression and fold change was calculated using $2^{-\Delta\Delta Ct}$ method. The primer sequences are provided in the Supplementary Table 8.

## Telomere length and mitochondrial copy number quantification

DNA from HCFs (P4 and P14) and snap-frozen liver, gastrocnemius, and vWAT was extracted with the E.Z.N.A. Tissue DNA Kit (D3396-02, Omega Bio-tek) according to the manufacturer's protocol. Telomere length and mitochondrial copy number for HCFs were evaluated by quantitative PCR with reverse transcription (RT–qPCR) with the Relative Human Telomere Length Quantification qPCR Assay Kit (8908, ScienCell) and Relative Human Mitochondrial DNA copy number Length Quantification qPCR Assay Kit (8938, ScienCell), respectively. Similarly, the telomere length and mitochondrial copy number for mouse tissues were evaluated by RT–qPCR with the Relative Mouse Telomere Length Quantification qPCR Assay Kit (M8908, ScienCell) and Relative Human Mitochondrial DNA copy number Length Quantification qPCR Assay Kit (M8938, ScienCell), respectively.

## Histology

Investigators performing histology and analysis were blinded to the genotype and treatment group.

**Haematoxylin and eosin staining.** Mouse vWAT were fixed in 10% neutral-buffered formalin (NBF) for 48 h, embedded in paraffin, cut into 4-µm sections followed by haematoxylin and eosin staining according to the standard protocol. Lipid droplet areas were quantified by ImageJ (version 1.53t, NIH) with the adipocytes tools plugin (https://github.com/MontpellierRessourcesImagerie/imagej_macros_and_scripts/wiki/Adipocytes-Tools) from 5 randomly selected fields at 200× magnification in vWAT images per mouse, and 5 mice per group were assessed. The mean value of lipid droplet areas per field was plotted for the final data presentation.

**Immunohistochemistry.** Four-micrometre mouse vWAT sections were dewaxed with histoclear and a gradient ethanol wash, followed by permeabilization using 1% Triton X-100 for 10 min and antigen retrieval process with Reveal Decloaker (RV1000M, Biocare Medical) using a double boiler method at 110 °C for 20 min. Slides were allowed to cool in the container together with the Reveal Decloaker solution for 10 min under running water. Double blocking was achieved with (1) $H_2O_2$ for 10 min and (2) 2.5% normal horse serum for 1 h (S-2012, Vector Labs). vWAT sections were incubated overnight at 4 °C with primary antibody (CD68, 1:100 in PBST) and visualized by probing with Horse Anti-Rabbit IgG Polymer Kit (MP-7401, Vector Labs) for 1 h at 37 °C and ImmPACT DAB Peroxidase Substrate Kit (SK-4105, Vector Labs). Haematoxylin (H-3401, Vector Labs) was used to counterstain the nuclei prior to imaging by light microscopy (Olympus IX73).

**Immunofluorescence.** Young (10-week) and aged (100-week) *Il11*EGFP/+ and aged wild-type *Il11*+/+ mice underwent perfusion-fixation with PBS and 4% paraformaldehyde for multi-organ collection at terminal euthanasia. Mouse liver, vWAT and gastrocnemius were further fixed in 4% paraformaldehyde at 4 °C and serial 15–30% sucrose dehydration over 48 h before they were cryo-embedded in OCT medium. 5 µm sections were heat antigen retrieved using Reveal Decloaker (RV1000M, Biocare), permeabilized with 0.5% Triton X-100, and blocked with 5% normal horse serum before probing with primary antibodies diluted in 2.5% normal horse serum at 4 °C overnight. The antibody dilutions used for immunofluorescence studies are as follows: adiponectin, GFP, PDGFRα and SLC10A1 (1:100); CD31, FHL1 and SM22α (1:200). Alexa Fluor-conjugated secondary antibodies (1:300 in 2.5% normal horse serum) were incubated for 2 h at room temperature for visualization. Autofluorescence was quenched with 0.1% Sudan Black B for 20 min. DAPI was included for nuclear staining before mounting and sealed. Photomicrographs were randomly captured by researchers blinded to the strain and age groups.

## RNA-seq libraries

Total RNA was isolated from liver, fat and skeletal muscle of mice receiving either IgG or X203 using RNeasy Mini Kit (74104, Qiagen) and quantified using Qubit RNA Broad Range Assay Kit (Q10210, Thermo Fisher Scientific). RNA quality scores (RQS) were assessed using the RNA Assay (CLS960010, PerkinElmer) and DNA 5 K/RNA/CZE HT Chip (760435, PerkinElmer) on a LabChip GX Touch HT Nucleic Acid Analyzer (CLS137031, PerkinElmer). TruSeq Stranded mRNA Library Prep kit (20020594, Illumina) was used to assess transcript abundance following the manufacturer's instructions. In brief, poly(A) + RNA was purified from 1 μg of total RNA with RQS > 6, fragmented, and used for cDNA conversion, followed by 3′ adenylation, adapter ligation, and PCR amplification. The final libraries were quantified using Qubit DNA Broad Range Assay Kit (Q32853, Thermo Fisher Scientific) according to the manufacturer's guide. The average fragment size of the final libraries was determined using DNA 1 K/12 K/Hi Sensitivity Assay LabChip (760517, PerkinElmer) and DNA High Sensitivity Reagent Kit (CLS760672, PerkinElmer). Libraries with unique dual indexes were pooled and sequenced on partial lanes targeting ~50 M reads per sample on a HiSeq or a NovaSeq 6000 sequencer (Illumina) using 150-bp paired-end sequencing chemistry.

## Data processing and analysis for RNA-seq

Fastq files were generated by demultiplexing raw sequencing files (.bcl) with Illumina's bcl2fastq v2.20.0.422 with the --no-lane-splitting option. Low quality read removal and adapter trimming was carried out using Trimmomatic V0.36 with the options ILLUMINACLIP: <keepBothReads > =TRUE MAXINFO:35:0.5 MINLEN:35. Reads were mapped to the *Mus musculus* GRCm39 using STAR v.2.7.9a with the options --outFilterType BySJout --outFilterMultimapNmax 20 --alignSJoverhangMin 8 --alignSJDBoverhangMin 1 --outFilterMismatchNmax 999 --alignIntronMin 20 --alignIntronMax 1000000 --alignMatesGapMax 1000000 in paired-end, single pass mode. Read counting at the gene-level was carried out using subread v.2.0.3: -t exon -g gene_id -O -s 2 -J -p -R -G. The Ensembl release 104 *M. musculus* GRCm39 GTF was used as annotation to prepare STAR indexes and for FeatureCounts. Principal component analysis clustered samples into tissue-types and conditions. Outlier samples that did not cluster with the expected group were removed. Differentially expressed genes were identified using R v4.2.0 using the Bioconductor package DESeq2 v1.36.0 using the Wald test for comparisons. IgG samples were used as the reference level for comparison with anti-IL-11 (X203) samples for vWAT, liver, and gastrocnemius. Mitocarta v3.0 gene list was downloaded and TPM values in Fat IgG and anti-IL-11 samples were plotted using pheatmap R package for genes which had TPM ≥ 5 in at least one condition. Gene set enrichment analysis was carried out using the fgsea v.1.22.0 R package for MSigDB Hallmark (msigdbr v.7.5.1) and MitoCarta v3.0 gene sets with 100,000 iterations. The 'stat' value quantified by DESeq2 was used to rank the genes, as an input for the enrichment analysis.

## Statistical analysis

Statistical analyses were performed using GraphPad Prism software (version 10). Datasets were tested for normality with Shapiro–Wilk tests. For normally distributed data, two-tailed Student's *t*-tests or one-way ANOVA were used for analysing experimental setups requiring testing of two conditions or more than two conditions, respectively. *P* values were corrected for multiple testing according to Dunnett (when several experimental groups were compared to a single control group) or Tukey (when several conditions were compared to each other within one experiment) tests. Non-parametric tests (Kruskal–Wallis with Dunn's correction in place of ANOVA and Mann–Whitney *U* test in place of two-tailed Student's *t*-tests) were conducted for non-normally distributed data. Comparison analysis for two parameters from two different groups were performed by two-way ANOVA and corrected with Sidak's multiple comparisons. Two-way repeated measures ANOVA (Geisser–Greenhouse correction) with Sidak's multiple comparisons was applied to temporal sampling in paired subjects for GTT, ITT and body weight. Individual endpoint frailty indices were ranked and compared using two-tailed Mann–Whitney test to compare (1) old *Il11*$^{-/-}$ versus wild-type females; (2) old *Il11*$^{-/-}$ versus wild-type males; or (3) IgG versus X203 groups in females, and the Kruskal–Wallis test with Dunn's multiple comparisons of untreated, IgG and X203 treatment groups in males. The two-population proportions analysis (two-tailed) was used for comparing the difference in the proportion of cancer occurrence and seminal vesicle dilatation between two groups. The criterion for statistical significance was set at *P* < 0.05. For the lifespan studies, differences in survival between the experimental groups (*Il11*$^{-/-}$ versus wild-type or X203 versus IgG) were compared using the Kaplan–Meier method implemented in IBM SPSS (release 29.0.1.0), and statistical significance (*P* value) was assessed by means of the log-rank (Mantel–Cox) test. In addition to the log-rank test (that gives equal weight to all time points), we used the Wilcoxon test (that gives more weight to deaths at early time points), which provided significant results in all comparisons, therefore yielding a similar conclusion to reject the null hypothesis. Both survival comparison methods are non-parametric tests based on the chi-square statistic and provide two-tailed *P* values. The complete list of exact p-values and terms for supporting statistical information is provided in Supplementary Table 9.

## Reporting summary

Further information on research design is available in the Nature Portfolio Reporting Summary linked to this article.

## Data availability

All data are available within the Article or Supplementary Information. The RNA-seq data reported in this paper are available on the Short Read Archive with Bioproject ID: PRJNA939262. Datasets used for analysis in this study are as follows: Ensembl release 104 *M. musculus* GRCm39 gene annotations (GRCm39, https://asia.ensembl.org/info/data/ftp/index.html), MSigDB Hallmark (v.7.5.1, https://www.gsea-msigdb.org/gsea/msigdb/human/collections.jsp) and MitoCarta (v3.0, https://www.broadinstitute.org/mitocarta/mitocarta30-inventory-mammalian-mitochondrial-proteins-and-pathways). Source data are provided with this paper.

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

**Acknowledgements** The authors thank M. M. Dy Varela, C. H. Lee and all other members of the Duke–NUS vivarium team for their invaluable assistance in monitoring and overseeing the care for our mice used in this study; and Enleofen and Boehringer Ingelheim for the permission to use X203 in this study. This research is supported by the National Medical Research Council (NMRC) MOH-STaR21nov-0003 (S.A.C.), NMRC Centre Grant to the NHCS (S.A.C.), MOH-CIRG18nov-0002 (S.A.C.), Tanoto Foundation (S.A.C.), Leducq Foundation (S.A.C.), NMRC/OFYIRG/0053/2017 (A.A.W.), NMRC MOH-OFIRG21nov-0006 (A.A.W.), NMRC MOHOFLCG22may-0003 (A.A.W.), Khoo Foundation (A.A.W. and E.A.), and Goh Foundation (S.A.C. and A.A.W.). N.H. is supported by Leducq Foundation 16CVD03, ERC advanced grant under the European Union Horizon 2020 Research and Innovation Program (AdG788970), and Deutsche Forschungsgemeinschaft (DFG-German Research Foundation) SFB 1470 HFpEF. W.-W.L. is supported by A*STAR AME YIRG (A2084c0157). B.K.S. is supported by MOH-OFIRG19may-0002. The D.J.W. lab is supported by the Wellcome Trust Grant 098565/Z/12/Z and the MRC grant MC-A654-5QB40. The J.G. lab is supported by MRC (MC-U120085810) and CRUK (C15075/A28647) grants. The D.C. lab is supported by the MRC grant MC-A654-5QB10. The S.A.C. laboratory receives core grant support from the MRC LMS and is also supported by the British Heart Foundation's Big Beat Challenge award to CureHeart (BBC/F/21/220106).

**Author contributions** A.A.W. and S.A.C. conceived, designed, funded and provided supervision for the study. A.A.W., W.-W.L., S.V., B.C., D.M.C., J.W.T.G, B.K.S., J.T., S.S., E.A., B.L.G., M.S. and M.T. performed in vitro and in vivo studies, biochemistry and molecular biology experiments.

W.-W.L., S.Y.L. and C.X. performed histology analysis. C.J.P. performed RNA-seq. R.L. and L.H performed the phenomaster study. A.A.W., W.-W.L., S.C., R.L., B.K.S., N.A.S., N.H., E.P., L.H. and S.A.C. analysed and interpreted the data. D.J.W., J.G. and D.C. provided intellectual input. A.A.W., W.-W.L. and S.A.C. prepared the manuscript with input from co-authors.

**Competing interests** A.A.W., B.C., B.K.S., S.S. and S.A.C. are co-inventors of a patent family that includes: WO2022090509A (methods to extend healthspan and treat age-related diseases) and WO2018109174 (IL-11 antibodies). S.S. and S.A.C. are co-founders and shareholders of Enleofen Bio Pte Ltd and VVB Bio Pte Ltd. A.A.W. had consulted for VVB Bio on work unrelated to the study presented here. J.G. has acted as a consultant for Unity Biotechnology, Geras Bio, Myricx Pharma and Merck KGaA. Pfizer and Unity Biotechnology have funded research in the J.G. laboratory unrelated to the work presented here. J.G. owns equity in Geras Bio. J.G. is a named inventor in MRC and Imperial College patents related to senolytic therapies (the patents are not related to the work presented here). The other authors declare no competing interests.

**Additional information**
**Correspondence and requests for materials** should be addressed to Anissa A. Widjaja or Stuart A. Cook.

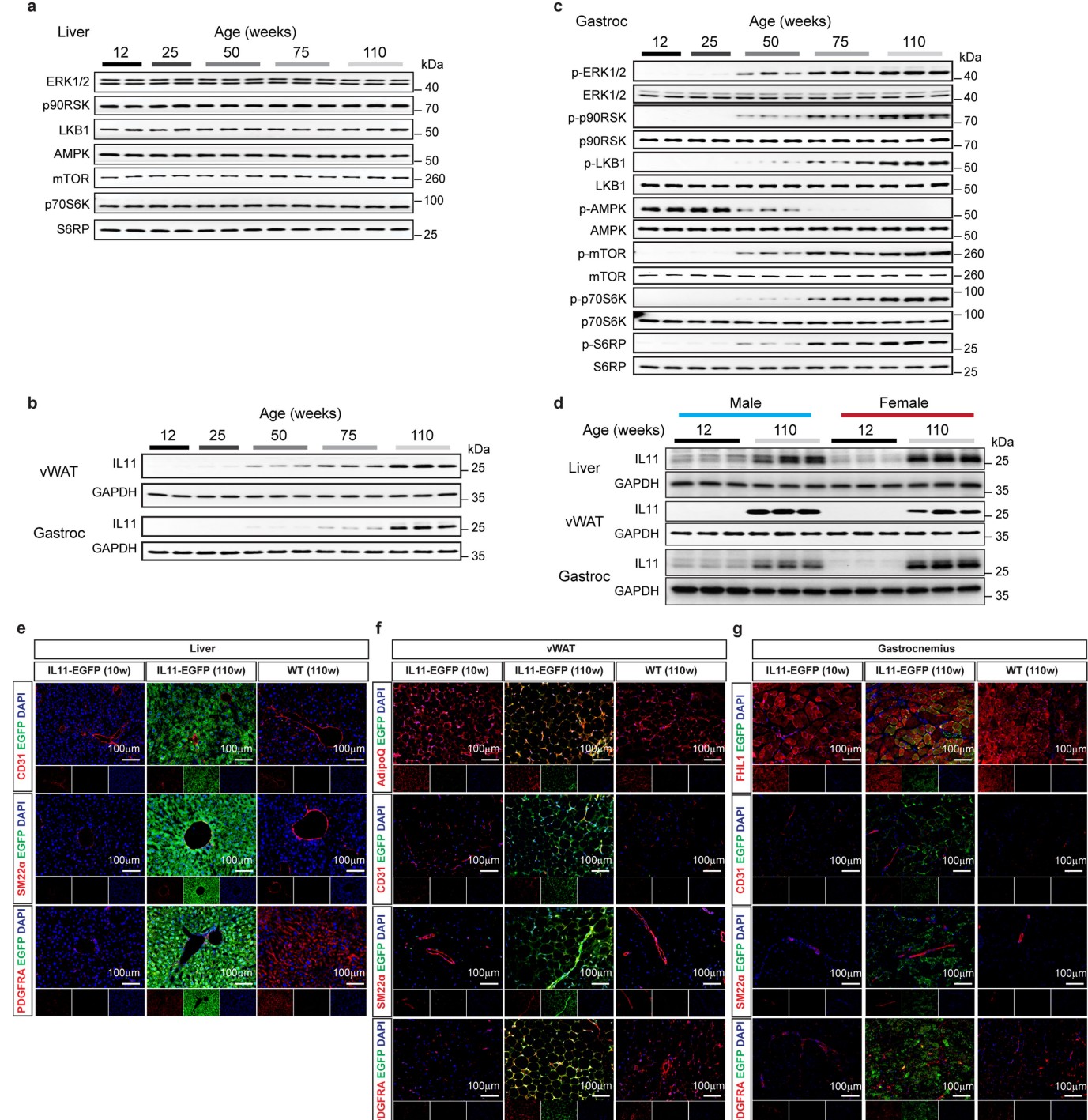

**Extended Data Fig. 1 | Age-dependent expression of IL11 in varied cell types across tissues. a** Western blots (WB) of total ERK1/2, p90RSK, LKB1, AMPK, mTOR, p70S6K, and S6RP in livers from 12, 25, 50, 75, and 110-week-old male mice for the respective phosphoproteins shown in Fig. 1b. **b** WB of IL11 and GAPDH in visceral gonadal white adipose tissue (vWAT) and gastrocnemius from 12, 25, 50, 75, and 110-week-old male mice (n = 5/group). **c** WB of p-ERK1/2, p-p90RSK, p-LKB1, p-AMPK, p-mTOR, p-p70S6K, p-S6RP, and their respective total proteins in gastrocnemius from 12, 25, 50, 75, and 110-week-old male mice

(n = 5/group). **d** WB of IL11 and GAPDH in the liver, vWAT and gastrocnemius from 12-week-old and 110-week-old male and female mice (n = 3/group). **e-g** Representative immunofluorescence images (scale bars, 100 µm) of EGFP expression in the livers, vWAT, and gastrocnemius, colocalized with parenchymal cell markers Adiponectin (AdipoQ) in vWAT and Four and a half LIM domains (FHL1) in gastrocnemius, endothelial cells (CD31), smooth muscle transgelin (SM22α), and pan-fibroblast marker (PDGFRα) of 10 and 110-week old *Il11-EGFP* mice (representative dataset from n = 3/group).

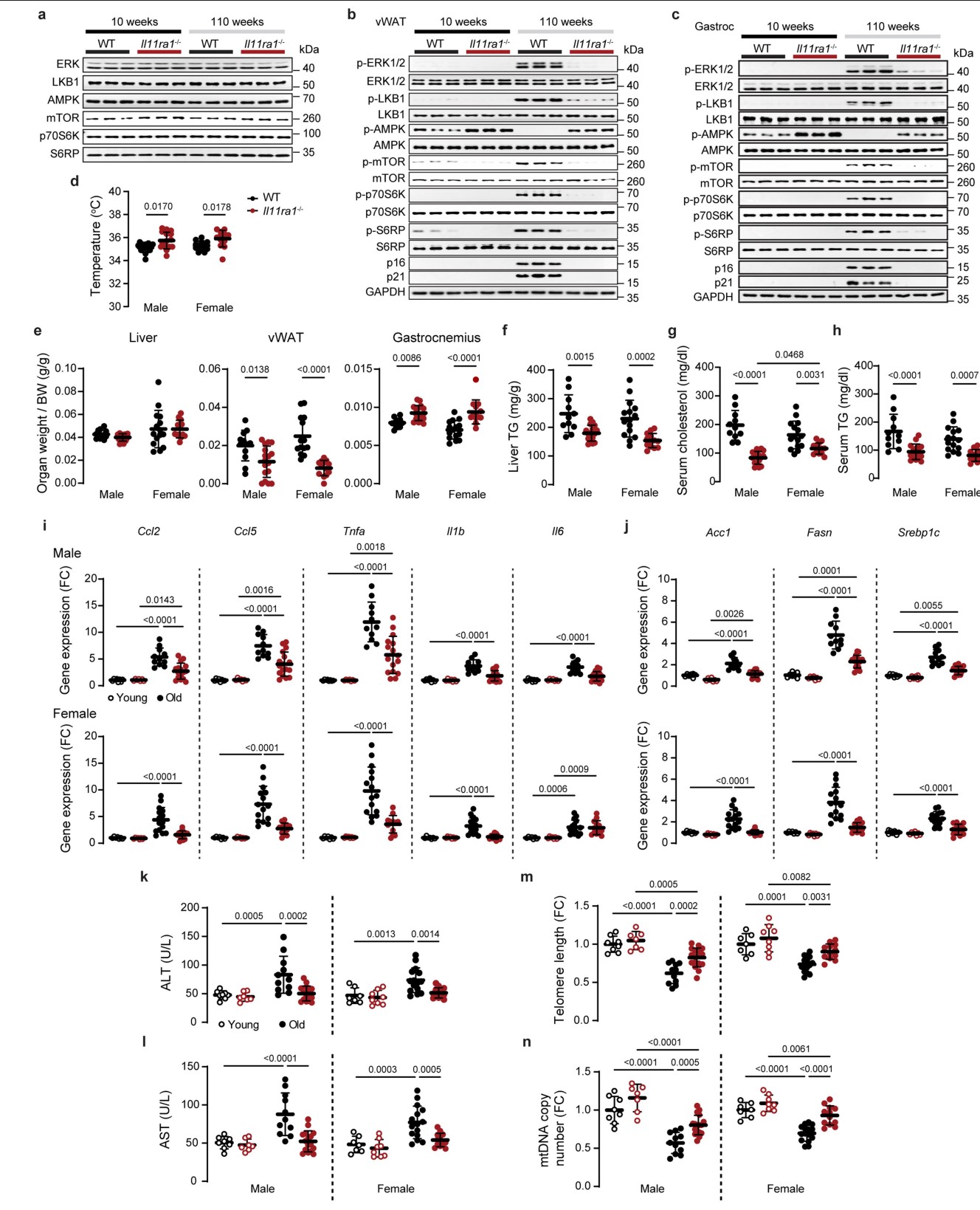

**Extended Data Fig. 2** | See next page for caption.

**Extended Data Fig. 2 | Beneficial signalling, metabolic, inflammation and ageing biomarker effects associated with *Il11ra1* deletion. a** WB of total proteins in livers for Fig. 1e. WB showing the activation status of ERK1/2, p90RSK, LKB1, AMPK, mTOR, p70S6K, S6RP, and protein expression levels of p16, p21 and GAPDH in **b** vWAT and **c** gastrocnemius from 10 and 110-week-old male WT and *Il11ra1⁻/⁻* mice (n = 5/group). **d** body temperatures, **e** indexed weights of liver, vWAT and gastrocnemius, and **f** the levels of liver triglycerides (TG), **g** serum cholesterol, and **h** serum triglycerides of 110-week-old male and female WT and *Il11ra1⁻/⁻* mice. Relative gene expression levels of **i** *Ccl2*, *Ccl5*, *Tnfα*, *Il1β*, *Il6*, **j** *Acc*, *Fasn* and *Srebp1c* in livers, and serum levels of **k** ALT and **l** AST in young and old male and female WT and *Il11ra1⁻/⁻* mice. Gastrocnemius **m** telomere length and **n** mitochondria DNA (mtDNA) copy number from young and old male and female WT and *Il11ra1⁻/⁻* mice. **d-l** Data are shown as mean ± SD. **d-n** young male WT, n = 8 (**i-n**, except for **i** (*Acc* and *Fasn*), n = 7); young male *Il11ra1⁻/⁻*, n = 7 (**i-n**, except for **i** (*Acc* and *Fasn*), n = 8); old male WT, n = 11 (**e** (liver), **f-n**), n = 12 (**d**, **e** (vWAT and gastrocnemius); old male *Il11ra1⁻/⁻*, n = 15 (**e** (liver)), n = 16 (**d**, **f-l**), n = 17 (**e** (vWAT and gastrocnemius), **i** (*Ccl5*), **m-n**); young female WT, n = 7; young female *Il11ra1⁻/⁻*, n = 8; old female WT, n = 14 (**e** (liver and vWAT)), n = 15 (**d**, **e** (gastrocnemius), **f-l**); old female *Il11ra1⁻/⁻*, n = 12 (**m-n**), n = 13 (**d-l**); two-way ANOVA with Sidak's correction. For gel source data, see Supplementary Fig. 1. BW: body weight; FC: fold change.

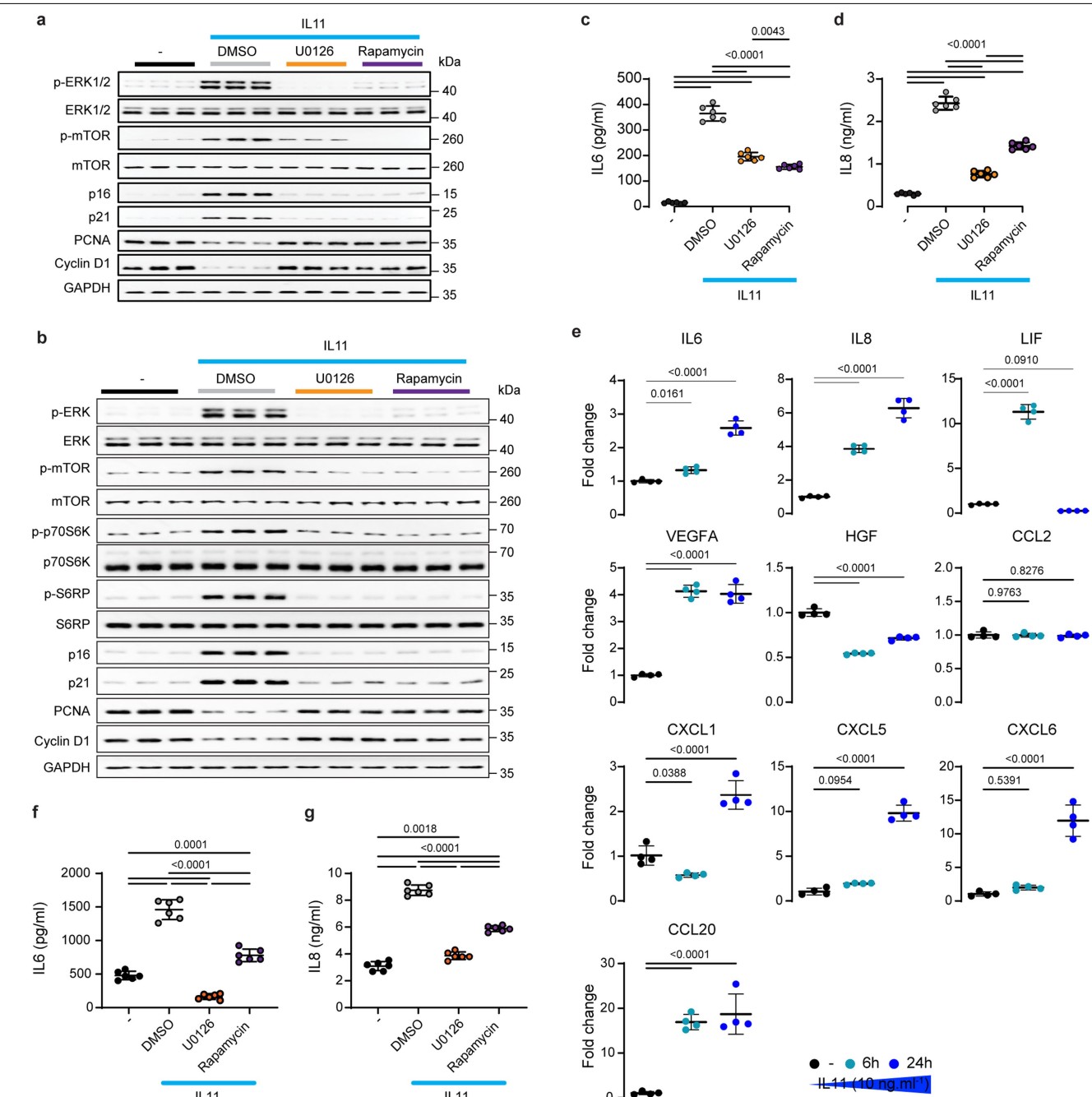

**Extended Data Fig. 3 | IL11 causes ERK and mTORC1-dependent senescence and senescence-associated secretory phenotypes. a-d**, **f,g** Data for IL11 (24 h)-stimulated primary human cells in the presence of either DMSO, U0126, or rapamycin (n = 6/group). **a-b** WB showing the activation status of ERK1/2, mTOR, p16, p21, Cyclin D1, and PCNA protein expression by WB from IL11-stimulated **a** primary human cardiac fibroblasts (HCFs) and **b** hepatocytes. Levels of secreted **c** IL6 and **d** IL8 by ELISA from HCF supernatant. **e** Relative levels of IL6, IL8, LIF, VEGFA, HGF, CCL2, CXCL1, CXCL5, CXCL6, and CCL20 in

the supernatant of IL11-stimulated primary human hepatocytes (6 and 24 h) as measured by Olink proximity extension assay (n = 4/group). Concentrations of **f** IL6 and **g** IL8 in the hepatocyte supernatant (as measured by ELISA). **a-d**, **f-g** IL11 (5 ng/ml for HCF, 10 ng/ml for hepatocytes), U0126 (10 μM), rapamycin (10 nM). **c-g** Data are shown as mean ± SD. **c**, **d**, **f**, **g** One-way ANOVA with Tukey's correction; **e** one-way ANOVA with Dunnett's correction. For gel source data, see Supplementary Fig. 1.

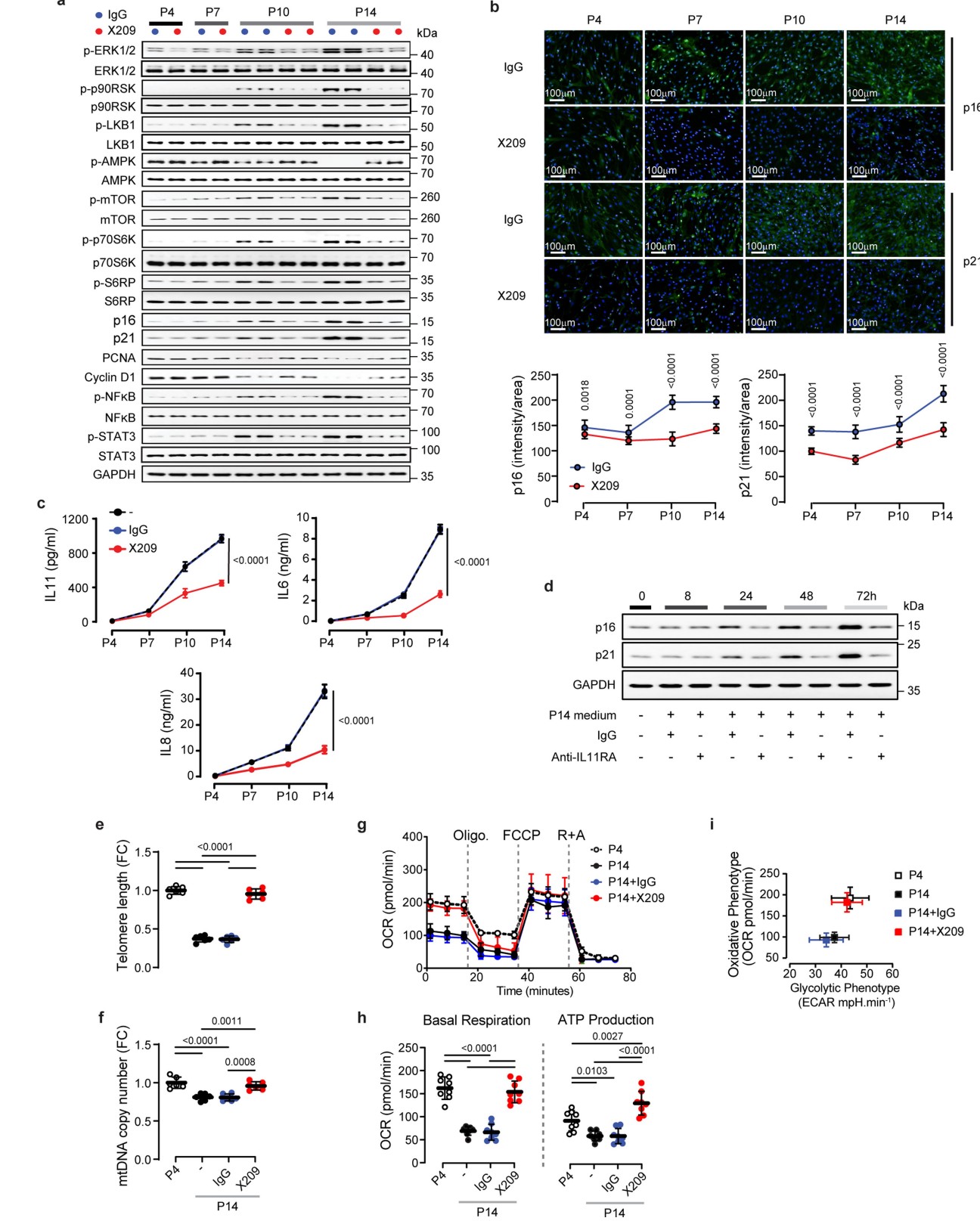

**Extended Data Fig. 4** | See next page for caption.

**Extended Data Fig. 4 | Inhibition of IL11 signalling reduces replicative senescence, inflammation, ageing biomarkers, and metabolic decline in human cardiac fibroblasts. a**-**c** Data for HCF at passage 4 (P4), 7, 10, and 14 that had been passaged in the presence of either IgG or anti-IL11RA (X209; 2 μg/ml) from P2. **a** WB of total and p-ERK1/2, p-p90RSK, p-LKB1, p-AMPK, p-mTOR, p-p70S6K, p-S6RP, p-NFκB, p-STAT3, p16, p21, PCNA, Cyclin D, and GAPDH (n = 6/group). **b** Immunofluorescence images (scale bars, 100 μm; representative datasets from n = 7/group) and quantification of intensity/area (n = 14/group) for p16 and p21 staining. **c** IL11, IL6 and IL8 levels in the supernatant based on ELISA (n = 6/group). **d** WB showing the expression levels of p16, p21, and GAPDH from HCFs P4 that were stimulated for 8, 24, 48, and 72 h with media collected from HCFs P14 that had been grown and passaged in the presence of either IgG or anti-IL11RA (X209; 2 μg/ml) from P2 (n = 4/group). **e** Telomere length (n = 6/group) and **f** mtDNA copy number (n = 6/group) and seahorse assay (n = 8/group) showing **g** mitochondrial oxygen consumption rate (OCR), **h** changes in OCR during basal respiration and ATP production states, and **i** oxidative and glycolytic energy phenotypes at baseline in HCFs P4 and P14 either untreated or in the presence of either IgG or anti-IL11RA (X209; 2 μg/ml). **b**, **c**, **e**-**i** Data are shown as mean ± SD. **b**, **c** Two-way ANOVA with Sidak's correction, **e**, **f**, **h** one-way ANOVA with Tukey's correction. For gel source data, see Supplementary Fig. 1.

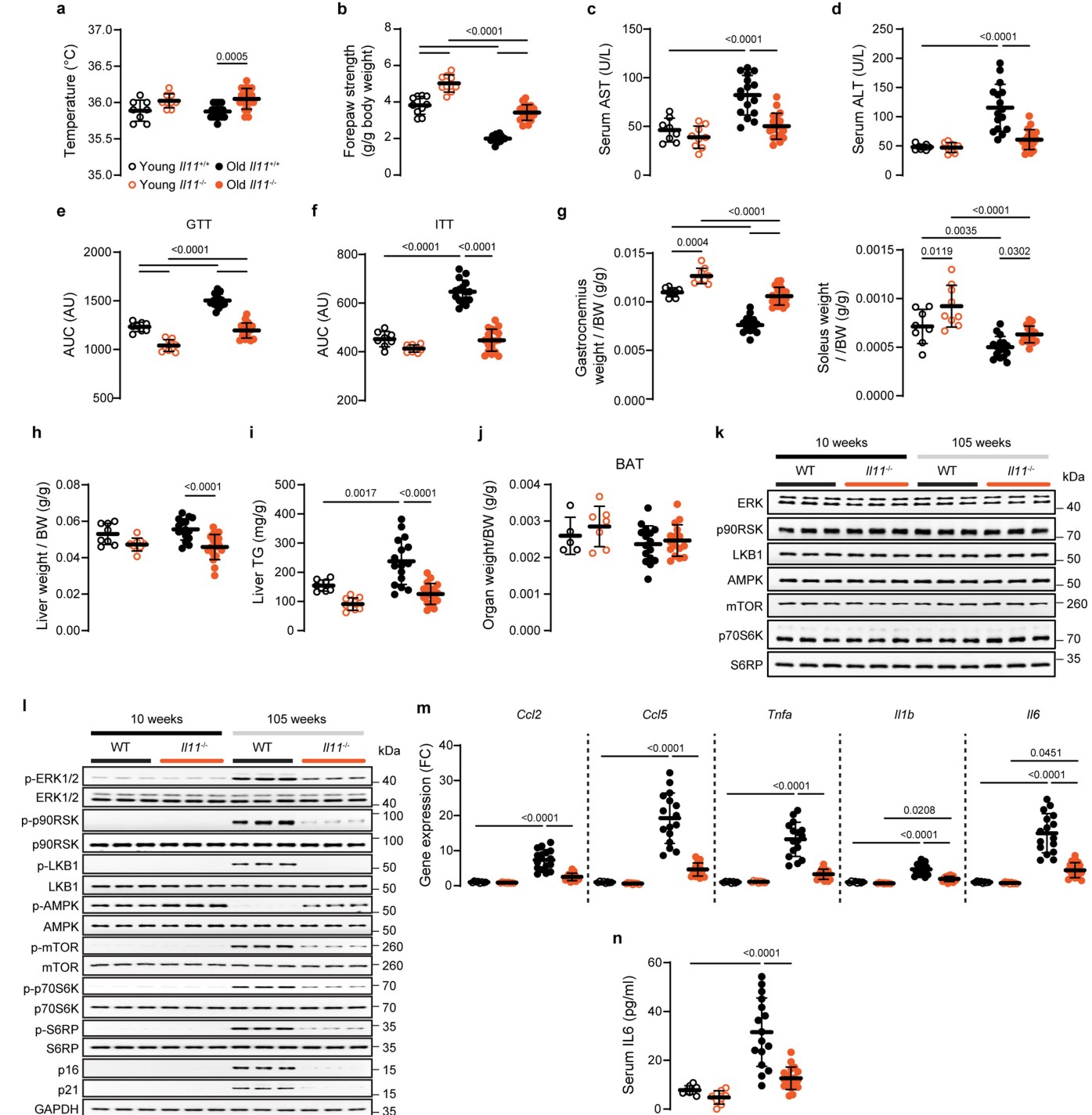

**Extended Data Fig. 5 | Female *Il11⁻/⁻* mice are protected from age-associated frailty and inflammation and have advantageous metabolic profiles. a** Body temperatures, **b** front paw grip strength, serum levels of **c** ALT, **d** AST, area under the curves (AUC) of **e** glucose tolerance tests (GTT) and **f** insulin tolerance tests (ITT), weights of **g** skeletal muscle (gastrocnemius and soleus) and **h** liver (normalised/indexed to BW), **i** liver triglyceride (TG) levels, **j** indexed brown adipose tissues (BAT) weight, **k** WB of total proteins for the respective phospho proteins in vWAT as shown in Fig. 2j, **l** WB showing ERK1/2, mTOR, p70S6K, and S6RP activation and p16, p21, and GAPDH protein expression levels (n = 6/group) in gastrocnemius, **m** relative pro-inflammatory gene expression (*Ccl2, Ccl5, Tnfα, Il1β* and *Il6*) levels in vWAT, and **n** serum IL6 levels from young (12-week-old) and old (105-week-old) female WT and *Il11⁻/⁻* mice. **a-j**, **m-n** Data are shown as mean ± SD, two-way ANOVA with Sidak's correction (young WT, n = 5 (**j**), n = 8 (**a**, **c-i**, **m-n**), n = 10 (**b**); young *Il11⁻/⁻*, n = 7 (**j**), n = 9 (**a-i**, **m-n**); old WT, n = 16; old *Il11⁻/⁻*, n = 16 (**j**), n = 18 (**a-i**, **m-n**). For gel source data, see Supplementary Fig. 1. AU: arbitrary units; BW: body weight; FC: fold change.

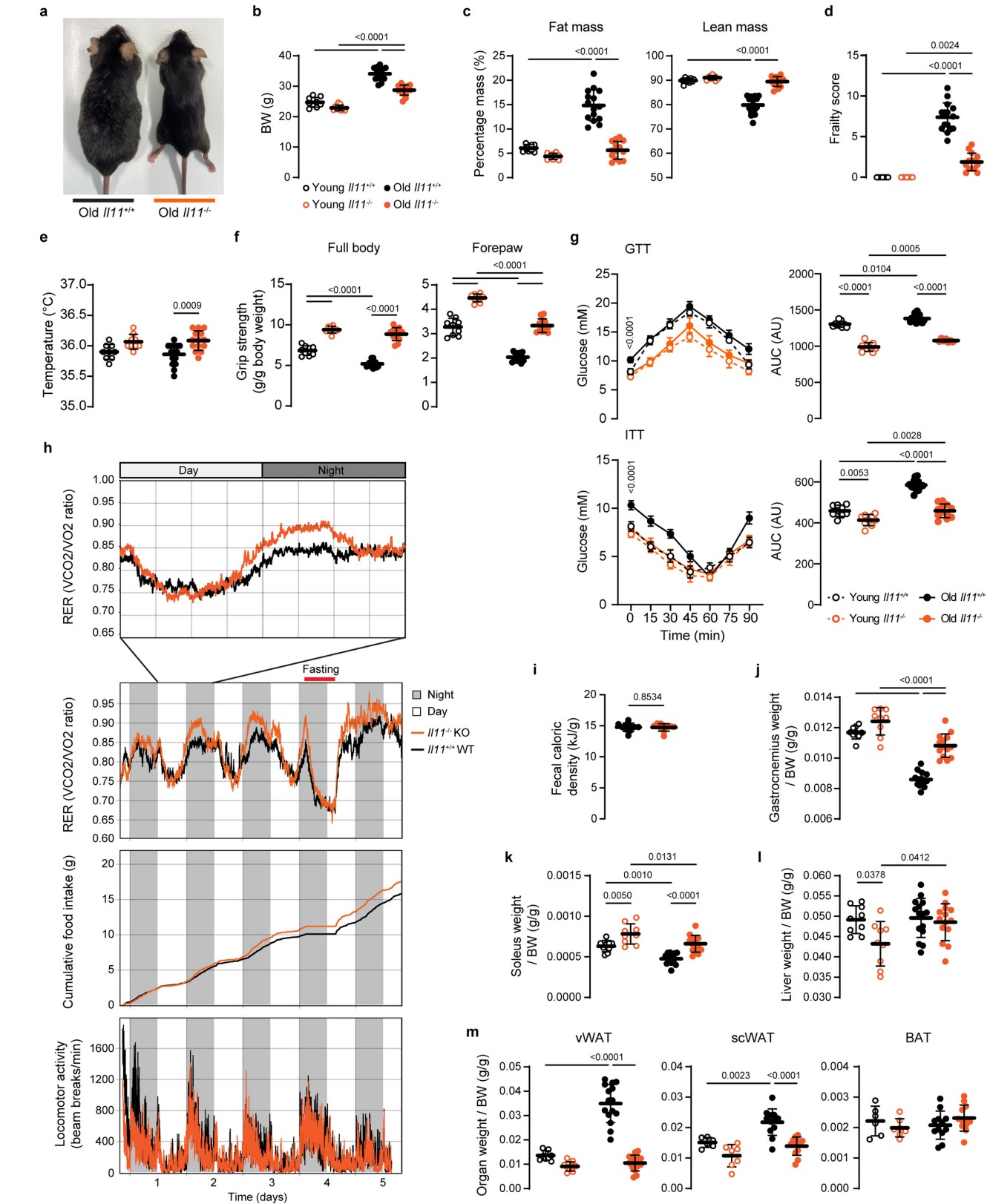

**Extended Data Fig. 6** | See next page for caption.

**Extended Data Fig. 6 | Old male *Il11*$^{-/-}$ mice are protected from age-associated metabolic decline. a** Representative image of 108-week-old WT and *Il11*$^{-/-}$ male mice. **b** Body weights, **c** percentages of fat and lean mass (normalised to BW), **d** frailty scores, **e** body temperatures, **f** full body and forepaw grip strength measurements, **g** glucose and insulin tolerance tests (GTT and ITT) from young (12-week-old) and old (105-week-old) male WT and *Il11*$^{-/-}$ mice. **h** Respiratory exchange ratio (RER) measurement at day 2 (top; 24 h) and assessment of RER (second panel), cumulative food intake, and locomotive activities using the phenomaster system over a 5-day period in 68–70-week-old male WT and *Il11*$^{-/-}$ mice (n = 10/group). **i** Faecal caloric density as measured by bomb calorimetry in 95–105-week-old male WT and *Il11*$^{-/-}$ mice (n = 10/group). Indexed weight of **j** gastrocnemius **k**, soleus, **l** liver, **m** vWAT, subcutaneous WAT (scWAT), and BAT. **b**-**g**, **i**-**m** Data are shown as mean ± SD. **b**-**g**, **j**-**m** Two-way ANOVA with Sidak's correction (young WT and *Il11*$^{-/-}$, n = 6 (**m** (scWAT and BAT)), n = 9 (**b**-**g**, **j**-**l**, **m** (vWAT)); old WT, n = 12 (**m** (scWAT and BAT)), n = 15 (**b**-**g**, **j**-**l**, **m** (vWAT)); old *Il11*$^{-/-}$, n = 12 (**f**-**g**, **m** (scWAT and BAT)), n = 14 (**b**-**e**,**j**-**l**, **m** (vWAT)); **i** two-tailed Mann Whitney test. AU: arbitrary units; BW: body weight; FC: fold change.

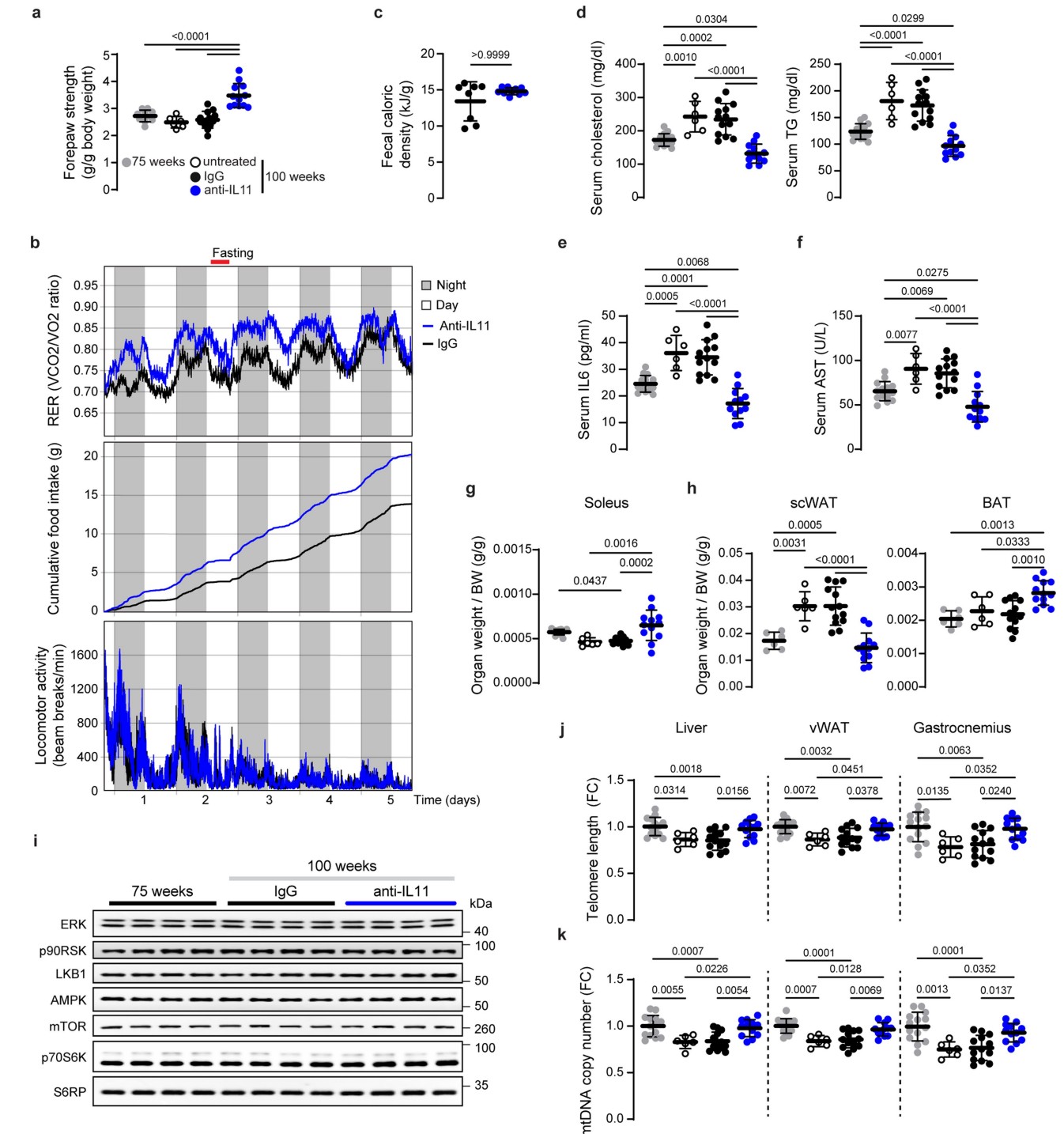

**Extended Data Fig. 7 | Anti-IL11 therapy improves muscle strength and metabolic health in old male mice. a-k** Data for anti-IL11 therapeutic dosing experiment as shown in Schematic Fig. 3a in which IgG or X203 were administered to male mice starting from the age of 75 weeks. **a** Forepaw grip strength, **b** RER measurements, cumulative food intake, and locomotive activities as measured by phenomaster for 5 days on IgG/X203-treated old (81-week-old) male mice – 6 weeks after IgG/X203 administration was started (n = 10/group). **c** Faecal caloric density as measured by bomb calorimetry in IgG and X203-treated 115-week-old male mice (IgG, n = 8; X203, n = 10). Serum levels of **d** cholesterol, TG, **e** IL6, and

**f** AST, indexed weight of **g** soleus, **h** scWAT and BAT, and **i** WB of total proteins for the respective phospho-proteins in vWAT shown in Fig. 3m (n = 6/group). **j** telomere length and **k** mtDNA copy number. **a**, **c-h**, **j-k** Data are shown as mean ± SD. **a**, **d-h**, **j-k** One-way ANOVA with Tukey's correction (75-week-old control, n = 6 (**h**), n = 10 (**a**), n = 14 (**d-g**, **j-k**); untreated 100-week-old, n = 6; IgG 100-week-old, n = 13; X203 100-week-old, n = 12); **c** two-tailed Mann Whitney test; **j** two-tailed Student's t-test. For gel source data, see Supplementary Fig. 1. BW: body weight; FC: fold change.

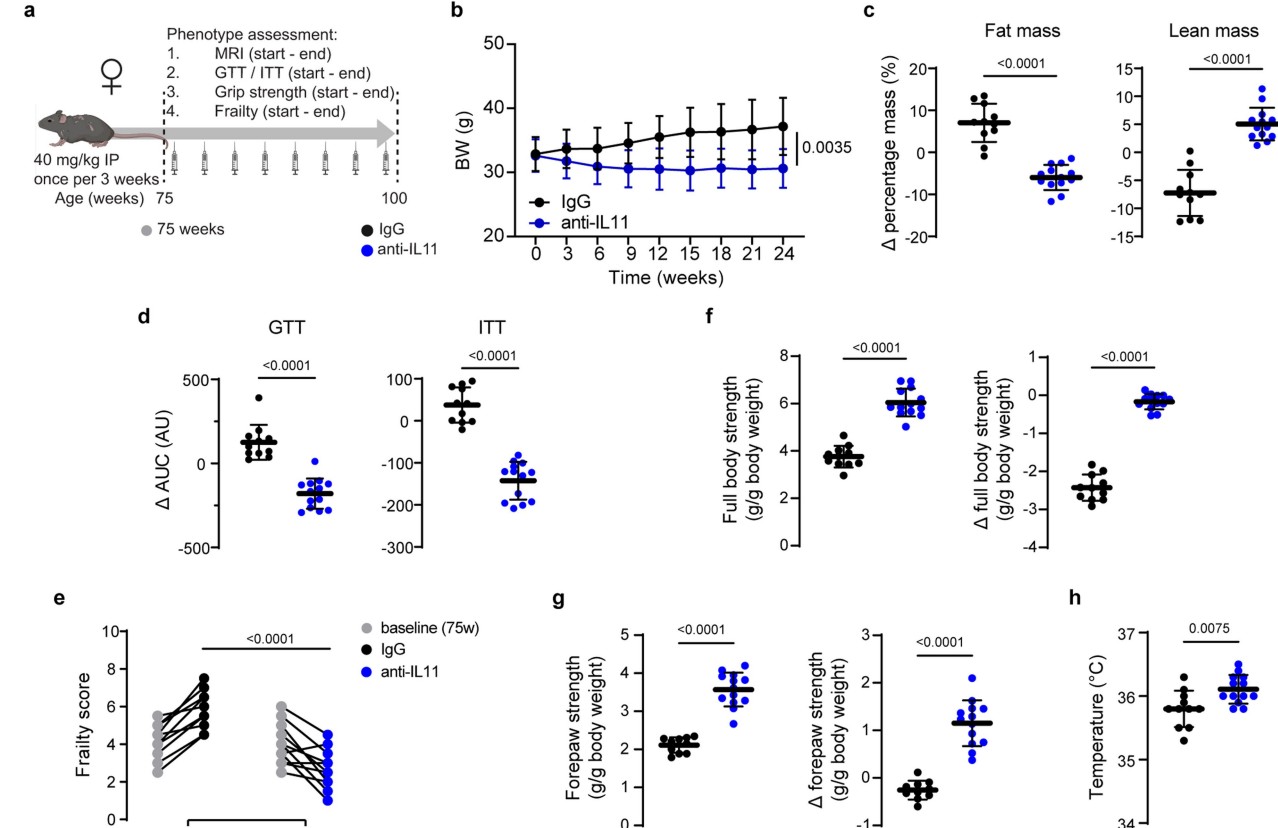

**Extended Data Fig. 8 | Therapeutic inhibition of IL11 reduces age-associated metabolic dysfunction, frailty and sarcopenia in female mice. a** Schematic of anti-IL11 (X203) therapeutic dosing experiment in old female mice for experiments shown in (**b-h**; IgG, n = 10 (**g**), n = 11 (**b-f, h**); X203, n = 13). Mice were given either X203 or an IgG control antibody (40 mg/kg, every 3 weeks) starting from 75 weeks of age for a duration of 25 weeks. Created with BioRender.com. **b** Body weights across time. **c-d** Changes (Δ; values at end-point (100-week-old) - values at starting point (75-week-old)) in **c** fat and lean mass percentage, and **d** area under the curve (AUC) of GTT and ITT. **e** Frailty scores at starting (75-week-old) and end-point (100-week-old). **f-g** Full body and front paw grip strength at end-point (100-week-old) and changes in full body and front paw grip strength over 25 weeks of treatment (values at end-point (100-week-old) - values at starting point (75-week-old)). **h** Body temperatures. **b-d**, **f-h** Data are shown as mean ± SD. **b** Two-way ANOVA, **c-h** two-tailed Student's t-test except for **d** Δ AUC GTT, which was analysed by two-tailed Mann Whitney test. AU: arbitrary units; BW: body weight.

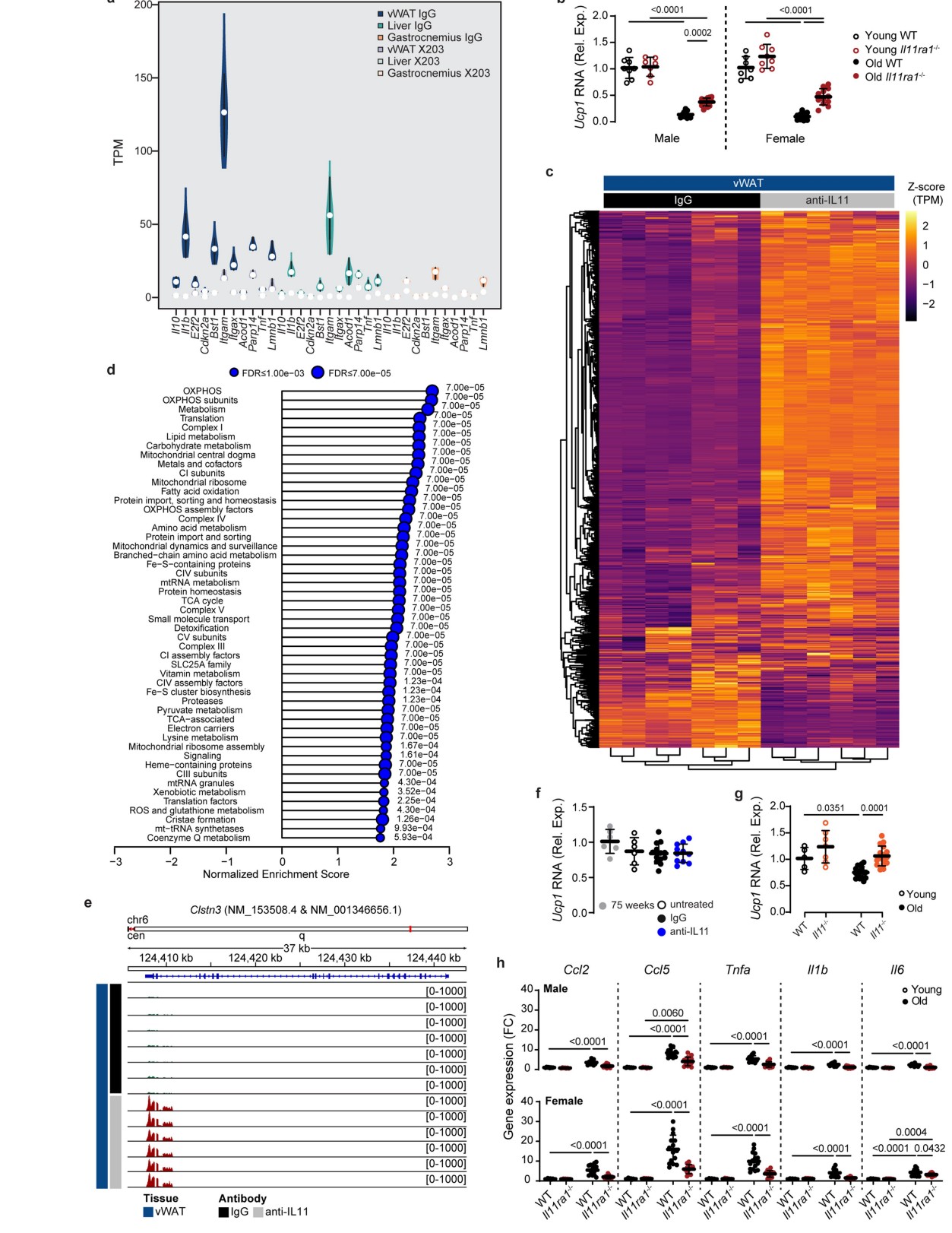

**Extended Data Fig. 9** | See next page for caption.

**Extended Data Fig. 9 | Beneficial effects of anti-IL11 in aged white adipose tissue. a**. Violin plot of Transcripts per million (TPM) values of senescence genes (based on Tabula Muris Senis consortium) in vWAT, liver, gastrocnemius samples from mice receiving either IgG or anti-IL11 as shown in schematic Fig. 3a. **b**. Relative *Ucp1* mRNA from 10-week-old and 110-week-old male and female WT and *Il11ra1*$^{-/-}$ mice (vWAT). **c**. Heatmap showing row-wise scaled TPM values for the gene-list in Mitocarta 3.0. (no. of genes = 1,019 with TPM > = 5 in at least one condition). **d**. A lollipop plot for top 50 significant Mitocarta 3.0 pathways (p-adj<0.05) in enrichment analysis using fgsea R package. No negative NES was found to be significant. **e**. Distribution of RNA-seq reads at the *Clstn3* locus from IgG or anti-IL11-treated vWAT. Relative *Ucp1* mRNA expression levels in BAT from **f** therapeutic dosing group (75-week-old, n = 6; untreated 100-week-old, n = 6; IgG-treated 100-week-old n = 13; X203-treated 100-week-old, n = 12) and **g** female WT and *Il11*$^{-/-}$ mice (young WT, n = 5; young *Il11*$^{-/-}$, n = 7; old WT and *Il11*$^{-/-}$, n = 16/group). **h** Relative vWAT mRNA expression of pro-inflammatory markers (*Ccl2*, *Ccl5*, *Tnfα*, *Il1β*, *Il6*) in young (10-week-old) and old (110-week-old) male and female WT and *Il11ra1*$^{-/-}$ mice. **a**, **c**, **d**, **e** Liver and gastrocnemius (n = 8/group), vWAT IgG, n = 7; vWAT anti-IL11, n = 6; **b**, **h** young male WT, n = 8; young male *Il11ra1*$^{-/-}$, n = 7; old male WT, n = 11; old male *Il11ra1*$^{-/-}$, n = 14; young female WT, n = 7; young female *Il11ra1*$^{-/-}$, n = 8; old female WT, n = 15; old female *Il11ra1*$^{-/-}$, n = 12. **a** Data are shown as violin plots with median ± min-max; **b**, **f-h** data are shown as mean ± SD. **b**, **g**, **h** Two-way ANOVA with Sidak's correction; **e** one-way ANOVA with Tukey's correction.

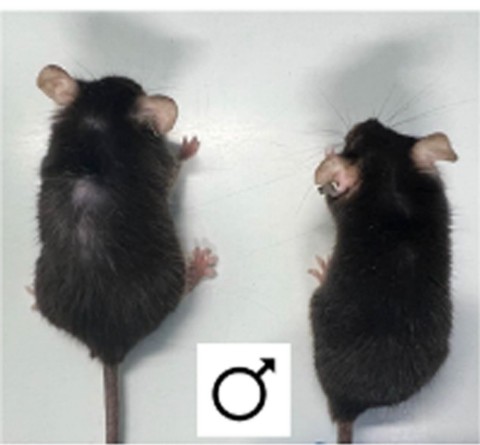

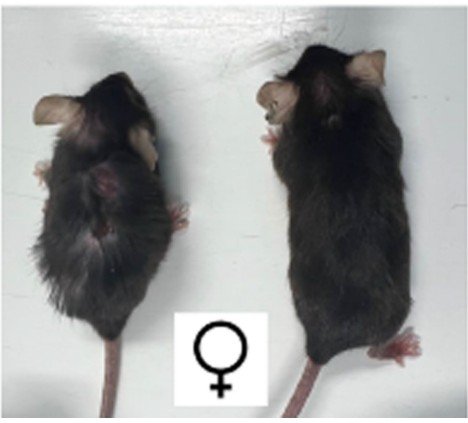

**Extended Data Fig. 10 | Gross appearance of mice receiving anti-IL11 versus IgG in lifespan studies.** Representative images of 130-week-old male (top) and female (bottom) mice from the lifespan therapeutic dosing study where mice received either IgG (mice on the left) or anti-IL11 (X203; mice on the right) from 75 weeks of age until death.

# Reporting Summary

## Statistics

For all statistical analyses, confirm that the following items are present in the figure legend, table legend, main text, or Methods section.

| n/a | Confirmed | |
|---|---|---|
| ☐ | ☒ | The exact sample size (*n*) for each experimental group/condition, given as a discrete number and unit of measurement |
| ☐ | ☒ | A statement on whether measurements were taken from distinct samples or whether the same sample was measured repeatedly |
| ☐ | ☒ | The statistical test(s) used AND whether they are one- or two-sided *Only common tests should be described solely by name; describe more complex techniques in the Methods section.* |
| ☒ | ☐ | A description of all covariates tested |
| ☐ | ☒ | A description of any assumptions or corrections, such as tests of normality and adjustment for multiple comparisons |
| ☐ | ☒ | A full description of the statistical parameters including central tendency (e.g. means) or other basic estimates (e.g. regression coefficient) AND variation (e.g. standard deviation) or associated estimates of uncertainty (e.g. confidence intervals) |
| ☐ | ☒ | For null hypothesis testing, the test statistic (e.g. *F*, *t*, *r*) with confidence intervals, effect sizes, degrees of freedom and *P* value noted *Give P values as exact values whenever suitable.* |
| ☒ | ☐ | For Bayesian analysis, information on the choice of priors and Markov chain Monte Carlo settings |
| ☒ | ☐ | For hierarchical and complex designs, identification of the appropriate level for tests and full reporting of outcomes |
| ☒ | ☐ | Estimates of effect sizes (e.g. Cohen's *d*, Pearson's *r*), indicating how they were calculated |

*Our web collection on statistics for biologists contains articles on many of the points above.*

## Software and code

Policy information about availability of computer code

| Data collection | The study does not involve software and code for data collection. |
|---|---|
| Data analysis | GraphPad Prism (version 10), ImageJ (version 1.53f51), bcl2fastq (v2.20.0.422), Trimmomatic (V0.36), STAR (V2.7.9), DESeq2 (V1.36.0), fgsea R package (v.1.22.0) |

For manuscripts utilizing custom algorithms or software that are central to the research but not yet described in published literature, software must be made available to editors and reviewers. We strongly encourage code deposition in a community repository (e.g. GitHub). See the Nature Portfolio guidelines for submitting code & software for further information.

## Data

Policy information about availability of data

All manuscripts must include a data availability statement. This statement should provide the following information, where applicable:
- Accession codes, unique identifiers, or web links for publicly available datasets
- A description of any restrictions on data availability
- For clinical datasets or third party data, please ensure that the statement adheres to our policy

All data are available within the Article or Supplementary Information. The RNAseq data reported in this paper are available on the Short Read Archive with Bioproject ID: PRJNA939262 (https://www.ncbi.nlm.nih.gov/bioproject/PRJNA939262). Datasets used for analysis in this study are as follows: Ensembl release 104 Mus musculus GRCm39 gene annotations (GRCm39, https://asia.ensembl.org/info/data/ftp/index.html), MSigDB Hallmark (v.7.5.1, https://www.gsea-msigdb.org/

gsea/msigdb/human/collections.jsp), MitoCarta (v3.0, https://www.broadinstitute.org/mitocarta/mitocarta30-inventory-mammalian-mitochondrial-proteins-and-pathways). Source data are provided with this paper

## Human research participants

Policy information about <u>studies involving human research participants and Sex and Gender in Research.</u>

| | |
|---|---|
| Reporting on sex and gender | The study did not involve human research participants |
| Population characteristics | N/A |
| Recruitment | N/A |
| Ethics oversight | N/A |

Note that full information on the approval of the study protocol must also be provided in the manuscript.

# Field-specific reporting

Please select the one below that is the best fit for your research. If you are not sure, read the appropriate sections before making your selection.

☒ Life sciences      ☐ Behavioural & social sciences      ☐ Ecological, evolutionary & environmental sciences

For a reference copy of the document with all sections, see nature.com/documents/nr-reporting-summary-flat.pdf

# Life sciences study design

All studies must disclose on these points even when the disclosure is negative.

| | |
|---|---|
| Sample size | Sample sizes were determined based on the authors' experience with the preliminary in vitro, in vivo, and molecular studies and by referencing a healthspan/lifespan study in mice by Ackert-Bicknell et al (PMID: 26069080) to detect a 20% change in phenotype between treatment groups or genotypes with 80% power (α=0.05).<br>Sample sizes for experiments involving Il11ra1-/- and Il11-/- mice (and their respective WT mice) varied depending on animal availability. Sample size for cell based assays were determined based on sample availability. |
| Data exclusions | Outlier tests were conducted using the ROUT method in GraphPad Prism, with the maximum false discovery rate (FDR) set to 1%. |
| Replication | All experiments were repeated with reproducibility. The replication number for each experiment is indicated in the legend of the corresponding figure. |
| Randomization | For in vitro studies, equal number of cells were seeded and allocated randomly into experimental groups. For in vivo studies, mice were randomly allocated to experimental groups on the day of the treatment except for Il11ra1-/- and Il11-/- in which randomization was not applicable. |
| Blinding | For in vitro experiments, investigators were not blinded to group allocation during data collection and analysis as they were performed by a single individual. For in vivo experiments, treatments/genotypes were not disclosed to investigators generating quantitative readouts during data collection but were revealed during the analysis. Histological analysis were performed blinded to treatments and genotypes. |

# Reporting for specific materials, systems and methods

We require information from authors about some types of materials, experimental systems and methods used in many studies. Here, indicate whether each material, system or method listed is relevant to your study. If you are not sure if a list item applies to your research, read the appropriate section before selecting a response.

## Materials & experimental systems

| n/a | Involved in the study |
|---|---|
| ☐ | ☒ Antibodies |
| ☐ | ☒ Eukaryotic cell lines |
| ☒ | ☐ Palaeontology and archaeology |
| ☐ | ☒ Animals and other organisms |
| ☒ | ☐ Clinical data |
| ☒ | ☐ Dual use research of concern |

## Methods

| n/a | Involved in the study |
|---|---|
| ☒ | ☐ ChIP-seq |
| ☒ | ☐ Flow cytometry |
| ☒ | ☐ MRI-based neuroimaging |

# Antibodies

| | |
|---|---|
| **Antibodies used** | For all the commercially available antibodies used in this study, the recommended dilutions from the manufacturer's instructions have been followed

The following commercially available primary antibodies were used for Western Blot. They are listed as antigen followed by catalog number, clone name (for monoclonal antibody), supplier name, and dilution ratio:
1. p-AMPK, 2535, clone 40H9, CST, 1:1000.
2. AMPK, 5832, clone D63G4, CST, 1:1000.
3. Cyclin D1, 55506, clone E3P5S, CST, 1:1000.
4. p-ERK1/2, 4370, clone D13.14.4E, CST, 1:1000.
5. ERK1/2, 4695, clone 137F5, CST, 1:1000.
6. GAPDH 2118, clone 14C10, CST, 1:1000.
7. IL11, NA, clone X203, Aldevron, 1:1000.
8. p-LKB1, 3482, clone C67A3, CST, 1:1000.
9. LKB1, 3047, clone D60C5, CST, 1:1000.
10. p-mTOR, 2971, NA, CST, 1:1000.
11. mTOR, 2972, NA, CST, 1:1000.
12. p-NFκB, 3033, clone 93H1, CST, 1:1000.
13. NFκB, 8242, clone D14E12, CST, 1:1000.
14. Human p16, ab108349, clone EPR1473, abcam, 1:1000.
15. Mouse p16, ab232402, clone EPR20418, abcam, 1:1000.
16. Human p21, ab109520, clone EPR362 abcam, 1:1000.
17. Mouse p21, ab188224, clone EPR18021, abcam, 1:1000.
18. p-p70S6K, 9234, clone 108D2, CST, 1:1000.
19. p70S6K, 2708, clone 49D7, CST, 1:1000.
20. p-p90RSK, 11989, clone D3H11, CST, 1:1000.
21. p90RSK, 9355, clone 32D7, CST, 1:1000.
22. PCNA, 13110, clone D3H8P, CST, 1:1000.
23. PGC1a, ab191838, NA, abcam, 1:1000.
24. p-S6RP, 4858, clone D57.2.2E, CST, 1:1000.
25. S6RP, 2217, clone 5G10, CST, 1:1000.
26. pSTAT3, 4113, clone M9C6, CST, 1:1000.
27. STAT3, 4904, clone 79D7, CST, 1:1000.
28. UCP1, 72298, clone E9Z2V, CST, 1:1000.
29. HRP conjugated anti-mouse IgG (H+L)/anti-mouse HRP, 7076, NA, CST, 1:1000.
30. HRP conjugated anti-rabbit IgG (H+L)/anti-rabbit HRP, 7074, NA, CST, 1:1000.

The following custom-made primary antibodies were used for neutralization study (in vitro/in vivo treatment). They are listed as antigen followed by catalog number, clone name (for monoclonal antibody), supplier name, and dilution ratio:
1. IgG, NA, clone 11E10, Aldevron, in vitro: 2 µg/ml, in vivo: 40 mg/kg
2. IL11, NA, clone X203, Aldevron, in vitro: 2 µg/ml, in vivo: 40 mg/kg
3. IL11RA, NA, clone X209, Aldevron, in vitro: 2 µg/ml.

The following commercially available primary antibodies were used for Operetta phenotyping assay. They are listed as antigen followed by catalog number, clone name (for monoclonal antibody), supplier name, and dilution ratio:
1. Human p16, ab108349, clone EPR1473, abcam, 1:1000.
2. Human p21, ab109520, clone EPR362 abcam, 1:1000.
3. Anti-rabbit Alexa Fluor 488, ab150077, NA, abcam, 1:1000.

The following commercially available primary antibodies were used for immunofluoresence. They are listed as antigen followed by catalog number, clone name (for monoclonal antibody), supplier name, and dilution ratio:
1. Adiponectin, 21613-1-AP, NA, Proteintech, 1:100
2. CD31, ab222783, clone EPR17260-263, abcam, 1:200
3. FHL1, 10991-1-AP, NA, Proteintech, 1:200
4. GFP, ab290, NA, abcam, 1:100
5. GFP, ab6673, NA, abcam, 1:100
6. PDGFRa, AF1062, NA, R&D systems, 1:100
7. SLC10A1, MBS177905, NA, MyBioSource, 1:100
8. SM22a, ab14106, NA, abcam, 1:200
9. Anti-rabbit Alexa Fluor 488, ab150077, NA, abcam, 1:300
10. Anti-goat Alexa Fluor 488, ab150129, NA, abcam, 1:300
11. Anti-rabbit Alexa Fluor 555, ab150074, NA, abcam, 1:300

The following commercially available primary antibody was used for immunohistochemistry. It is listed as antigen followed by catalog number, clone name (for monoclonal antibody), supplier name, and dilution ratio:
1. CD68, ab125212, NA, abcam, 1:100. |
| **Validation** | All commercially available antibodies have been validated by the manufacturers for the applications used in this study as indicated on the respective manufacturer's website. The custom-made antibodies i.e: IgG (11E10), anti-IL11 (X203), and anti-IL11RA (X209) were validated by citations. Manufacturer's website containing validation data (also listed here) for the commercially available antibodies and citations for the custom-made antibodies are listed below: |

1. Adiponectin (https://www.ptglab.com/products/ADIPOQ-Antibody-21613-1-AP.htm); species reactivity: mouse; validated application: IF (in NIH/3T3 cells).
2. p-AMPK (https://www.cellsignal.com/products/primary-antibodies/phospho-ampka-thr172-40h9-rabbit-mab/2535); species reactivity: human, mouse; validated application: WB (using 293T (human) and C2C12 (mouse) cell lysates).
3. AMPK (https://www.cellsignal.com/products/primary-antibodies/ampka-d63g4-rabbit-mab/5832); species reactivity: human, mouse; validated application: WB (using HeLa, K-562 (human) and Neuro-2a (mouse) cell lysates).
4. CD31 (https://www.abcam.com/products/primary-antibodies/cd31-antibody-epr17260-263-ab222783.html); species reactivity: mouse; validated application: IF (on bEND.3 and NIH/3T3 cells).
5. CD68 (https://www.abcam.com/products/primary-antibodies/cd68-antibody-ab125212.html); species reactivity: mouse; validated application: IHC (on paraformaldehyde-fixed, paraffin-embedded mouse brain, skin, spleen, and liver).
6. Cyclin D1 (https://www.cellsignal.com/products/primary-antibodies/cyclin-d1-e3p5s-xp-rabbit-mab/55506); species reactivity: human, mouse; validated application: WB (using SH-SY5Y cell (human) and mouse liver and kidney lysates).
7. p-ERK1/2 (https://www.cellsignal.com/products/primary-antibodies/phospho-p44-42-mapk-erk1-2-thr202-tyr204-d13-14-4e-xp-rabbit-mab/4370); species reactivity: human, mouse; validated application: WB (using 293 (human) and NIH/3T3 (mouse) cell lysates).
8. ERK1/2 (https://www.cellsignal.com/products/primary-antibodies/p44-42-mapk-erk1-2-137f5-rabbit-mab/4695); species reactivity: human, mouse; validated application: WB (using HeLa (human) and NIH/3T3 (mouse) cell lysates).
9. FHL1 (https://www.ptglab.com/products/FHL1-Antibody-10991-1-AP.htm); species reactivity: mouse; this antibody is not validated for IF but it's listed as the suitable application - validated application: WB (using mouse skeletal muscle lysates).
10. GAPDH (https://www.cellsignal.com/products/primary-antibodies/gapdh-14c10-rabbit-mab/2118); species reactivity: human, mouse; validated application: WB (using HeLa (human) and NIH/3T3 (mouse) cell lysates).
11. GFP (ab290) (https://www.abcam.com/products/primary-antibodies/gfp-antibody-ab290.html); species reactivity: mouse; validated application: IF (using GFP-transfected NIH/3T3 cells).
12. GFP (ab6673) (https://www.abcam.com/products/primary-antibodies/gfp-antibody-ab6673.html); species reactivity: mouse; validated application: IF (in Hex-GFP transgenic mouse embryo se).
13. IgG (11E10) was validated for its non-ability to have any effect on human/mouse cells in PMID: 34108253.
14. IL11 (X203) was validated for neutralization of human and mouse IL11 in PMID: 31554736 and for WB (mouse liver lysates) in PMID: 31078624; species reactivity: human and mouse.
15. IL11RA (X209) was validated for neutralization of human IL11RA in PMID: 31078624, 34108253, 36470928; species reactivity: human.
16. p-LKB1 (https://www.cellsignal.com/products/primary-antibodies/phospho-lkb1-ser428-c67a3-rabbit-mab/3482); species reactivity: human, mouse; validated application: WB (using Caki (human) and L929 (mouse) cell lysates).
17. LKB1 (https://www.cellsignal.com/products/primary-antibodies/lkb1-d60c5-rabbit-mab/3047); species reactivity: human, mouse; validated application: WB (using Caki (human) and L929 (mouse) cell lysates).
18. p-mTOR (https://www.cellsignal.com/products/primary-antibodies/phospho-mtor-ser2448-antibody/2971); species reactivity: human, mouse; validated application: WB (using 293, MCF-7 cell (human) and mouse liver, heart, skeletal muscle lysates).
19. mTOR (https://www.cellsignal.com/products/primary-antibodies/mtor-antibody/2972); species reactivity: human, mouse; validated application: WB (using 293, HeLa (human) and C2C12 (mouse) cell lysates).
20. p-NFκB (https://www.cellsignal.com/products/primary-antibodies/phospho-nf-kb-p65-ser536-93h1-rabbit-mab/3033); species reactivity: human; validated application: WB (using HeLa cell lysates).
21. NFκB (https://www.cellsignal.com/products/primary-antibodies/nf-kb-p65-d14e12-xp-rabbit-mab/8242); species reactivity: human; validated application: WB (using HeLa, MCF-7 cell lysates).
22. p16 (https://www.abcam.com/products/primary-antibodies/cdkn2ap16ink4a-antibody-epr1473-c-terminal-ab108349.html); species reactivity: human; validated application: WB (using HeLa cell lysates), this antibody is not validated for IF but it's listed as one of the possible applications.
23. p16 (https://www.abcam.com/products/primary-antibodies/cdkn2ap16ink4a-antibody-epr20418-bsa-and-azide-free-ab232402.html); species reactivity: mouse; validated application: WB (using MEF cell lysates).
24. p21 (https://www.abcam.com/products/primary-antibodies/p21-antibody-epr362-ab109520.html); species reactivity: human; validated application: WB (using HeLa cell lysates), IF (in MCF-7 cells).
25. p21 (https://www.abcam.com/products/primary-antibodies/p21-antibody-epr18021-ab188224.html); species reactivity: mouse; validated application: WB (using NIH/3TC cell lysates).
26. p-p70S6K (https://www.cellsignal.com/products/primary-antibodies/phospho-p70-s6-kinase-thr389-108d2-rabbit-mab/9234); species reactivity: human, mouse; validated application: WB (using 293T (human) and NIH/3T3 (mouse) cell lysates).
27. p70S6K (https://www.cellsignal.com/products/primary-antibodies/p70-s6-kinase-49d7-rabbit-mab/2708); species reactivity: human, mouse; validated application: WB (using HeLa, MCF-7 cell (human) and mouse kidney, liver, skeletal muscle, heart, adipocyte lysates).
28. p-p90RSK (https://www.cellsignal.com/products/primary-antibodies/phospho-p90rsk-ser380-d3h11-rabbit-mab/11989); species reactivity: human, mouse; validated application: WB (using HeLa (human) and NIH/3T3 (mouse) cell lysates).
29. p90RSK (https://www.cellsignal.com/products/primary-antibodies/rsk1-rsk2-rsk3-32d7-rabbit-mab/9355); species reactivity: human, mouse; validated application: WB (using HeLa (human) and L929 (mouse) cell lysates).
30. PCNA (https://www.cellsignal.com/products/primary-antibodies/pcna-d3h8p-xp-rabbit-mab/13110); species reactivity: human; validated application: WB (using HeLa, MCF-7 cell lysates).
31. PDGFRa (https://www.rndsystems.com/products/mouse-pdgf-ralpha-antibody_af1062); species reactivity: mouse; validated application: IF (on mouse intestinal villus tip telocytes).
32. PGC1a (https://www.abcam.com/products/primary-antibodies/pgc1-alpha-antibody-n-terminal-ab191838.html); species reactivity: mouse; validated application: WB (using mouse skin lysates).
33. p-S6RP (https://www.cellsignal.com/products/primary-antibodies/phospho-s6-ribosomal-protein-ser235-236-d57-2-2e-xp-rabbit-mab/4858); species reactivity: human, mouse; validated application: WB (using MCF-7 cell (human) and mouse liver, skeletal muscle, heart lysates).
34. S6RP (https://www.cellsignal.com/products/primary-antibodies/s6-ribosomal-protein-5g10-rabbit-mab/2217); species reactivity: human, mouse; validated application: WB (using HeLa (human) and NIH/3T3 (mouse) cell lysates).
35. SLC10A1 (https://www.mybiosource.com/polyclonal-mouse-rat-antibody/slc10a1/177905); species reactivity: mouse; validated application: IF (on paraffin-embedded section of mouse liver tissues).
36. SM22a (https://www.abcam.com/products/primary-antibodies/tagIntransgelin-antibody-ab14106.html); species reactivity: mouse; validated application: IF (on mouse muscle cells).
37. p-STAT3 (https://www.cellsignal.com/products/primary-antibodies/phospho-stat3-tyr705-m9c6-mouse-mab/4113); species

reactivity: human; validated application: WB (using HeLa cell lysates).
38. STAT3 (https://www.cellsignal.com/products/primary-antibodies/stat3-79d7-rabbit-mab/4904); species reactivity: human; validated application: WB (using HeLa cell lysates).
39. UCP1 (https://www.cellsignal.com/products/primary-antibodies/ucp1-e9z2v-xp-rabbit-mab/72298); species reactivity: mouse; validated application: WB (using mouse brown adipose tissues)
40. Anti-rabbit HRP (https://www.cellsignal.com/products/secondary-antibodies/anti-rabbit-igg-hrp-linked-antibody/7074); designed for use with rabbit polyclonal and monoclonal antibodies, this affinity purified goat anti-rabbit IgG (heavy and light chain) antibody is conjugated to horseradish peroxidase (HRP) for chemiluminescent detection; validated application: WB (This product is thoroughly validated with CST primary antibodies).
41. Anti-mouse HRP (https://www.cellsignal.com/products/secondary-antibodies/anti-mouse-igg-hrp-linked-antibody/7076); affinity purified horse anti-mouse IgG (heavy and light chain) antibody is conjugated to HRP for chemiluminescent detection; validated application: WB (This product is thoroughly validated with CST primary antibodies).
42. Goat anti-rabbit Alexa Fluor 488 (https://www.abcam.com/goat-rabbit-igg-hl-alexa-fluor-488-ab150077.html); validated application: IF.
43. Donkey anti-goat Alexa Fluor 488 (https://www.abcam.com/donkey-goat-igg-hl-alexa-fluor-488-ab150129.html); validated application: IF.
44. Donkey anti-goat Alexa Fluor 555(https://www.abcam.com/products/secondary-antibodies/donkey-rabbit-igg-hl-alexa-fluor-555-ab150074.html); validated application: IF.

# Eukaryotic cell lines

Policy information about cell lines and Sex and Gender in Research

| | |
|---|---|
| Cell line source(s) | Primary human cardiac fibroblasts (HCFs, 52-year-old male, 6330, lot #9580, ScienCell).<br>Primary human hepatocytes were isolated from a 22-week-old foetus (5200, lot #34967, ScienCell). |
| Authentication | Primary human cardiac fibroblasts (HCFs) and primary human hepatocytes were authenticated by ScienCell based on their morphology and by using immunofluorescence (IF) for cell-specific markers, as detailed in the respective product datasheet and certificate of analysis (COA).<br><br>Primary HCFs were authenticated by their fibroblast morphology and phenotype, characterised by IF staining for fibronectin and vimentin.<br><br>Primary human hepatocytes were authenticated by their hepatocyte morphology and phenotype, characterised by positive IF for cytokeratin-18 and WB for albumin. |
| Mycoplasma contamination | All cell lines were tested to be free of mycoplasma contamination. |
| Commonly misidentified lines<br>(See ICLAC register) | No commonly misidentified cell lines were used in the study. |

# Animals and other research organisms

Policy information about studies involving animals; ARRIVE guidelines recommended for reporting animal research, and Sex and Gender in Research

| | |
|---|---|
| Laboratory animals | All mice were housed at 21-24? with 40-70% humidity on a 12-hour light/dark cycle and provided food and water ad libitum. Our mouse colonies hold specific pathogen free (SPF) status and undergo quarterly and annual tests for common pathogens. The room housing our animals is Murine Norovirus and Helicobacter positive, and these particular pathogens are deemed acceptable within our SPF facility. The mouse strains used in our study are as follow:<br>1. Il11ra1-deleted mice (Il11ra1−/− or Il11ra1 KO)<br>Male and female Il11ra1+/+ (wild-type) and Il11ra1-/- mice25 (B6.129S1-Il11ratm1Wehi/J, The Jackson Laboratory) were sacrificed at 110 weeks of age for blood and tissue collection; 10-12 weeks old male and female mice of the respective genotypes were used as controls.<br>2. Il11-deleted mice (Il11−/−)<br>Male and female mice lacking functional alleles for Il11 (Il11−/−), which were generated and characterised previously31,51, and their wild-type counterparts were sacrificed at 10-12 weeks of age (young controls) and 104-108 weeks of age (old mice).<br>3. Il11-EGFP reporter mice<br>Young (10-week-old) and old (100-week-old) transgenic mice (C57BL/6J background) with EGFP knocked-into the Il11 gene (Il11-EGFP mice, Cyagen Biosciences Inc)29 were sacrificed for IF staining studies of liver, gastrocnemius, and visceral gonadal white adipose tissue (referred in the text as vWAT). Old wild-type littermates were used as aged negative controls.<br>4. In vivo administration of anti-IL11<br>Male and female C57BL/6J mice (Jackson Laboratory) were randomised prior to receiving either no treatment, anti-IL11 (X203) or IgG (11E10). X203 or 11E10 (40 mg/kg, every 3 weeks) were administered by intraperitoneal injection, starting from 75 weeks of age for a duration of 25 weeks; mice were then sacrificed at 100 weeks of age. |
| Wild animals | The study does not involve wild animals. |

| | |
|---|---|
| Reporting on sex | This study use both male and female mice. |
| Field-collected samples | The study does not involve field collected animals. |
| Ethics oversight | Animal studies were carried out in compliance with the recommendations in the Guidelines on the Care and Use of Animals for Scientific Purposes of the National Advisory Committee for Laboratory Animal Research (NACLAR). All experimental procedures were approved (SHS/2019/1481 and SHS/2019/1483) and conducted in accordance with the SingHealth Institutional Animal Care and Use Committee (IACUC). Certified veterinarians were responsible for all animal experiment procedures according to the laws governing animal research in Singapore. |

Note that full information on the approval of the study protocol must also be provided in the manuscript.

