## [Peer Review File · Nature]

Manuscript Title: Inhibition of IL11 signalling extends mammalian health span and lifespan.

Editorial Notes:

Redactions – unpublished data

Reviewer Comments & Author Rebuttals

Reviewer Reports on the Initial Version:

Referees' comments:

Referee #1 (Remarks to the Author):

Review of Widjaja...Cook

May 2023

Inhibition of an immunometabolic axis of mTORC1 activation extends mammalian healthspan

General summary:

1. This paper makes a number of highly interesting observations, which may deserve publication in a widely-read journal. The analyses of the effects of mutations that blunt IL11 action, and of anti-IL11 antibody, on metabolic indices and protein kinase substrates make quite a strong case that IL11 could play an under-appreciated role in aging and merits future study. The material on visceral adipose tissue (Fig 4) is particularly strong.

2. There are portions of the paper that seem to me to be of much lower interest.

2a. The authors appear to believe that age-related appearance of senescent cells is a critical element in aging, but in my view the evidence for this idea is quite weak. Data showing effects, in mice, of IL11 on p16, p21, and multiple inflammatory effectors (which the authors, in keeping with too many others, lump together as “senescence associate secretory proteins”) are an important element of the paper and should be retained, although the interpretation of all these changes as reflecting creation of one single cell type (the “senescent” cells) is unpersuasive. But portions of the paper in which IL11 is added to cultured cells, in contrast, do not add materially to our understanding of the biology in the IL11-inhibited mice, and could be deleted.

2b. The material in Figure 5, focused on production of new antibodies to IL11, should be deleted. It might deserve publication in a technical specialist journal, or could perhaps serve as Figure 1 in a separate paper focused on clinical trials, but does not add to the main theme of the current manuscript. I would just remove it.

3. The authors are, in my view, often too quick to assert that some molecular phenomenon is “due to IL11 action.” I have noted a couple of specific instances in the detailed notes below. It seems at least equally likely that many of these effects are secondary, i.e. dependent on one or more upstream changes that are themselves due to IL11 action. For an example, if IL11 causes adipose thermogenesis via UCP1, this could in turn modulate adipokines or cytokine levels in blood, which in turn alter muscle or liver function, and so forth. The data are of interest – but the interpretation needs to be more cautious, with recognition that IL11 dependent effects in unknown cells and tissues may have repercussions in many other parts of the system.

4. The authors very often make a (very common) statistical mistake. For example, when they have compared control to mutant mice at two ages, and see a statistically significant effect at the older age, they infer (incorrectly) that IL11 has affected the aging process. But the correct method is a two-factor ANOVA (age, mutation) with interaction term. If the interaction term is significant, this suggests that the IL11 effect at one age is different from that at the other age.

5. The main gap in the evidence presented is that the authors do not ever show us data to test the hypothesis that mice with lower IL11 signals are aging slowly. They do show data that IL11-dependent events differ in normal vs IL11-blunted conditions – but they do not test age-related functions not already known to be IL11-modulated.

The most important of these is lifespan – I would be more enthusiastic for this already strong paper if they had data showing longer lifespan in one or more of the low-IL11 mouse models. In a revision, they should state clearly whether they have such data, and, if so, what they found – it would be unethical to withhold such data from the reader if it already exists.

Similarly, if they have data on age-dependent functional outcomes, they should say so, and, if possible, show the results. Have they tested cognition? Have they tested collagen cross-linking, or hearing, or response to vaccination, or bone density, etc.? Is cancer delayed? Data showing deceleration of age-related changes in one or many of these tests would strengthen the paper. Data showing that these tests were NOT altered in one or more of the IL11-modulated mice would also be important to include, even if it complicates the reader’s opinion about the potential use of anti-IL11 as an anti-aging therapy.

The data they do provide – on muscle strength, see Fig 2g – sometimes illustrate the complexities of interpretation. IL11-negative mice have higher muscle strength (adjusted for weight) even when young, and retain this strength advantage when older. Thus it would be incorrect to conclude that the IL11-negative mice are slower than controls at age-dependent muscle strength decline.

I have provided below some more detailed comments for the authors to consider should they decide to revise the paper.

Detailed comments:

The authors introduce their study by asserting that "Ageing studies to date have focused mostly on lifespan extension that can be achieved at the expense of healthspan, merely represent extended periods of frailty and, in mice, primarily reflects reduced murine cancers." I do not think they are correct on any of these points. The best-known diets, mutations, and drugs that extend mouse lifespan have been shown to postpone multiple aspects of aging (cataracts, cognitive change, change in activity, kidney and heart pathology, and many others) in addition to their effects on cancer and lifespan. The assertion that these interventions merely create a long period of frailty is not, as far as I know, supported by any evidence. The authors are setting up a straw man here, and it's likely to mislead many readers.

The idea that IL11 is "related to IL6" and must, by implication, be related to "inflammaging" does not make any sense to me.

It is a serious oversimplification to state, as the authors do, that IL11 ".. causes liver steatosis, visceral adiposity and impaired muscle cell function, known features of ageing." Steatosis and visceral adiposity are sometimes present in aged mice and people, and sometimes not, depending on diet, exercise, and genetics among other things. The implication that IL11 "causes...impaired muscle function" in aging is, similarly, quite misleading – there are many age-related changes in muscle cell biology, motor end-plate interactions, muscle fiber type switching, fibrosis, and myocyte volume that are multifactorial. To say that these changes are "caused by" IL11 is inaccurate.

The stated hypothesis is that "IL11 is an inflammaging factor." This is untestably vague. Hypotheses that include causal networks, for example that involve IL11 as a required element in some specific aspect(s) of aging, would be potentially valuable. The authors' hypothesis is much less precise – it implies merely that some aspects of aging (unspecified) might be caused by, or accompanied by, or related to, IL11, and that modulation of IL11 in unspecified circumstances might produce some unspecified changes in aging via inflammation.

The authors state that "with age, there was progressive activation of ERK/p90RSK, inactivation of LKB1/AMPK and mTOR/p70S6K activation, which comprise the IL11 signalling module, in liver and muscle." They do not provide any evidence that these changes in levels of various phosphoproteins actually are caused by age-related changes in IL11, either on its own or in conjunction with other age-dependent changes in cell biology, organ composition, or other stimuli in vivo. They could, plausibly, reflect secondary changes triggered by IL11 dependent modification of neuroendocrine or paracrine factors.

Fig 1b: no statistics are shown. The authors show phosphoproteins only, but not total protein level for an of the corresponding enzyme substrates, so it is not possible to determine if the ratio of phosphoprotein to total protein is age-sensitive or not.

Fig 1c: the colors add no information – they just make the numbers harder to read. The normalization to GAPDH is appropriate only if there is no age effect on GAPDH in any of the tissues – have the authors checked this out?

It is not good English to say that IL11 is “lowly expressed.”

Figure 1d is far too small to be useful.

The authors conclude that “IL11 activates ERK/mTOR and increases senescence markers with age”. But this is a premature conclusion. It is clear that the IL11ra1-negative mice have many abnormalities documented in Fig 1e. But whether this is due to direct effects of IL11 *per se*, or indirect effects (perhaps a change in neuroendocrine or cytokines that modulate liver biology), or a developmental abnormality in some IL11-dependent lineage, or lower body weights, or higher body temperatures, or altered liver cell death (AST, ALA), etc., is unknown.

It is not relevant that “BHB is advocated as an anti-ageing supplement.” If the authors know of data to show that BHB actually slows aging in mice, they should state it.

Fig 1m is hard to interpret without evidence that the “clock” is an effective index of age in normal mice. What age estimate does it give for 20 week old mice, for example? And does the apparent difference in biological age between mutant and normal mice get larger as the mice get older, as the model would predict?

Fig 1n: did the authors look at other indices of “senescence” in IL11-treated fibroblasts, or just p16 and p21? Did the authors look at other “SASP” cytokines, or just at IL6 and IL8? Since most of the paper involves mouse data, why in Fig 1n did the authors switch to a human cell line?

The author conclude from Sup Fig 3d that IL11 induces “SASP” proteins in primary human hepatocytes. There are no controls, i.e. no samples cultured for the same time periods without IL11. These results could, for example, simply show cytokine production from hepatocytes cultured for varying lengths of time.

The experiment shown in Fig 1o seems to me to have little or no relevance to the main hypotheses. The idea that passage of human fibroblast cells to late passage is a useful model for aging is not well supported. In addition, the authors passage their cells only from passage 4 to 14, i.e. to a point far short of the passage number thought to reflect *in vitro* “senescence.” And the relevance of this data to mouse aging is in any case very speculative – mouse cells do not, for example, undergo senescence.

Many of the authors’ conclusions rest on differences between WT and IL11-negative (or IL11RA-negative) mice. But in many cases the effect of lower IL11 was notable even in young mice. (Examples: serum cholesterol, GTT and ITT, triglycerides, gastroc and soleus weight, etc.) Data of this sort do suggest that IL11 modulates metabolism, but they do not necessarily show that IL11 modulates aging. A related point: a claim that “IL11 modulates effects of aging on Trait X” requires a two-factor ANOVA, with age and IL11 status as cofactors, and an interaction term whose significance tests the claim. Merely

showing that an effect is significant in one genotype but not in the other does not test the validity of the conclusion of interest, i.e. that lower IL11 would slow aging.

Many of the other panels show the same problem in inference. Fig 3m is typical. It compares 75 week old mice, 100 week old mice, and 100 week old mice that received 25 weeks of treatment with anti-IL11. The data presented show that there are many phosphoproteins, as well as p16 and p21, that are lower in the antibody-treated mice. The authors conclude that these biochemical changes are due to “reduced IL11 signalling activity.” But they have not considered scenarios in which the anti-IL11 treatment leads to some alteration in neuro-endocrine or metabolic status that alters the endpoints shown in the figure panel. To pick one speculative example, if anti-IL11 antibody causes mice to lose appetite, and to eat less and lose weight, this would lead to changes in cellular activity in many tissues, but it would not be safe to attribute the changes to “IL11 signalling activity.”

The emphasis in Fig 4 on IL11-dependent changes in visceral adipose tissue seem to me to be important and well documented.

Figure 5 seems to me not to be appropriate for this paper. It could usefully be a first figure in a paper focused on administration of antibodies to IL11 in human clinical trials, but the data presented do not contribute materially to the conclusions drawn in the rest of the paper.

The discussion is brief, but judicious and on point; it raises a number of good ideas about parallels in the published literature and cogent future studies.

I note that body length is shorter in low-IL11 mice. It would be important to know if any of the IL11 models have lower IGF1 (or GH, though GH is much harder to measure), since low GH/IGF1 is often associated with longer lifespan in mice. Can the authors exclude the idea that the mutants with low IL11 simply are low in GH/IGF1, and show slower aging for that reason? (Admittedly, the experiments using anti-IL11 antibody in adult mice argue that developmental IL11 deficiency are not the whole story.)

The authors should report whether their colony is known (proven) to be specific pathogen free. They should state how frequently the colony is tested for SPF status, and state (if it's true) that all such tests were negative throughout the period of study.

In some cases the experimental data seem to have used male mice only. The authors should state clearly which data come from one sex only, and note that this is a limitation in their study. Many drug and mutant effects on mouse longevity affect one but not both sexes.

Some of their datasets use mice of both sexes. In these cases the authors need to tell us whether the comparison groups (e.g. IL11 mutant and controls) had equal numbers of males and females, and also do a statistical test to see if the results varied between the sexes.

As far as I can tell, nearly all of the data come from a single inbred strain of mice (B6, as usual). This is a fairly serious flaw – in some direct comparison, genetic effects noted in B6 mice are reproducible in other strain backgrounds less than half the time (see, for example, PMID 27618673, PMID 30595332.) I

would ask the authors to note this as a weakness of their approach, and if possible to include tests of genetically diverse mice (e.g. multiple F1s or similar) in future studies.

Referee #2 (Remarks to the Author):

Summary

In this manuscript, Widjaja et al. report on the pro-aging effects of interleukin 11 (IL11) in mice. The authors demonstrate that IL11 expression levels were elevated in various organs, including liver, white adipose tissue (WAT), and skeletal muscle, correlating with the activation of the ERK and mTOR pathway as well as the inhibition of the AMPK axis. Regulation of these pathways in mice upon aging was abrogated in animals lacking the IL11 receptor, aligning with an improved metabolic profile upon aging as compared with wild-type littermates. IL11 activity associated with aging biomarkers including epigenetic clocks, telomere length and mitochondrial DNA copy number. Further, IL11 triggered senescence and SASP phenotypes in fibroblasts and hepatocytes. Whole body deletion of IL11 protected animals from age-related increases in adipose tissue mass while preserving lean mass, correlating with increased muscle strength in IL11-deficient mice. IL11 deficiency in old age led to improved glucose handling, insulin sensitivity, and higher glucose usage. Inhibition of IL11 by an IL11-neutralizing antibody in old animals provoked an overall improvement of energy metabolism over a period of 25 weeks of treatment, documented by a decrease in WAT mass, liver damage and fibrosis, as well as preserved AMPK signaling. Anti-IL11 antibody treatment was particularly associated with an upregulation of beiging markers in WAT and a concomitant reduction in pro-inflammatory macrophages in this tissue. Overall, the authors conclude that IL11 may represent a therapeutic target for improving mammalian health span.

Main comments

In Western countries, an aging population imposes a significant burden on metabolic health. Indeed, obesity-related diseases, including diabetes, cardio-vascular disease, and cancer are expected to further increase in numbers. In this respect, the current manuscript by Widjaja et al. addresses an interesting and important topic in biomedical research, i.e. not primarily aiming at the expansion of life span but rather focusing on extending health span. This study employs an appropriate range of model systems and uses a large variety of state-of-the-art technology to address the study's hypothesis. The conclusions drawn by the authors are mostly supported by the experimental data. The study provides a clear translational angle on the question on how to improve mammalian health span which is a clear plus. On the other side, a number of main issues require additional attention by the authors: a) The authors imply that a beiging phenotype may be responsible for the major metabolic benefits of IL11 inhibition. As food intake is increased but body weight decreases, either physical activity, fecal energy excretion or enhanced energy expenditure will be responsible for the IL11-dependent effects. Actual measurements of these parameters should be included in the study to make firm conclusions. This holds also true for the potential translational potential of the study's findings as beige/brown fat does not play a predominant role in human energy balance. Do the effects of IL11 antibody treatment still appear under thermoneutral conditions? b) How do IL11 levels behave in young vs. old humans? Do you

observe the same regulation? Given the translational ambition of the manuscript, a clear indication for human relevance should be provided. c) The inhibitory approaches (IL11ra KO, IL11 KO, IL11 ab treatment) represent rather global, system-wide modalities. It remains unclear which organs mainly contribute to the overall metabolic improvements. While attempts have been made to characterize WAT in more detail, corresponding data on liver, pancreas and skeletal muscle are missing. Many studies have reported on IL11 as an anti-fibrotic, anti-senescence mediator so that the main novelty of this study relies on its potential targeting in metabolic health. A more comprehensive understanding of the main organ compartments involved thus appears appropriate. d) It remains somewhat unclear whether the authors propose IL11 to act in a para-/auto-/endocrine fashion and/or to which extent the intracellular levels in individual tissues contribute to the age-dependent increase in circulating levels.

Addition of corresponding data would significantly strengthen the case for publication.

Specific comments are listed below.

Specific comments

1. Fig 2a: Why is the WB showing male mice whilst the figure deals with the female phenotype? Please correct this inconsistency.
2. Fig 2s: Again, male mice are shown in this panel. How does RER behave in female animals?
3. Fig. 3: Will antibody treatment also prevent aging in female animals?
4. Fig. 3: The effects on fibrosis and aging biomarkers remain largely descriptive. As outlined in the main comments, further insights into the tissue-specific contributions to the systemic phenotype would be helpful to understand the MoA.
5. Fig. 3: Please demonstrate antibody specificity in the in vivo experiment.
6. Fig. 1 intermingles a number of model systems and different cell-types which is somewhat confusing and hard to follow. Please re-arrange and possibly use meaningful sub-sections.

Referee #3 (Remarks to the Author):

In the current manuscript, Widjaja and colleagues examine the role of the cytokine IL-11 in healthy aging. While the studies are, in the main comprehensive and well performed, I have some concerns with both the novelty and strategy of the studies.

Firstly, in a previous paper, this group already identified the pathway by which IL-11 inhibited the LKB1/AMPK pathway (Widjaja et al iScience 2022). As activation of LKB1/AMPK either genetically or pharmacologically increases age related impairments in synaptic remodelling (Samuel et al Nat Neuro, 2014) and rapamycin is known to be anti-aging in mice (Johnson et al Science 2013), the results reported here, while interesting, are somewhat expected.

IL-11 is well known to be a target in gastric cancer and in this context, targeting IL-11 appears to be a sound strategy. While the data presented here are comprehensive and compelling, this reviewer is not convinced that IL-11 inhibition could be a wise therapeutic strategy for age related metabolic disease. IL-11 is well known to be involved in osteoblast differentiation and mono-PEGylated recombinant human interleukin-11 has entered pre-clinical, non-human primate trials to increase bone density. What happens to bone density in the mice? Is there a risk that such a strategy could enhance age related osteoporosis?

As a follow up, I have some other concerns regarding IL-11 inhibition as a strategy for mammalian health span. Firstly, and somewhat paradoxically, other members of the gp130 receptor family, namely IL-6 (Carey et al Diabetes 2006) and CNTF (Watt et al. Nat Med 2006) activate AMPK and improve metabolic health. In fact, Axokine, human recombinant CNTF made it to Phase III clinical trials but failed due to autoantibody development. The mechanism behind the positive effects of these cytokines is via activation of STAT3. In addition, by blocking STAT3 in the pathway, as proposed in figure 1, IL-33 will be inhibited. It is well known that IL-33 plays critical roles in both innate and adaptive immune responses in mucosal organs, providing an essential axis for rapid immune responses and tissue homeostasis. Therefore, there is a degree of risk in targeting IL-11 therapeutically and one wonders whether activators of LKB1/AMPK wouldn't be a more sound therapeutic strategy.

I suggest that you consider Nature Aging as a suitable venue for your work. To transfer your manuscript there, please use our manuscript transfer portal. You will not have to re-supply manuscript metadata and files, unless you wish to make modifications, but please note that this link can only be used once and remains active until used. For more information, please see our manuscript transfer FAQ page.

Note that any decision to opt in to In Review at the original journal is not sent to the receiving journal on transfer. You can opt in to In Review at receiving journals that support this service by choosing to modify your manuscript on transfer. In Review is available for primary research manuscript types only.

Author Rebuttals to Initial Comments:

Point-by-point responses to the comments made by Reviewers at *Nature*

Referee #1 (Remarks to the Author):

Review of Widjaja...Cook

May 2023

Inhibition of an immunometabolic axis of mTORC1 activation extends mammalian healthspan

General summary:

1. This paper makes a number of highly interesting observations, which may deserve publication in a widely-read journal. The analyses of the effects of mutations that blunt IL11 action, and of anti-IL11 antibody, on metabolic indices and protein kinase substrates make quite a strong case that IL11 could play an under-appreciated role in aging and merits future study. The material on visceral adipose tissue (Fig 4) is particularly strong.

Response: We thank the reviewer for their most helpful comments, which we believe we have addressed in full and have greatly improved the revised manuscript.

2. There are portions of the paper that seem to me to be of much lower interest.

2a. The authors appear to believe that age-related appearance of senescent cells is a critical element in aging, but in my view the evidence for this idea is quite weak. Data showing effects, in mice, of IL11 on p16, p21, and multiple inflammatory effectors (which the authors, in keeping with too many others, lump together as “senescence associate secretory proteins”) are an important element of the paper and should be retained, although the interpretation of all these changes as reflecting creation of one single cell type (the “senescent” cells) is unpersuasive. But portions of the paper in which IL11 is added to cultured cells, in contrast, do not add materially to our understanding of the biology in the IL11-inhibited mice, and could be deleted.

Response: We understand from this comment that the Reviewer is perhaps sceptical of the specific importance of senescence in ageing and/or our representation of this. We wish to clarify that we did not intend to claim that senescence is the only hallmark of ageing regulated by IL11, indeed quite the opposite. Nevertheless, we did go into this aspect in some detail by exploring and providing examples of the effects of IL11 in primary human cells. This experiment allowed us to more thoroughly probe the direct and cell autonomous effects of IL11 on cells, mitochondrial function and inflammation (pSTAT3, pNF-kB) in the absence of secondary, endocrine, nerve-mediated or systemic effects that the reviewer rightly mentioned below. We were also able to establish the ERK- and mTOR-dependence of these IL11-mediated effects. For these reasons, we think it is important to retain the data relating to senescence in primary human cells in the manuscript. However, in not wishing to overstate the particular importance of senescence - we have moved these data to the supplement and reduced and rephrased the manuscript text relating to this aspect.

In fact, it was our intention, to demonstrate that IL11 regulates a variety of the hallmarks of ageing that include mitochondrial number (and function), telomere attrition, epigenetic alterations (in the broadest sense), chronic inflammation, altered intercellular communication and deregulated nutrient sensing, which comprise 6 of the 12 revised hallmark of ageing, as proposed in a recent review in *Cell*¹ (**Rebuttal Fig. 1**). This more generalised impact of IL11 on ageing pathobiology is made clearer in the revised manuscript and is specifically evident in the aggregate effect on lifespan, which we now show (**Fig. 5**)

Rebuttal Figure 1. The hallmark of ageing (from PMID: 36599349).

2b. The material in Figure 5, focused on production of new antibodies to IL11, should be deleted. It might deserve publication in a technical specialist journal, or could perhaps serve as Figure 1 in a separate paper focused on clinical trials, but does not add to the main theme of the current manuscript. I would just remove it.

Response: We agree with the Reviewer and have replaced **Fig. 5** with data from genetic and therapeutic studies on lifespan in both male and female mice. All data relating to the therapeutic IL11 antibody have been removed from the manuscript.

3. The authors are, in my view, often too quick to assert that some molecular phenomenon is “due to IL11 action.” I have note a couple of specific instances in the detailed notes below. It seems at least equally likely that many of these effects are secondary, i.e. dependent on one or more upstream changes that are themselves due to IL11 action. For an example, if IL11 causes adipose thermogenesis via UCP1, this could in turn modulate adipokines or cytokine levels in blood, which in turn alter muscle or liver function, and so forth. The data are of interest – but the interpretation needs to be more cautious, with recognition that IL11 dependent effects in unknown cells and tissues may have repercussions in many other parts of the system.

Response: We acknowledge this point but it does not detract from the logical argument that aggregate IL11 effects on healthspan (and now lifespan) are primary and important, even if some of the observed molecular or physiological effects may be ‘secondary’. As mentioned above, our data (**Extended Data Fig. 3**) showing direct, cell-autonomous effects of IL11 in human cells - in the absence of secondary paracrine, autocrine or nervous system-mediated effects - is useful in supporting our interpretations of a direct effect of IL11 on varied phenotypes, including signalling. The text has been adjusted to acknowledge interplay between cells and systems, which is mentioned in the revised discussion.

4. The authors very often make a (very common) statistical mistake. For example, when they have compared control to mutant mice at two ages, and see a statistically significant effect at the older age, they infer (incorrectly) that IL11 has affected the aging process. But the correct method is a two-factor ANOVA (age, mutation) with interaction term. If the interaction term is significant, this suggests that the IL11 effect at one age is different from that at the other age.

Response: We draw the reviewer’s attention to the figure legends of the original manuscript where it is stated that we performed “Two-way ANOVA with Sidak’s correction”, where appropriate. However, we acknowledge that we did not show, or discuss, interaction terms (whether significant or not) in the original manuscript, which was suboptimal.

As the reviewer highlights, some phenotypes are apparent in young adults in the two strains of knockout mice we used. We now detail age, sex and interaction terms in a new and very large **Extended Data Table 9**. It is the case that a few phenotypes do not have significant interaction terms, whereas the majority do. This may reflect underlying biology, phenotypic variance, and/or statistical power. Body composition and molecular phenotypes generally have significant interaction terms in both *Il11ra1* and *Il11*-deleted mice. We believe that frailty scores, a phenotype apparent only in aged mice, serve as a useful internal control for statistically significant interaction in *Il11*-deleted mice.

The therapeutic studies of healthspan that we present address the effects of IL11 in old age [only]. These data (**Fig. 3**, male mice and new **Extended Data Fig. 8**, female mice) show that for many of the phenotypes where we are not able to identify a significant interaction term between wildtype and knockout mice (e.g. body weight or telomere length in *Il11ra1*-deleted mice), we detect a significant effect of anti-IL11 (X203) as compared to IgG when administered to randomised mice late in life.

5. The main gap in the evidence presented is that the authors do not ever show us data to test the hypothesis that mice with lower IL11 signals are aging slowly. They do show data that IL11-dependent

events differ in normal vs IL11-blunted conditions – but they do not test age-related functions not already know to be IL11-modulated.

Response: We beg to disagree with the reviewer’s proposal that we did not “*test age-related functions not already know to be IL11-modulated*”. By way of example, we show that IL11 is an important contributor to age-related frailty, as indicated by a 27-point frailty score encompassing multiple ageing pathologies (including sight, hearing, fine motor function, gait, etc), which was previously unknown. We also demonstrated that IL11 impacts mitochondrial copy number, telomere lengths, and global measures of epigenetic age, which represent the established hallmarks of ageing and are novel findings. Additionally, we show evidence that IL11 upregulation in old age impacts a large program of white fat beiging leading to age-related adiposity, which is novel. We now also show therapeutic extension of lifespan across genders with anti-IL11 therapy, which is new. These are the first studies to show regulation of healthspan (and now lifespan) by a cytokine in a vertebrate.

As regards “the authors do not ever show us data to test the hypothesis that mice with lower IL11 signals are aging slowly”.

Response: In the original manuscript, we showed that mice randomised to anti-IL11 (X203) or IgG had similar baseline biomarkers of ageing, age-related organ dysfunction and frailty scores, the progression of which were collectively delayed by X203. This shows that reduced IL11 signalling is associated with slower ageing. We now also show and discuss interaction terms for the phenotypes studied in the *IL11* and *IL11ra1* deleted mice. This reveals age-specific effects on most phenotypes, as discussed above.

The most important of these is lifespan – I would be more enthusiastic for this already strong paper if they had data showing longer lifespan in one or more of the low-IL11 mouse models. In a revision, they should state clearly whether they have such data, and, if so, what they found – it would be unethical to withhold such data from the reader if it already exists.

Response: When we submitted the original study, our primary focus was on healthspan as IL11 effects on healthspan were novel and the implications for translation were readily apparent to us. We have - of course - been conducting lifespan studies in parallel but these studies lagged behind our healthspan studies, which we felt constituted a meaningful unit of publication on their own. That said, our lifespan studies have progressed over the past nine months, since our original submission and, in response to the specific request of the reviewer, we have now included these data that have been [and continue to be] acquired over the last three years, in our revised manuscript.

While the inclusion of these data at the current stage of ascertainment precludes us from reporting detailed median lifespan extension, lifespan extension at 90% mortality, and/or maximal lifespan, it does not diminish the overall findings. The genetic studies show that *Il11* KO mice, irrespective of sex, exhibit significantly longer lifespans compared to their wildtype littermates (**Rebuttal Fig. 2a, b** and new **Fig. 5a-d**).

To complement genetic studies of lifespan, and account for developmental and/or early-life effects of *Il11* deletion that might influence lifespan, we show that anti-IL11 therapy administered to randomised male and female mice from 525 days (75 weeks) of age extends lifespan (**Rebuttal Fig. 2c, d** and new **Fig. 5e-i**). As such, inhibition of IL11 not only extends healthspan but also increases lifespan, across genders. Autopsy data suggests that lifespan extension may be attributed to a reduced incidence of cancers, the most common cause of death in mice (*Il11* deletion: WT, 40/71; *Il11*^{-/-}, 3/15; P=0.0105 and anti-IL11 treatment: IgG, 19/33; X203, 2/10; P=0.0375 based on 2-population proportions analysis); however, these findings are preliminary and only presented in a descriptive fashion in the revised manuscript (new **Extended Data Table 6,7**).

Results from lifespan studies will be updated during the review process as the data accumulate. We anticipate that a sufficient number of additional mice will have died during this period to allow us to provide a more detailed description of lifespan effects.

Rebuttal Figure 2. Inhibition of IL11 signalling extends lifespan of male and female mice. Kaplan-Meier survival curves showing the cumulative survival probabilities for male and female **a,b** WT and *Il11*-deleted mice and **c,d** IgG and X203-treated mice starting from the age of 75 weeks (red dotted line). Statistical significance (p-value) was assessed by means of the log-rank (Mantel-Cox) test.

Similarly, if they have data on age-dependent functional outcomes, they should say so, and, if possible, show the results. Have they tested cognition? Have they tested collagen cross-linking, or hearing, or response to vaccination, or bone density, etc.? Is cancer delayed? Data showing deceleration of age-related changes in one or many of these tests would strengthen the paper. Data showing that these tests were NOT altered in one or more of the IL11-modulated mice would also be important to include, even if it complicates the reader's opinion about the potential use of anti-IL11 as an anti-aging therapy.

Response: The Reviewer rightly points out that changes occur in all cells, all organ systems and all physiologies with age. Given the limitations of examining this level of complexity in one laboratory and

one study, we directed our focus on key ageing phenotypes important for humans - frailty, sarcopenia/muscle function and age-related adiposity - all of which are adversely impacted by IL11.

It's important to note - and we hope that the Reviewer appreciates this point - that the frailty score we used was developed by the Jackson Laboratory's Mouse Neurobehavioral Phenotyping Facility². This 27-point frailty score evaluates a range of ageing phenotypes that include hearing (as suggested by the reviewer), eyesight, gait, vestibular disturbance, fine motor function, kyphosis, and other phenotypes. A noteworthy, idiosyncratic and distinctive age-specific [non-metabolic] phenotype observed in male mice is seminal vessel dilatation that we found to be modulated by inhibition of IL11 in both healthspan and lifespan studies (**Extended Data Table 2, 3, 6, 7**)³.

Regarding the delay in cancer development, the frailty score only takes into account gross and externally visible tumours that are rare and challenging to detect with our current statistical power. It is reported that 50-70% of mice die from cancers, and the observed lifespan extension with anti-IL11 therapy suggests a potential reduction or delay in the occurrence of cancers. This hypothesis is supported by the autopsy data presented in the supplementary information on lifespan (**Extended Data Table 6, 7**).

[REDACTED]

Following the suggestion of the Reviewer, we also assessed the effect of IgG versus anti-IL11 on bone parameters in collaboration with Dr Natalie Sims, Head of the Bone Cell Biology and Disease Unit at St. Vincent's Institute of Medical Research, Melbourne, Australia. We found that administration of anti-IL11 to a preliminary cohort of mice from mid-life to 100 w/o, as compared to IgG control did not show any adverse or protective effects across all examined parameters i.e femoral cortical shape/dimension (mediolateral width, anteroposterior width), cortical thickness, and bone strength (ultimate force and ultimate displacement) (**Rebuttal Fig. 5**). These data provide additional assurance on the safety of anti-IL11 for bone health but also show that, in this experiment, age-related bone health was not improved.

Rebuttal Figure 5. Anti-IL11 has no effect on bone shape, thickness, and strength in ageing mice.

Assessment of **a** mediolateral width, **b** antero-posterior width, **c** cortical thickness, **d** ultimate force, and **e** ultimate displacement from bones of mice that were given either IgG or X203 from 50 w/o to 110 w/o. **a-e** Data are shown as mean \pm SD; two-way ANOVA with Sidak's correction.

The data they do provide – on muscle strength, see Fig 2g – sometimes illustrate the complexities of interpretation. IL11-negative mice have higher muscle strength (adjusted for weight) even when young, and retain this strength advantage when older. Thus it would be incorrect to conclude that the IL11-negative mice are slower than controls at age-dependent muscle strength decline.

Response: With regard to muscle strength (original Fig. 2g is now **Fig. 2f**), the interaction term for full body grip strength is significant for both males ($P=0.0031$, **Extended Data Fig. 6f**) and females ($P=0.0068$, **Fig. 2f**). However, we stated “Age-dependent sarcopenia was apparent in both old female WT and IL11^{-/-} mice but with overall greater indexed muscle weights seen in IL11^{-/-} mice” in the original manuscript, so as not to over interpret the data.

It is the case that studies of germline genetic loss-of-function effects on healthspan and/or lifespan should be complemented by late-life interventions, even if two-way ANOVA interaction terms are supportive of late-life effects [we have added this phrase to the revised main text]. For this reason, we conducted therapeutic experiments in mice from the age of 75 to 100 weeks of age. At the onset of the study, we assessed the physiology (body weight, fat and lean mass, GTT, ITT, and frailty scores) of the mice, and subsequently randomly assigned them to one of three groups: no treatment, IgG control, or anti-IL11. These therapeutic studies which investigated multiple ageing pathologies, including muscle strength, show that anti-IL11 therapy slows, arrests or reverses ageing physiologies, depending on the phenotype. In the revised manuscript, we have adjusted the text to note the potential confounding impact of early life phenotypes in studies involving mice with germline loss-of-function.

I have provided below some more detailed comments for the authors to consider should they decide to revise the paper.

Detailed comments:

The authors introduce their study by asserting that "Ageing studies to date have focused mostly on lifespan extension that can be achieved at the expense of healthspan, merely represent extended periods of frailty and, in mice, primarily reflects reduced murine cancers." I do not think they are correct on any of these points. The best-known diets, mutations, and drugs that extend mouse lifespan have been shown to postpone multiple aspects of aging (cataracts, cognitive change, change in activity, kidney and heart pathology, and many others) in addition to their effects on cancer and lifespan. The assertion that these interventions merely create a long period of frailty is not, as far as I know, supported by any evidence. The authors are setting up a straw man here, and it's likely to mislead many readers.

Response: We apologise if our statement came across in this manner. It is the case, and supported by data (please refer to ⁶⁻⁹) that lifespan extension can come at the expense of healthspan and mice are thought mostly to die from cancers, which we support with citations. However, we recognize that we overstated this point and it was not balanced. We have largely rewritten the introduction accordingly and refer the reviewer to this section of the manuscript.

The idea that IL11 is "related to IL6" and must, by implication, be related to "inflammaging" does not make any sense to me.

Response: The IL6 family of cytokines are pro-inflammatory and co-regulated as an inflammatory module by LKB1¹⁰. IL6 is upregulated in ageing and is recognized as an inflammaging factor that predicts cognitive function¹¹. However, we recognise the lack of clarity about the idea that IL11 is an inflammaging factor and the text has been changed accordingly with more careful mention and explanation of the role of inflammation in ageing in the introduction and now in the discussion. The term 'inflammaging' may indeed be confusing to some and has been removed in its entirety from the revised manuscript

It is a serious oversimplification to state, as the authors do, that IL11 “.. causes liver steatosis, visceral adiposity and impaired muscle cell function, known features of ageing.” Steatosis and visceral adiposity are sometimes present in aged mice and people, and sometimes not, depending on diet, exercise, and genetics among other things. The implication that IL11 “causes...impaired muscle function” in aging is, similarly, quite misleading – there are many age-related changes in muscle cell biology, motor end-plate interactions, muscle fiber type switching, fibrosis, and myocyte volume that are multifactorial. To say that these changes are “caused by” IL11 is inaccurate.

Response: We have extensively rewritten the introductory text to avoid over-interpretation of the background information relating to IL11. We still introduce simple concepts that stimulated the study but in a much reduced fashion and on a more factual basis. We thank the reviewer for highlighting this point.

The stated hypothesis is that “IL11 is an inflammaging factor.” This is untestably vague. Hypotheses that include causal networks, for example that involve IL11 as a required element in some specific aspect(s) of aging, would be potentially valuable. The authors’ hypothesis is much less precise – it implies merely that some aspects of aging (unspecified) might be caused by, or accompanied by, or related to, IL11, and that modulation of IL11 in unspecific circumstances might produce some unspecified changes in aging via inflammation.

Response: As stated above, we have reworded the text accordingly and removed the word 'inflammaging' from the revised manuscript, in its entirety.

The authors state that “with age, there was progressive activation of ERK/p90RSK, inactivation of LKB1/AMPK and mTOR/p70S6K activation, which comprise the IL11 signalling module, in liver and muscle.” They do not provide any evidence that these changes in levels of various phosphoproteins actually are caused by age-related changes in IL11, either on its own or in conjunction with other age-

dependent changes in cell biology, organ composition, or other stimuli in vivo. They could, plausibly, reflect secondary changes triggered by IL11 dependent modification of neuroendocrine or paracrine factors.

Response: We agree that some signalling changes in tissues will involve secondary cell types or result from secondary physiologies. However, these observed outcomes do not detract from the fact that these readouts follow the primacy of IL11 inhibition, either genetically (*Il11* or *Il11ra1* KO) or therapeutically (anti-IL11).

We would like to restate that IL11 controls this [ERK/AMPK/mTOR] pathway, with immediate (15 min) and direct effect, in the cell types that we studied previously *in vitro*¹². Furthermore, in the manuscript we used IL11:EGFP mice to show that IL11 is upregulated in adipose cells, hepatocytes, fibroblasts and other cell types (**Fig. 1d** and **Extended Data Fig. 1e-g**). Our *in vitro* data in the manuscript show direct, ERK- and mTORC1-dependent, regulation of this specific pathway and paracrine effects of IL11, in relevant primary human cells (**Extended Data Fig. 3a, b**). These findings further support a cell autonomous action of IL11 on this signalling axis. In other *in vivo* studies, IL11 upregulation in cells is shown to colocalize with autocrine ERK activation, reaffirming its autocrine effects^{13,14}. We do however acknowledge the possibility of secondary effects and have reworded the manuscript accordingly, specifically in the revised discussion.

Fig 1b: no statistics are shown. The authors show phosphoproteins only, but not total protein level for any of the corresponding enzyme substrates, so it is not possible to determine if the ratio of phosphoprotein to total protein is age-sensitive or not.

Response: The reviewer may have missed **Extended Data Fig. 1a** of the original manuscript where the total proteins were presented. We now provide extensive densitometry analyses along with respective statistical data for all blots as a new **Source Datafile 1**.

Fig 1c: the colors add no information – they just make the numbers harder to read. The normalization to GAPDH is appropriate only if there is no age effect on GAPDH in any of the tissues – have the authors checked this out?

Response: There is no effect of GAPDH with age and the blots for **Fig. 1c** were provided as original **Extended Data Fig. 1c**. Colours have also been changed to grayscale for improved readability.

It is not good English to say that IL11 is “lowly expressed.”

Response: The wording has been revised.

Figure 1d is far too small to be useful.

Response: We have moved the *in vitro* senescence data to the supplement and made the immunofluorescence micrograph panels for **Fig. 1d** much larger. We thank the Reviewer for this comment as we believe this panel is important for the readership’s understanding of IL11 upregulation in various parenchymal and stromal cell types, across tissues.

The authors conclude that “IL11 activates ERK/mTOR and increases senescence markers with age”. But this is a premature conclusion. It is clear that the IL11ra1-negative mice have many abnormalities documented in Fig 1e. But whether this is due to direct effects of IL11 per se, or indirect effects (perhaps a change in neuroendocrine or cytokines that modulate liver biology), or a developmental abnormality in some IL11-dependent lineage, or lower body weights, or higher body temperatures, or altered liver cell death (AST, ALA), etc., is unknown.

Response: The Reviewer comments that “*IL11ra1-negative mice have many abnormalities documented in Fig 1e*”. However, the original **Fig. 1e** represents an immunoblot and it is not clear to which panel the reviewer is referring. As it happens, *IL11ra1-negative* mice do not exhibit abnormalities relating to *IL11ra1* deletion in adulthood but these mice do have some limited developmental deformities of the snout and facial bones, which are not apparent in *IL11-negative* mice¹⁵.

We believe the Reviewer is perhaps alluding to potential indirect/secondary effects of developmental abnormalities but these effects are taken into account in the therapeutic studies, which only begin when the mice are 75 weeks of age and the phenotypic, signalling, and molecular data largely phenocopy the genetic studies in both *IL11ra1-* and *IL11-*deleted mice.

It is not relevant that “BHB is advocated as an anti-ageing supplement.” If the authors know of data to show that BHB actually slows aging in mice, they should state it.

Response: All references to BHB and the associated data have been removed from the revised manuscript.

Fig 1m is hard to interpret without evidence that the “clock” is an effective index of age in normal mice. What age estimate does it give for 20 week old mice, for example? And does the apparent difference in biological age between mutant and normal mice get larger as the mice get older, as the model would predict?

Response: We measured the epigenetic age through a third party CRO (Zymo Research: <https://zymoresearch.eu/pages/mouse-dnage>) that has established clocks for blood, liver, muscle, and brain in the mouse. To develop and validate the accuracy of these clocks, Zymo has provided confirmation to us that they have conducted an in-house validation using the respective organs isolated from mice aged 12, 25, 40, 60, and 70 weeks. These are proprietary data that we do not have access to. We did not measure serial differences in the clock for our ageing *Il11*-deleted versus wildtype mice. However, we did assess the epigenetic clock in therapeutic studies, where we observed similarly large effect sizes.

Fig 1n: did the authors look at other indices of “senescence” in IL11-treated fibroblasts, or just p16 and p21? Did the authors look at other “SASP” cytokines, or just at IL6 and IL8? Since most of the paper involves mouse data, why in Fig 1n did the authors switch to a human cell line?

Response: Original Fig. 1n is now **Extended Data Fig. 3a**. We believe cross-species validation, and in particular the use of human cells is important for robustness and translatability of findings in model organisms. For this reason and to study the direct effects of IL11 on a range of molecular ageing phenotypes (p16/21 expression, mitochondrial number/function, telomere length, cell cycling conditions, SASP secretion, signalling activation, pro-inflammatory signalling), we conducted studies on human cells.

For senescence, we focused on p16 and p21, which are the classical senescence markers and for SASP proteins, our focus was on IL6, IL8, and IL11 itself. In hepatocytes, which are less well characterised from a senescence/SASP point of view, we extended SASP analysis to IL6, IL8, LIF, VEGFA, HGF, CCL2, CXCL1, CXCL5, CXCL6 and CCL20, as detailed in the original submission.

The particular importance of p16 for senescence, healthspan and lifespan in mice has been established in a number of studies, including by the selective deletion of p16 cells in the adult mouse as described in a paper published in Nature entitled “*Naturally occurring p16(Ink4a)-positive cells shorten healthy lifespan*”¹⁶.

The author conclude from Sup Fig 3d that IL11 induces “SASP” proteins in primary human hepatocytes. There are no controls, i.e. no samples cultured for the same time periods without IL11. These results could, for example, simply show cytokine production from hepatocytes cultured for varying lengths of time.

Response: Original Extended Data Fig. 3d is now shown as **Extended Data Fig. 3e**. We believe this comment has come about as a result of the Reviewer's misunderstanding. We used time-matched, unstimulated controls that have been cultured and harvested simultaneously with other time points to specifically avoid the perception of “*simply show cytokine production from hepatocytes cultured for varying lengths of time*”. In this experiment, IL11 was added at different timings to stimulate cells; for instance, at 3 PM on Day 1 for the 24-hour time point and at 9 AM on Day 2 for the 6-hour time point. Supernatants for the unstimulated control, 6, and 24-hour time points were collected at the same time. We have revised the methods on Olink proximity assay to improve clarity on this matter.

The experiment shown in Fig 1o seems to me to have little or no relevance to the main hypotheses. The idea that passage of human fibroblast cells to late passage is a useful model for aging is not well supported. In addition, the authors passage their cells only from passage 4 to 14, i.e. to a point far short of the passage number thought to reflect in vitro “senescence.” And the relevance of this data to mouse aging is in any case very speculative – mouse cells do not, for example, undergo senescence.

Response: Original Fig. 1o is now **Extended Data Fig. 4a**. We beg to disagree with the reviewer on this point and believe that studying human cells undergoing replicative senescence is useful: it enables direct assessment of IL11 effects (non-endocrine and not secondary) and - importantly - extends findings beyond mice to human samples, which is a prerequisite for translatability. Human fibroblasts in culture senesce at different rates depending on donor age, fibroblast type, and also on the number of population doublings, which is related to plating density rather than passage number. In our experimental set up, cells were no longer replicating at passage 14 and experiments at later passage were not possible. As to the model of replicative senescence, we use this as a model and - as all models have limitations - we acknowledge that this model has limitations. However, we believe that the replicative senescence model is informative, as shown previously^{17–21}.

As regards “*mouse cells do not, for example, undergo senescence*”: the importance of senescence for lifespan and healthspan in mice has been established in a number of studies¹⁶.

These things said, we do not intend to overemphasise ‘senescence’ as the dominant ageing hallmark regulated by IL11. Our study encompasses the impacts of IL11 on up to 6 out of the 12 accepted hallmarks of ageing (as mentioned above). Thus, to simplify the manuscript and to de-emphasise this point, we have moved the data relating to replicative senescence to the supplement (original Fig. 1o, p which are now **Extended Data Fig. 4a, c**) and have reduced the text of this section of the main manuscript.

Many of the authors’ conclusions rest on differences between WT and IL11-negative (or IL11RA-negative) mice. But in many cases the effect of lower IL11 was notable even in young mice. (Examples: serum cholesterol, GTT and ITT, triglycerides, gastroc and soleus weight, etc.) Data of this sort do suggest that IL11 modulates metabolism, but they do not necessarily show that IL11 modulates aging. A related point: a claim that “IL11 modulates effects of aging on Trait X” requires a two-factor ANOVA, with age and IL11 status as cofactors, and an interaction term whose significance tests the claim. Merely showing that an effect is significant in one genotype but not in the other does not test the validity of the conclusion of interest, i.e. that lower IL11 would slow aging.

Response: Comments relating to early life effects are valid and we were aware of these points. This is why we used anti-IL11 therapy, as compared to IgG (placebo) to study healthspan (and now lifespan) in old mice. This experiment addresses potential confounding factors associated with inhibition of IL11 signalling during development or early life, which was apparent for some phenotypes but not for others.

While we had already used two-way ANOVA, as suggested by the reviewer, interaction terms were not presented in our first submission, which was an error on our part and may have coloured the reviewer’s point-of-view. This omission has now been rectified and the additional statistical evidence provided further enables distinction between follow through of early life phenotypes and age-specific effects, which are many. The text has been revised accordingly and we present all interaction terms in new **Extended Data Table 9**.

Many of the other panels show the same problem in inference. Fig 3m is typical. It compares 75 week old mice, 100 week old mice, and 100 week old mice that received 25 weeks of treatment with anti-IL11. The data presented show that there are many phosphoproteins, as well as p16 and p21, that are lower in the antibody-treated mice. The authors conclude that these biochemical changes are due to “reduced IL11

signalling activity.” But they have not considered scenarios in which the anti-IL11 treatment leads to some alteration in neuro-endocrine or metabolic status that alters the endpoints shown in the figure panel. To pick one speculative example, if anti-IL11 antibody causes mice to lose appetite, and to eat less and lose weight, this would lead to changes in cellular activity in

many tissues, but it would not be safe to attribute the changes to “IL11 signalling activity.”

Response: In the therapeutic studies, we show that anti-IL11 given to mice aged 75-100 weeks slows, and in some instances reverses ageing phenotypes as compared to 75 w/o pathologies. These effects are associated with a reduction in all the components of the IL11-regulated ERK/LKB1/AMPK/mTOR signalling pathway. This represents a cell autonomous pathway that acts in autocrine in most cell types, eg. hepatocytes, muscle cells, fibroblasts. While it is possible that inhibition of IL11 has secondary and indirect effects on the IL11 signalling pathway via additional/unknown molecules this seems less likely than a direct effect of IL11, which is upregulated in the tissues where we show effects. In the current manuscript we show a direct effect of IL11 on this pathway *in vitro* (in primary human cells), in the absence of other (metabolic, neuro-endocrine, systemic) effects.

Thus, in keeping with Occam’s razor principle, we think that the observed effects on the IL11 signalling pathway that occur with anti-IL11 therapy most likely represent inhibition of said pathway by the IL11 neutralising therapy. We do however acknowledge possible secondary effects in general in the revised manuscript.

Regarding the Reviewer’s speculative example “*if anti-IL11 antibody causes mice to lose appetite, and to eat less and lose weight*”: We point out that this is not valid as our data on metabolic cage assessment by phenomaster presented in the original manuscript showed that administration of X203 (anti-IL11) is associated with increased food intake whilst activity levels are similar between the study groups, perhaps the reviewer missed these data.

The emphasis in Fig 4 on IL11-dependent changes in visceral adipose tissue seem to me to be important and well documented.

Response: We thank the reviewer for recognising the strengths of this section of the study which we focused on given the notable effects of IL11 loss-of-function on metabolic phenotypes.

Figure 5 seems to me not to be appropriate for this paper. It could usefully be a first figure in a paper focused on administration of antibodies to IL11 in human clinical trials, but the data presented do not contribute materially to the conclusions drawn in the rest of the paper.

Response: We agree with and thank the Reviewer for this suggestion, This figure, along with all related supplementary panels and corresponding text and discussion, has been removed and replaced with lifespan data.

The discussion is brief, but judicious and on point; it raises a number of good ideas about parallels in the published literature and cogent future studies.

Response: Thank you, we have updated the discussion section in the light of the lifespan studies. We now also discuss some of the limitations raised by the reviewer above and have expanded the discussion a little.

I note that body length is shorter in low-IL11 mice. It would be important to know if any of the IL11 models have lower IGF1 (or GH, though GH is much harder to measure), since low GH/IGF1 is often associated with longer lifespan in mice. Can the authors exclude the idea that the mutants with low IL11 simply are low in GH/IGF1, and show slower aging for that reason? (Admittedly, the experiments using anti-IL11 antibody in adult mice argue that developmental IL11 deficiency are not the whole story.)

Response: The published data suggests that IL11 has direct effects on osteoclasts, osteoblasts, and osteocytes although the literature is convoluted with IL11 variably reported to stimulate or inhibit bone phenotypes²². As such, it would seem unlikely that IL11, which has no known function in the hypothalamus or pituitary gland, additionally impacts the GH/IGF1 axis.

For completeness, and to directly address the reviewer's question, we have performed new experiments during revision to assess IGF1 levels (Mouse/Rat IGF-1 Quantikine ELISA kit, #MG100, R&D systems) in male and female mice with *Il11* deletion, as compared to littermate controls. In addition, we measured IGF1 levels in mice treated with IgG or anti-IL11 from 75-100 weeks of age. With either genetic or pharmacologic inhibition of IL11 signalling we found that IGF1 levels are mildly and significantly increased and this appears to be an age related phenomenon by two-way Anova with Sidak's correction (**Rebuttal Fig. 6**).

The physiological or pathophysiological relevance of this effect, if any, is unknown but we can exclude the idea that the beneficial effects associated with IL11 loss-of-function are due to low IGF1 levels. We feel these data are somewhat tangential to the main findings of our study and have not included them in the revised manuscript and show them here only, for the reviewer's information.

For the benefit of future readers of our manuscript, we have removed the data relating to body lengths from the text as these data are not central to the study and may be conflated given the published associations of dwarfism and lifespan, as exemplified in this example.

Rebuttal Figure 6. Beneficial effects associated with IL11 inhibition are not due to low IGF1 levels. IGF-1 levels in **a** young (12 w/o) and old (105 w/o) *Il11*^{-/-} male and female mice, and **b** in 100 w/o mice treated with IgG or X203. **a,b** Data are shown as mean \pm SD. **a** Two-way ANOVA with Sidak's correction (males: genotype $P=0.0821$, age $P=4.63e^{-11}$, interaction $P=0.0164$; females: genotype $P=0.1299$, Age $p=6.45e^{-6}$, interaction $P=0.0402$), **b** two-tailed Student's t-test.

The authors should report whether their colony is known (proven) to be specific pathogen free. They should state how frequently the colony is tested for SPF status, and state (if it's true) that all such tests were negative throughout the period of study.

Response: Our mice colonies hold SPF status and they undergo quarterly and annual tests for a common list of pathogens. The room housing our animals is MNV and Helicobacter positive, and these particular

pathogens are deemed acceptable within our SPF facility. These details have been added into the methods section: 'Animal models'.

In some cases the experimental data seem to have used male mice only. The authors should state clearly which data come from one sex only, and note that this is a limitation in their study. Many drug and mutant effects on mouse longevity affect one but not both sexes.

Response: For *Il11*-deleted mice, healthspan is assessed in both male and female mice. We found that healthspan indices are reduced to approximately equal extents in male and female mice, with greater magnitude of adiposity reductions in female *Il11* KO mice. We have now made the text more clear as to when we are studying male or female mice.

In the revision, we have added new healthspan (revised **Extended Data Fig. 8**) data in female mice administered anti-IL11 to complement the male data that we presented in the original submission. Female mice administered anti-IL11 recapitulated the metabolic and other (GTT, ITT, body composition, body weight, grip strength and frailty) healthspan phenotypes observed in male mice.

In the new lifespan studies that we present in revised **Fig. 5**, male and female mice were used in both the genetic and the therapeutic experiments. While the reviewer notes "*Many drug and mutant effects on mouse longevity affect one but not both sexes*", this does not apply to inhibition of IL11 that similarly improves lifespan in both sexes.

*Some of their datasets use mice of both sexes. In these cases the authors need to tell us whether the comparison groups (e.g. *Il11* mutant and controls) had equal numbers of males and females, and also do a statistical test to see if the results varied between the sexes.*

Response: The number of mice per group are reported in the figure legends. We do not think it wise to directly compare data between male and female mice as these experiments were performed in batches and over many years with variations in staff, reagents, time of the year, etc. We thus prefer to compare littermates of the same sex that were phenotyped at the same time.

As far as I can tell, nearly all of the data come from a single inbred strain of mice (B6, as usual). This is a fairly serious flaw – in some direct comparison, genetic effects noted in B6 mice are reproducible in other strain backgrounds less than half the time (see, for example, PMID 27618673, PMID 30595332.) I would

ask the authors to note this as a weakness of their approach, and if possible to include tests of genetically diverse mice (e.g. multiple F1s or similar) in future studies.

Response: It is inaccurate to say that our data are derived solely from a “single inbred strain of mice”. We show the role for IL11 in healthspan using *Il11ra1*-deleted mice and *Il11*-deleted mice. *Il11ra1*-deleted mice are a recombinant strain derived from an intercross of C57BL/6J and 129 genotypes, and represent a recombinant inbred line from these lineages²³. These mice differ in genetic background from *Il11*-deleted mice that are on a pure C57BL/6J background. For this reason, these strains do not always phenocopy each other in physiology or pathology, as we have detailed in the publication “*Similarities and differences between IL11 and IL11RA1 knockout mice ...*”¹⁵. Furthermore, we show effects of IL11 on ageing phenotypes in primary human cells, not just mice. In therapeutic studies, we show that anti-IL11 extended healthspan in mice on a C57BL/6J background across different batches, and in both sexes and also in a local C57BL/6J strain (that differs from the JAX strain) in preliminary studies, which we do not show. We acknowledge the importance of replicating our findings in additional mouse strains, and strongly encourage this experiment.

Referee #2 (Remarks to the Author):

Summary

In this manuscript, Widjaja et al. report on the pro-ageing effects of interleukin 11 (IL11) in mice. The authors demonstrate that IL11 expression levels were elevated in various organs, including liver, white adipose tissue (WAT), and skeletal muscle, correlating with the activation of the ERK and mTOR pathway as well as the inhibition of the AMPK axis. Regulation of these pathways in mice upon aging was abrogated in animals lacking the IL11 receptor, aligning with an improved metabolic profile upon aging as compared with wild-type littermates. IL11 activity associated with aging biomarkers including epigenetic clocks, telomere length and mitochondrial DNA copy number. Further, IL11 triggered senescence and SASP phenotypes in fibroblasts and hepatocytes. Whole body deletion of IL11 protected animals from age-related increases in adipose tissue mass while preserving lean mass, correlating with increased muscle strength in IL11-deficient mice. IL11 deficiency in old age led to improved glucose handling, insulin sensitivity, and higher glucose usage. Inhibition of IL11 by an IL11-neutralizing antibody in old animals provoked an overall improvement of energy metabolism over a period of 25 weeks of treatment, documented by a decrease in WAT mass, liver damage and fibrosis, as well as preserved AMPK signaling. Anti-IL11 antibody treatment was particularly associated with an upregulation of beiging markers in WAT and a concomitant reduction in pro-inflammatory macrophages in this tissue. Overall, the authors conclude that IL11 may represent a therapeutic target for improving mammalian health span.

Main comments

In Western countries, an aging population imposes a significant burden on metabolic health. Indeed, obesity-related diseases, including diabetes, cardio-vascular disease, and cancer are expected to further increase in numbers. In this respect, the current manuscript by Widjaja et al. addresses an interesting and important topic in biomedical research, i.e. not primarily aiming at the expansion of life span but rather focusing on extending health span. This study employs an appropriate range of model systems and uses a large variety of state-of-the-art technology to address the study's hypothesis.

The conclusions drawn by the authors are mostly supported by the experimental data. The study provides a clear translational angle on the question on how to improve mammalian health span which is a clear plus.

Response: We thank the reviewer for recognising the strengths and importance of our study as well as its translational value. We now include new data on lifespan extension following inhibition of IL11 signalling either genetically or therapeutically, in both male and female mice.

On the other side, a number of main issues require additional attention by the authors:

a) The authors imply that a beige phenotype may be responsible for the major metabolic benefits of IL11 inhibition. As food intake is increased but body weight decreases, either physical activity, fecal energy excretion or enhanced energy expenditure will be responsible for the IL11-dependent effects. Actual measurements of these parameters should be included in the study to make firm conclusions. This holds also true for the potential translational potential of the study's findings as beige/brown fat does not play a predominant role in human energy balance. Do the effects of IL11 antibody treatment still appear under thermoneutral conditions?

Response: We thank the reviewer for this comment, which is important to address. With regard to physical activity, we presented locomotor activity data in the original submission as part of **Extended Data Fig. 6h** (*Il11*^{-/-} mice) and **Extended Data Fig. 7b** (anti-IL11 treatment). These data showed no difference in the physical activity of mice with IL11 inhibition as compared to control mice.

In response to the reviewer's comment that energy may be lost by faecal excretion, we have now performed bomb calorimetry of faecal samples from *Il11*-deleted mice as compared to wildtype mice

and also aged mice administered IgG, as compared to mice given anti-IL11. These data show no differences in faecal energy excretion with inhibition of IL11 signalling (**Rebuttal Fig. 7a,b** and new **Extended Data Fig. 6i, 7c**).

Rebuttal Figure 7. No differences in faecal energy excretion with inhibition of IL11 signalling. Faecal caloric density as measured by bomb calorimetry in **a** 95-105 w/o male *Il11*^{-/-} mice compared to WT mice and in **b** 115 w/o male mice treated with IgG or X203, initiated at the age of 75 weeks. **a,b** Data are shown as mean \pm SD; two-tailed Mann Whitney test.

Given that there is no difference in physical activity or energy excretion in the context of increased food intake the only explanation for weight loss is - as outlined by the reviewer - related to *enhanced energy expenditure*. This is in keeping with the increased core body temperature seen in *Il11ra1*-deleted (**Fig. 1h**), *Il11*-deleted (**Fig. 2e; Extended Data Fig. 6e**), and anti-IL11 treated (**Fig. 3h; Extended Data Fig. 8h**) mice in both sexes, as compared to their respective controls, which reflects reactivation of an age-repressed program of white fat beiging.

It must be said that while we showed improved metabolic flexibility in mice with IL11 inhibition, we were not able to demonstrate a difference in energy expenditure by metabolic cage analysis. This is not unique to our study and other studies that show white fat beiging have not succeeded in showing increased energy expenditure using metabolic cages and have reverted to *in vitro* methods, which we do not have access to and we suggest is not needed given increased energy expenditure is evident by exclusion of other possibilities²⁴.

It is not possible for us to house aged mice for extended periods in isolation in metabolic cages, to assess if metabolic benefits are influenced by body temperature. We acknowledge this limitation in the

revised discussion but point out that IL11 upregulation is unrelated to ambient temperature as all mice (young and old) are housed at the same temperature but IL11 is only increased by age.

We agree with the reviewer that white fat beiging, specifically UCP1-mediated thermogenesis, is particularly important in mice; it remains contentious in humans. We note that loss of WAT beiging is an ageing phenotype in mice and humans and that, as with all other ageing phenotypes we studied, inhibition of IL11 mitigated this pathology. It is possible that in humans, ageing-related loss of WAT beiging is less important and that other human-specific ageing phenotypes are more relevant for anti-IL11 therapy, which we mention. We highlight that inhibition of IL11 signalling was associated with improvement in all ageing phenotypes we studied, a number of which are not directly under metabolic control, which includes thymic function (please refer to comments to Reviewer 1 and **Rebuttal Fig. 3, 4**). We now also show lifespan data with the knowledge that mice die mostly from cancers, rather than from metabolic dysfunction.

b) How do IL11 levels behave in young vs. old humans? Do you observe the same regulation? Given the translational ambition of the manuscript, a clear indication for human relevance should be provided.

Response: IL11 is upregulated in old humans, which we referenced in the original discussion “*Recently, serum IL11 levels were shown to be elevated in the very old²⁵*”.

[REDACTED]

c) The inhibitory approaches (IL11ra KO, IL11 KO, IL11 ab treatment) represent rather global, system-wide modalities. It remains unclear which organs mainly contribute to the overall metabolic improvements. While attempts have been made to characterize WAT in more detail, corresponding data on liver, pancreas and skeletal muscle are missing. Many studies have reported on IL11 as an anti-fibrotic, anti-senescence mediator so that the main novelty of this study relies on its potential targeting in metabolic health. A more comprehensive understanding of the main organ compartments involved thus appears appropriate.

Response: We show that inhibition of IL11 in old age modulates the key ageing pathways to mitigate the effects of ageing in all organs and physiologies that we studied that included global assessment of frailty (hearing, sight, posture, gait, fine motor function, etc). We now also show significant effects on lifespan across genders using genetic and pharmacologic means. As such, our study assesses the global effects of IL11 on ageing in the broadest sense, including thymic ageing and immunosenescence (as discussed above). We acknowledge that we focused on mechanisms associated with lesser adiposity in mice lacking IL11 signalling given the large effect sizes seen in this phenotype. We did provide molecular data

regarding liver and skeletal muscle in the original submission, which were presented in the main and Extended Data Figures.

In response to the Reviewer's suggestion and to better orient our study on ageing *per se* (as opposed to age-related metabolism), we have reworded the title and changed the emphasis in the abstract, results, and discussion. This is particularly so as the new lifespan data (**Fig. 5**) that we now show, to some extent, supersedes some of the more detailed tissue-specific aspects of metabolism presented in the original manuscript. Of note, mice die primarily from cancers and thus by including the lifespan data, there is now some lesser emphasis now on metabolism overall.

d) It remains somewhat unclear whether the authors propose IL11 to act in a para-/auto-/endocrine fashion and/or to which extent the intracellular levels in individual tissues contribute to the age-dependent increase in circulating levels.

Response: We, and others, have shown that IL11 primarily signals in autocrine and paracrine, which we reconfirmed in the *in vitro* studies of the current manuscript. We are not aware of an (neuro)endocrine effect of IL11. We observed IL11 upregulation in all tissues where we have measured it in old age (including brain, data not shown). As highlighted in the skeletal muscle example, it is likely that multiple cell types (parenchymal and stromal) upregulate IL11 in old age.

Addition of corresponding data would significantly strengthen the case for publication.

Specific comments are listed below.

Specific comments

1. Fig 2a: Why is the WB showing male mice whilst the figure deals with the female phenotype? Please correct this inconsistency.

Response: This has been corrected: original Fig. 2a is now **Fig. 1n**.

2. Fig 2s: Again, male mice are shown in this panel. How does RER behave in female animals?

Response: RER data from original Fig. 2s is now moved to (as part of) **Extended Data Fig. 6h**. We studied mice in metabolic cages in the context of genetic loss-of-function or after administration of therapeutic antibody or IgG control. In both instances, male mice were used as sufficient female mice were not available when metabolic cage studies were performed. In the revision, we have added new healthspan (revised **Extended Data Fig. 8**) and lifespan (revised **Fig. 5**) data in female mice. Female mice administered anti-IL11 recapitulated the metabolic and other (GTT, ITT, body composition, body weight, grip strength and frailty) healthspan phenotypes observed in male mice, in the original submission. Female mice administered anti-IL11 also live longer than those given IgG control. We did not study these additional female mice in metabolic cages as they were prioritised for lifespan studies that are now described in the revision and in accordance with the timeline granted to us for this resubmission. We acknowledge this is a shortcoming but do not think it detracts from the study overall. We hope the reviewer understands this point.

3. Fig. 3: Will antibody treatment also prevent aging in female animals?

Response: In the revision, we present new data showing anti-IL11 reduces progression, stabilises, or reverses a range of ageing phenotypes in females (revised **Extended Data Fig. 8**). Of equal importance, we now show anti-IL11 extends lifespan in aged female mice (revised **Fig. 5**).

4. Fig. 3: The effects on fibrosis and aging biomarkers remain largely descriptive. As outlined in the main comments, further insights into the tissue-specific contributions to the systemic phenotype would be helpful to understand the MoA.

Response: We refer the Reviewer to our comments above. Fibrosis is a general feature of ageing, and a major hallmark of senescence, and we show that age-related fibrosis is reversed by anti-IL11, for the first time.

5. Fig. 3: Please demonstrate antibody specificity in the in vivo experiment.

Response: This antibody has been characterised in great detail in a study that we cited in the method section: 'Custom-made antibodies'.

6. Fig. 1 intermingles a number of model systems and different cell-types which is somewhat confusing and hard to follow. Please re-arrange and possibly use meaningful sub-sections.

Response: We apologise for this, it is indeed a very large study with many data. In the revision we have tried to improve clarity by defining the study in subsections, as suggested by the reviewer. In addition, we have moved a large amount of data to the Extended Data Figures and added the lifespan data in lieu of the therapeutic antibody data. We think these changes have simplified matters and thank the Reviewer for this comment.

Referee #3 (Remarks to the Author):

In the current manuscript, Widjaja and colleagues examine the role of the cytokine IL-11 in healthy aging. While the studies are, in the main comprehensive and well performed, I have some concerns with both the novelty and strategy of the studies.

Firstly, in a previous paper, this group already identified the pathway by which IL-11 inhibited the LKB1/AMPK pathway (Widjaja et al iScience 2022). As activation of LKB1/AMPK either genetically or pharmacologically increases age related impairments in synaptic remodelling (Samuel et al Nat Neuro, 2014) and rapamycin is known to be anti-aging in mice (Johnson et al Science 2013), the results reported here, while interesting, are somewhat expected.

Response: IL11 has never before been linked before with healthspan (and now lifespan). We show that IL11 is upregulated across tissues, in distinct cell types to drive ageing-related pathologies across tissues (now including lifespan). These data are novel and not expected. The fact that the benefits of IL11 inhibition are mediated via its effects on the key ageing pathways (ERK, AMPK, mTOR, JAK/STAT and NFκB) is expected as any/all healthspan and/or lifespan extending interventions should impact these pathways, to some degree. Very many stimuli variably activate ERK/mTORC and inhibit LKB1/AMPK but extremely few, indeed no cytokine ever, have been shown to regulate healthspan and lifespan in mice. This is the first study to show regulation of healthspan and lifespan by a cytokine in a vertebrate. Moreover, we show that a neutralising IL11 antibody provides a therapeutic opportunity for extension of these beneficial phenotypes, which is also novel.

IL-11 is well known to be a target in gastric cancer and in this context, targeting IL-11 appears to be a sound strategy. While the data presented here are comprehensive and compelling, this reviewer is not convinced that IL-11 inhibition could be a wise therapeutic strategy for age related metabolic disease. IL-11 is well known to be involved in osteoblast differentiation and mono-PEGylated recombinant human interleukin-11 has entered pre-clinical, non-human primate trials to increase bone density. What happens to bone density in the mice? Is there a risk that such a strategy could enhance age related osteoporosis?

Response: The author refers to gain-of-function effects of recombinant human IL11 (rhIL11) on bone health. We would like to point out to the reviewer that, as outlined in previous publications and reviews, the effects of high dose rhIL11 (IL11 gain-of-function) in mice and humans is unrelated to its true biology and cannot be used as a basis to infer the effects of IL11 loss-of-function²⁶.

By way of example, injection of rhIL11 to mice, rats, monkeys or humans causes acute thrombocytosis and increases in platelet counts. In contrast, (1) genetic deletion of *Il11ra1* or *Il11* in mice or *IL11RA* in humans or (2) administration of anti-IL11 or anti-IL11RA to mice, monkeys or humans has no effect on platelet counts. Thus, the comment on the use of “*mono-PEGylated recombinant human interleukin-11*” by the reviewer is unrelated to our study (and IL11 biology).

To reassure the reviewer of the safety of anti-IL11 for bone health, we have generated data (**Rebuttal Fig. 5** and shown again below) in collaboration with an expert in bone health: Dr Natalie Sims, Head of the Bone Cell Biology and Disease Unit at St. Vincent's Institute of Medical Research, Melbourne, Australia. In these studies, we examined the effects of anti-IL11 given to mice from mid-life to 100 w/o, as compared to IgG control. In short, there was no adverse effect of anti-IL11 (X203) seen for all bone parameters examined.

Rebuttal Figure 5. Anti-IL11 administration for 60 weeks has no effect on bone shape, thickness, and strength in old mice. Assessment of **a** mediolateral width, **b** antero-posterior width, **c** cortical thickness, **d** ultimate force, and **e** ultimate displacement from bones of mice that were given either IgG or X203 from 50 w/o to 110 w/o. **a-e** Data are shown as mean \pm SD; two-way ANOVA with Sidak's correction.

As a follow up, I have some other concerns regarding IL-11 inhibition as a strategy for mammalian health span. Firstly, and somewhat paradoxically, other members of the gp130 receptor family, namely IL-6 (Carey et al Diabetes 2006) and CNTF (Watt et al. Nat Med 2006) activate AMPK and improve metabolic health. In fact, Axokine, human recombinant CNTF made it to Phase III clinical trials but failed due to autoantibody development. The mechanism behind the positive effects of these cytokines is via activation of STAT3. In addition, by blocking STAT3 in the pathway, as proposed in figure 1, IL-33 will be inhibited. It is well known that IL-33 plays critical roles in both innate and adaptive immune responses in mucosal organs, providing an essential axis for rapid immune responses and tissue homeostasis. Therefore, there is a degree of risk in targeting IL-11 therapeutically and one wonders whether activators of LKB1/AMPK wouldn't be a more sound therapeutic strategy.

Response: The Reviewer raises hypothetical concerns about targeting IL11 in humans clinical settings, which can only truly be addressed in late stage clinical trials. However, data from human *IL11RA* knockouts²⁷, mouse *Il11ra1* knockouts (this study), mouse *Il11* knockouts (this study), and experiments using long term antibody dosing in mice (this study and ²⁸) suggest a favourable safety profile associated with therapeutic inhibition of IL11. As is evident in our current study, anti-IL11 therapy in old mice extends healthspan and also lifespan rather than being detrimental. It's worth noting that the first data released from a recent phase 1 clinical trial of anti-IL11 (NCT05331300) reported an “*excellent safety profile [sic]*” ²⁹.

The reviewer refers to the putative positive effect of axokine (recombinant CNTF) and IL6 gain-of-function induced STAT3 signalling as of possible relevance. We argue that this is not the case: While IL11 and other members of the IL6 family of cytokines share the common co-receptor gp130, they have distinct biological effects in various cell types as downstream signalling pathways can involve varied signalling molecules. CNTF and IL6 signalling differ from IL11 signalling: IL11 inactivates AMPK, whereas CNTF and IL6, as mentioned by the Reviewer above, activate AMPK. Furthermore, the purported benefits of axokine/STAT3 activation were disproved in a clinical trial over two decades ago and chronic STAT3 activation is now recognised as a detrimental and pro-ageing pathway³⁰.

We beg to differ with the reviewer that “*activators of LKB1/AMPK*” is a better therapeutic strategy. While such trials with metformin are ongoing, this approach is not without its critics and side effects. More importantly, experts in the field have shown convincingly the significance of “*7 evolutionarily conserved signalling cascades, the insulin/TOR network, three MAPK (ERK, p38, JNK), JAK/STAT, TGF- β , and Nf- κ B pathways that act pleiotropically on ageing and immunity*”³⁰. Stimulation of LKB1/AMPK to extend healthspan/lifespan does not directly address the activation of ERK, JAK/STAT, NF κ B or TGF- β , all of which are modulated by anti-IL11. For the reviewer’s information, we have an ongoing study where we only activate LKB1 signalling in ageing mice. Our preliminary findings suggest that the effects of this specific intervention on ageing pathologies is limited.

References

1. López-Otín, C., Blasco, M. A., Partridge, L., Serrano, M. & Kroemer, G. The hallmarks of aging. *Cell* **153**, 1194–1217 (2013).
2. Sukoff Rizzo, S. J. *et al.* Assessing Healthspan and Lifespan Measures in Aging Mice: Optimization of

- Testing Protocols, Replicability, and Rater Reliability. *Curr. Protoc. Mouse Biol.* **8**, e45 (2018).
3. Finch, C. E. & Girgis, F. G. Enlarged seminal vesicles of senescent C57BL-6J mice. *J. Gerontol.* **29**, 134–138 (1974).
 4. Thomas, R., Wang, W. & Su, D.-M. Contributions of Age-Related Thymic Involution to Immunosenescence and Inflammaging. *Immun. Ageing* **17**, 2 (2020).
 5. Baran-Gale, J. *et al.* Ageing compromises mouse thymus function and remodels epithelial cell differentiation. *Elife* **9**, (2020).
 6. Bansal, A., Zhu, L. J., Yen, K. & Tissenbaum, H. A. Uncoupling lifespan and healthspan in *Caenorhabditis elegans* longevity mutants. *Proc. Natl. Acad. Sci. U. S. A.* **112**, E277–86 (2015).
 7. Willows, J. W. *et al.* Age-related changes to adipose tissue and peripheral neuropathy in genetically diverse HET3 mice differ by sex and are not mitigated by rapamycin longevity treatment. *Aging Cell* e13784 (2023).
 8. Garmany, A., Yamada, S. & Terzic, A. Longevity leap: mind the healthspan gap. *NPJ Regen Med* **6**, 57 (2021).
 9. Correia-Melo, C. *et al.* Rapamycin improves healthspan but not inflammaging in *nfk1b1*^{-/-} mice. *Aging Cell* **18**, e12882 (2019).
 10. Compton, S. E. *et al.* LKB1 controls inflammatory potential through CRTC2-dependent histone acetylation. *Mol. Cell* **83**, 1872–1886.e5 (2023).
 11. Puzianowska-Kuźnicka, M. *et al.* Interleukin-6 and C-reactive protein, successful aging, and mortality: the PolSenior study. *Immun. Ageing* **13**, 21 (2016).

12. Widjaja, A. A. *et al.* IL11 stimulates ERK/P90RSK to inhibit LKB1/AMPK and activate mTOR initiating a mesenchymal program in stromal, epithelial, and cancer cells. *iScience* **25**, 104806 (2022).
13. Ng, B. *et al.* Interleukin-11 causes alveolar type 2 cell dysfunction and prevents alveolar regeneration. *bioRxiv* 2022.11.11.516109 (2022) doi:10.1101/2022.11.11.516109.
14. Nishina, T. *et al.* Interleukin-11-expressing fibroblasts have a unique gene signature correlated with poor prognosis of colorectal cancer. *Nat. Commun.* **12**, 2281 (2021).
15. Ng, B. *et al.* Similarities and differences between IL11 and IL11RA1 knockout mice for lung fibro-inflammation, fertility and craniosynostosis. *Sci. Rep.* **11**, 14088 (2021).
16. Baker, D. J. *et al.* Naturally occurring p16(Ink4a)-positive cells shorten healthy lifespan. *Nature* **530**, 184–189 (2016).
17. Marthandan, S. *et al.* Similarities in Gene Expression Profiles during In Vitro Aging of Primary Human Embryonic Lung and Foreskin Fibroblasts. *Biomed Res. Int.* **2015**, 731938 (2015).
18. Martens, U. M., Chavez, E. A., Poon, S. S., Schmoor, C. & Lansdorp, P. M. Accumulation of short telomeres in human fibroblasts prior to replicative senescence. *Exp. Cell Res.* **256**, 291–299 (2000).
19. Purcell, M., Kruger, A. & Tainsky, M. A. Gene expression profiling of replicative and induced senescence. *Cell Cycle* **13**, 3927–3937 (2014).
20. Kreiling, J. A. *et al.* Age-associated increase in heterochromatic marks in murine and primate tissues. *Aging Cell* **10**, 292–304 (2011).
21. Bernadotte, A., Mikhelson, V. M. & Spivak, I. M. Markers of cellular senescence. Telomere shortening as a marker of cellular senescence. *Aging* **8**, 3–11 (2016).

22. Cook, S. A. Understanding interleukin 11 as a disease gene and therapeutic target. *Biochem. J* **480**, 1987–2008 (2023).
23. Nandurkar, H. H. *et al.* Adult mice with targeted mutation of the interleukin-11 receptor (IL11Ra) display normal hematopoiesis. *Blood* **90**, 2148–2159 (1997).
24. Polak, P. *et al.* Adipose-specific knockout of raptor results in lean mice with enhanced mitochondrial respiration. *Cell Metab.* **8**, 399–410 (2008).
25. Pinti, M. *et al.* A Comprehensive Analysis of Cytokine Network in Centenarians. *Int. J. Mol. Sci.* **24**, (2023).
26. Widjaja, A. A. *et al.* Redefining IL11 as a regeneration-limiting hepatotoxin and therapeutic target in acetaminophen-induced liver injury. *Sci. Transl. Med.* **13**, (2021).
27. Keupp, K. *et al.* Mutations in the interleukin receptor IL11RA cause autosomal recessive Crouzon-like craniosynostosis. *Mol Genet Genomic Med* **1**, 223–237 (2013).
28. Widjaja, A. A. *et al.* Inhibiting Interleukin 11 Signaling Reduces Hepatocyte Death and Liver Fibrosis, Inflammation, and Steatosis in Mouse Models of Nonalcoholic Steatohepatitis. *Gastroenterology* **157**, 777–792.e14 (2019).
29. Swaney, J. *et al.* A First-in-Class IL-11 Receptor Blocking Antibody as a Treatment for Pulmonary Fibrosis. in *D19. NEW PERSPECTIVES ON TRANSLATIONAL STUDIES FOR LUNG FIBROSIS* A6262–A6262 (American Thoracic Society, 2023).
30. Fabian, D. K., Fuentealba, M., Dönertaş, H. M., Partridge, L. & Thornton, J. M. Functional conservation in genes and pathways linking ageing and immunity. *Immun. Ageing* **18**, 23 (2021).

Reviewer Reports on the First Revision:

Referees' comments:

Referee #1 (Remarks to the Author):

Comments on Cook et al. R1

Overall evaluation:

This paper is much improved, in my view. The authors have added a good deal of important new information (including the lifespan result, but more besides), have removed much of the data that were

less than satisfactory, and have relegated much of the less pertinent data to the “Extended” section. They now make quite a strong case that many aspects of aging in mice involve harm done by IL11, and show three different lines of evidence that blocking IL11 may be beneficial. Over-statements and interpretive over-reach have been largely muted. The amount of work shown is impressive, and the consistency among their findings builds confidence in their main conclusions. The discussion is on point, concise, insightful, and balanced. I think the paper is likely to prove highly influential and to be seen as a significant step forward in the biology of aging and age-related pathology. I hope it will be published soon, and am confident it will have an impact.

That said, there are still several matters that may require attention and some re-writing or re-interpretation. I will list these below, in the hopes that the authors will consider these suggestions as ways to strengthen further their already strong paper.

Major importance:

I would urge the authors to be much more careful to distinguish effects of IL11 (or IL11ra) seen in young mice, from those that appear only in old mice. Fig 2i is a good example (there are many others): muscle weight is higher in the IL11-deficient mice even when young, and this genetic effect is also seen in the old mice. There are two points, both missed by the authors. First, the effects of IL11 on this phenotype (and many others which have a similar age pattern, i.e. seen in young mice) are established prior to aging, giving clues to possible mechanisms by which IL11 tone may modulate aging. Second, data that take this form cannot be interpreted as evidence for slower aging; in this case the authors cannot justifiably conclude that sarcopenia is decelerated in the IL11-deficient mice, since the effect is clear even in young mice. The authors have plenty of evidence to support their most important conclusion, i.e. that many aspects of aging are delayed in the low-IL11 models – but data on variables (here, muscle strength) that are altered in young mice cannot be taken as evidence for this conclusion. This is not merely a quibble: things that are changed early in life (by IL11 modulation) are plausibly causal to those that occur at older ages.

Considerable importance:

The authors place more reliance than I would on the “liver DNA methylation age” shown in Fig 1m. The control mice were 110 weeks old, but the assay suggested that they were about 78 weeks old – not a confidence-builder. The IL11-ra mutants were estimated to be 20 weeks old, i.e. about a quarter of the age of the controls, which is out of line with the lifespan result and most of the other health indices in this paper. The design did not include estimation of the methylation age of young mice, and the data are (according to the helpful rebuttal letter) proprietary and thus available for inspection of confirmation. I would take out this panel, since the other data in the figure make a strong case without raising similar technical reservations.

I remain unconvinced that the data on senescence in human fibroblast cell lines adds anything of value to this paper, and would recommend that this material be removed, but the authors have now moved this to the Extended Figure section (EF 4), which is going in the right direction.

I remain disappointed that the authors continue to make inferences that can only be tested by interaction terms in two-way ANOVA, without showing (or mentioning) the interaction terms. Statements like “IL11 affects X in old but not in young mice” cannot legitimately be supported by showing significance in old mice but not in young mice. The authors understand the idea full well, and have now included a new table (Extended Data Table 9) that includes these results, but very few readers will dig this far into the extended materials, and the approach used in the main text and figures is not valid on its own. This is something I’d urge the authors to correct, and urge the editors to require.

Minor:

The authors have a lot more faith than I do in the mouse “frailty index,” but most of their conclusions based on this questionable index are also supported by other lines of evidence, so little harm is done.

The abbreviation “w/o” for “weeks old” is not a good idea, since many people use this to mean “without.”

I would not agree that Fig 3f shows “reversal” of muscle dysfunction. I think the data show prevention of muscle dysfunction, a different thing.

When the authors write “lesser tumours” they mean “fewer tumours.”

Referee #2 (Remarks to the Author):

The authors added a significant amount of new data and addressed all major concerns appropriately. One remaining concern is the lack of actual energy expenditure measurements, which would have further strengthened the study.

Referee #3 (Remarks to the Author):

The authors have significantly improved the MS. Notwithstanding, they have largely dismissed my concerns.

Author Rebuttals to First Revision:

Second point-by-point responses to to the Reviewers' comments at Nature

Referee #1 (Remarks to the Author):

Comments on Cook et al. R1

Overall evaluation:

This paper is much improved, in my view. The authors have added a good deal of important new

information (including the lifespan result, but more besides), have removed much of the data that were less than satisfactory, and have relegated much of the less pertinent data to the “Extended” section. They now make quite a strong case that many aspects of aging in mice involve harm done by IL11, and show three different lines of evidence that blocking IL11 may be beneficial. Over-statements and interpretive over-reach have been largely muted. The amount of work shown is impressive, and the consistency among their findings builds confidence in their main conclusions. The discussion is on point, concise, insightful, and balanced. I think the paper is likely to prove highly influential and to be seen as a significant step forward in the biology of aging and age-related pathology. I hope it will be published soon, and am confident it will have an impact.

Response: We are grateful to the reviewer for these comments and we aim to improve the manuscript further in this revised version. In particular, we have more fully addressed potential over-statements from data derived from studies of mice with germline *Il11* or *Il11ra1* deletion. We have revised the main figures to show significance terms associated with two-way ANOVA analyses, where relevant. The text has been further revised to improve clarity, including a slight expansion of the discussion that we feel is now more balanced and better aligned with the experimental findings and published literature.

That said, there are still several matters that may require attention and some re-writing or re-interpretation. I will list these below, in the hopes that the authors will consider these suggestions as ways to strengthen further their already strong paper.

Major importance:

I would urge the authors to be much more careful to distinguish effects of IL11 (or IL11ra) seen in young mice, from those that appear only in old mice. Fig 2i is a good example (there are many others): muscle weight is higher in the IL11-deficient mice even when young, and this genetic effect is also seen in the old mice. There are two points, both missed by the authors. First, the effects of IL11 on this phenotype (and many others which have a similar age pattern, i.e. seen in young mice) are established prior to aging, giving clues to possible mechanisms by which IL11 tone may modulate aging. Second, data that take this form cannot be interpreted as evidence for slower aging; in this case the authors cannot justifiably conclude that sarcopenia is decelerated in the IL11-deficient mice, since the effect is clear even in young mice. The authors have plenty of evidence to support their most important conclusion, i.e. that many aspects of aging and delayed in the low-IL11 models – but data on variables (here, muscle strength) that are altered in young mice cannot be taken as evidence for this conclusion. This is not merely a quibble: things that are changed early in life (by IL11 modulation) are plausibly causal to those that occur at older ages.

Response: We are strongly aligned with the Reviewer’s point of view that phenotypes that are different between WT and *Il11* or *Il11ra1* KO mice in early life and are also evident in aged animals cannot be taken as evidence of slowed or delayed ageing of said phenotypes. We agree we a improve further on

our representations of the data from our studies of *Il11* and *Il11ra1* KO mice (both graphical and in the text).

In the revised manuscript, we have improved the clarity of the main text and expanded on the concept that beneficial phenotypes evident in mice with germline gene deletion in early life can carry through to later life and this is something that must be taken into account and addressed using late life experiments. This aspect is now also specifically mentioned. In addition we have removed phraseology such as "age-dependent reductions" when referring to old KO mice. We discuss this more below.

Considerable importance:

*The authors place more reliance than I would on the "liver DNA methylation age" shown in Fig 1m. The control mice were 110 weeks old, but the assay suggested that they were about 78 weeks old – not a confidence-builder. The *IL11-ra* mutants were estimated to be 20 weeks old, i.e. about a quarter of the age of the controls, which is out of line with the lifespan result and most of the other health indices in this paper. The design did not include estimation of the methylation age of young mice, and the data are (according to the helpful rebuttal letter) proprietary and thus available for inspection of confirmation. I would take out this panel, since the other data in the figure make a strong case without raising similar technical reservations.*

Response: The company that ran these experiments (<https://zymoresearch.eu/pages/mouse-dnage>) used mice from 20-70 weeks of age to generate the standard curves that they use. Thus, the oldest age that can be ascribed accurately to a mouse using this assay is between 70 to 80 weeks, as the standard curve does not go higher. The relationship between epigenetic signature and age up to 70 weeks is linear and robust ($R^2=0.97$) and ages derived from the standard curve up to 70-week-old should be accurate.

There is great interest in biomarkers of ageing, which includes epigenetic read outs, and prefer to retain these data in the manuscript as it is a question that very many readers of the manuscript will want to know. To mitigate for the range limitations of the assay we have (1) added details on the age range to the methods, (2) showed only relative differences in ages (as opposed to absolute age, which is inaccurate at the upper end), and also (3) moved the data on epigenetic age to the supplementary: old Fig. 1m is now **revised Extended Data Fig. 2o** and old Fig. 3n is now **revised Extended Data Fig. 7j**). The text has also been revised accordingly.

I remain unconvinced that the data on senescence in human fibroblast cell lines adds anything of value to this paper, and would recommend that this material be removed, but the authors have now moved this

to the Extended Figure section (EF 4), which is going in the right direction.

Response: We believe that studies of human cells are important for establishing the translational relevance of murine experiments. Indeed in previous studies, when we only presented mouse data, we have been specifically asked to show human cell data by reviewers, for this very reason: to show the translational significance of the said study. We now recognize the importance of showing that the same pathway is regulated by the same stimulus in human cells to drive a relevant pathological phenotype (senescence in this case) as a prerequisite for advancing a new therapeutic concept towards clinical application. Furthermore, these studies show the cell-autonomous effects of IL11 independently/in the absence of nervous, immune, inter-organ, or endocrine effects, while also confirming the dependence of IL11 effects on ERK and mTOR in human cells. For these various reasons, we ask the reviewer to consider the added value of these data, which would otherwise remain unpublished. We strongly prefer to retain these data in the current reduced form and only presented in the Extended Data Figure section.

I remain disappointed that the authors continue to make inferences that can only be tested by interaction terms in two-way ANOVA, without showing (or mentioning) the interaction terms. Statements like “IL11 affects X in old but not in young mice” cannot legitimately be supported by showing significance in old mice but not in young mice. The authors understand the idea full well, and have now included a new table (Extended Data Table 9) that includes these results, but very few readers will dig this far into the extended materials, and the approach used in the main text and figures is not valid on its own. This is something I’d urge the authors to correct, and urge the editors to require.

Response: We refer the reviewer to our initial comments above where this issue was discussed. As highlighted by the reviewer, we show all interaction terms in a large **Extended Data table 9**. We propose to further improve representations by adding two-way ANOVA significance terms below the relevant figure panels of the main figures and by toning down inferences of ageing-specific effects in studies of mice with germline gene deletion. We now highlight more clearly beneficial phenotypes that are seen in both young and old mice with germline IL11 loss-of-function.

Minor:

The authors have a lot more faith than I do in the mouse “frailty index,” but most of their conclusions based on this questionable index are also supported by other lines of evidence, so little harm is done.

Response: Frailty was assessed with observers blinded to treatment, using a 27-point frailty scoring system¹. While there is no perfect way to perform semi-quantitative phenotyping, such as frailty scoring,

the test we used is widely accepted as one of the most robust there is in mice and the data were acquired using rigorous conditions. Frailty is an important clinical phenotype (again, semi-quantitative in humans) and predicts clinical outcomes and we believe that it is important to represent this phenotype in studies of old age in mice. A nice review of the interaction of frailty and metabolism was recently published in Cell Metabolism, which highlights the importance of measuring frailty, and how to do so in mice and humans, and we refer the reviewer to this paper for further insights (PMID: 38614092).

The abbreviation “w/o” for “weeks old” is not a good idea, since many people use this to mean “without.”

Response: We thank the reviewer for this observation, we have left the word ‘week-old’ as it is (unabbreviated) in the revised main manuscript.

I would not agree that Fig 3f shows “reversal” of muscle dysfunction. I think the data show prevention of muscle dysfunction, a different thing.

Response: The data the reviewer refers to shows that muscle function in 110-week-old mice on anti-IL11 are improved as compared to that of the same mice when at 75 weeks old. Given that muscle function in mice declines from middle adulthood, it is difficult to imagine how the observed data does not reflect a reversal of muscle dysfunction. Nonetheless, we have modified the wording to the following: *“Muscle strengths of 100-week-old mice receiving anti-IL11 were higher than aged-matched controls on IgG or untreated, and also those of 75-week-old mice (Fig. 3f; Extended Data Fig. 7a).”* We have also toned down any reference to reversal throughout the manuscript.

When the authors write “lesser tumours” they mean “fewer tumours.”

Response: This has been amended as suggested.

Referee #2 (Remarks to the Author):

The authors added a significant amount of new data and addressed all major concerns appropriately. One remaining concern is the lack of actual energy expenditure measurements, which would have

further strengthened the study.

Response: The limitations of the energy expenditure studies are now mentioned in the revised discussion.

Referee #3 (Remarks to the Author):

The authors have significantly improved the MS. Notwithstanding, they have largely dismissed my concerns.

Response: We thank the Reviewer for recognizing our efforts.

References:

1. Sukoff Rizzo, S. J. *et al.* Assessing Healthspan and Lifespan Measures in Aging Mice: Optimization of Testing Protocols, Replicability, and Rater Reliability. *Curr. Protoc. Mouse Biol.* **8**, e45 (2018).

Reviewer Reports on the Second Revision:

Referees' comments:

Referee #1 (Remarks to the Author):

Widjaja, Cook, et al., "Inhibition of IL11 signalling extends mammalian healthspan and lifespan."

The modifications made in this already strong paper have (with one exception, noted below, related to assertions about "methylation age") improved it still further. I am glad that the authors now provide ANOVA statistics in key panels to support inferences about sex-specific effects of blocking IL11 actions.

Senescence: I take their point that many readers will be interested in the idea that inhibition of IL11 signals might somehow alter phenotypes these readers view as characteristic of “senescence.” These readers are, however, confused by the admittedly prevalent view that there is such a thing as a “senescent” cell, rather than an large group of disparate age-dependent cell types that, in many cases, contribute to illness by secretion of varying sets of cytokines, or by not dividing when they ought. The idea that aging is do to accumulation of a specific kind of cell, the “senescent” cell, is akin to the alarmingly under-specified notion that infections are caused by “microbes.” I am sorry that the authors decided to kowtow to this misunderstanding, but grateful that they have relegated the senescent bits to extended data.

Frailty: I agree with the authors that they have used an approach to estimation of “frailty” that is now considered optimal by many laboratories, but the approach is still awful. The basic idea is to measure a whole lot of things that change with age, assign each equal weight, and add them up. A valid approach, in contrast, would consider multi-collinearity among related measures, would assign variable weights that optimize ability to predict something of interest (like age, or remaining life, or composite disease risk), and would then validate this in multiple settings, including a variety of slow-aging mouse models (calorie restriction, Ames dwarf, rapamycin, etc.) varying in sex, strain, diet, etc. Assigning each element in the test battery a weight of 1 is indefensible, and the authors’ assessment that doing this is currently the practice in many laboratories does not make it a good idea. It’s also worth noting that the “frailty index” used in most mouse work is quite different from the use of the term “frailty” in human clinical practice or in colloquial usage. The authors’ reply memo does not address any of these problem areas. I stand by my comments about the R1 version: “most of their conclusions based on this questionable index are also supported by other lines of evidence, so little harm is done.”

Methylation age: this is the only element of R2 that I would ask the authors to reconsider. I had pointed out that the method they used for calculation of methylation age was producing results that called the value and accuracy of the method into question. I wrote (in connection with R1):

“The control mice were 110 weeks old, but the assay suggested that they were about 78 weeks old – not a confidence-builder. The IL11-ra mutants were estimated to be 20 weeks old, i.e. about a quarter of the age of the controls, which is out of line with the lifespan result and most of the other health indices in this paper. The design did not include estimation of the methylation age of young mice, and the data are (according to the helpful rebuttal letter) proprietary and thus available for inspection of confirmation. I would take out this panel, since the other data in the figure make a strong case without raising similar technical reservations.”

In their reply memo, the authors helpfully offer plausible explanations for these problems: the secret proprietary method used by their contract lab was standardized on mice 20 – 70 weeks old, and is considered to be unreliable for mice outside this range. In response to my concerns, they converted the data in the relevant figure panels from “absolute” age in weeks to “relative age,” and delegated the information to extended data figures.

My feeling is that if you know a method is inappropriate – in this case because the test animals lie outside the standard curve – you should not use it. The use of proprietary methods is also a problem, because replication of these results will not be possible except by using the same, secret, recipe. And, importantly, I do not think the authors' solution of showing only relative ages is an improvement – it does not mitigate the inaccuracy of the method, it just conceals from the reader the information he or she needs to notice the technical problem.

My recommendation would be to remove these figure panels (now Extended Fig 2o and 7j). But if the authors decide to keep them, I would urge them to use the earlier versions, with estimated absolute ages, rather than conceal that information from the reader.

=====

On the whole, I think the research group should be complimented on an important set of discoveries, which I think are likely to have a strong impact on our concepts of how aging and inflammation are related.

Author Rebuttals to Second Revision:

Third point-by-point responses to the Reviewers' comments at Nature

Referees' comments:

Referee #1 (Remarks to the Author):

Widjaja, Cook, et al., "Inhibition of IL11 signalling extends mammalian healthspan and lifespan."

The modifications made in this already strong paper have (with one exception, noted below, related to assertions about "methylation age") improved it still further. I am glad that the authors now provide ANOVA statistics in key panels to support inferences about sex-specific effects of blocking IL11 actions.

Senescence: I take their point that many readers will be interested in the idea that inhibition of IL11 signals might somehow alter phenotypes these readers view as characteristic of "senescence." These readers are, however, confused by the admittedly prevalent view that there is such a thing as a "senescent" cell, rather than an large group of disparate age-dependent cell types that, in many cases, contribute to illness by secretion of varying sets of cytokines, or by not dividing when they ought. The idea that aging is do to accumulation of a specific kind of cell, the "senescent" cell, is akin to the alarmingly under-specified notion that infections are caused by "microbes." I am sorry that the authors decided to kowtow to this misunderstanding, but grateful that they have relegated the senescent bits to extended data.

Response: We thank the reviewer for this comment and understand his/her position on this.

Frailty: I agree with the authors that they have used an approach to estimation of "frailty" that is now considered optimal by many laboratories, but the approach is still awful. The basic idea is to measure a whole lot of things that change with age, assign each equal weight, and add them up. A valid approach, in contrast, would consider multi-collinearity among related measures, would assign variable weights that optimize ability to predict something of interest (like age, or remaining life, or composite disease risk), and would then validate this in multiple settings, including a variety of slow-aging mouse models (calorie restriction, Ames dwarf, rapamycin, etc.) varying in sex, strain, diet, etc. Assigning each element in the test battery a weight of 1 is indefensible, and the authors' assessment that doing this is currently the practice in many laboratories does not make it a good idea. It's also worth noting that the "frailty index" used in most mouse work is quite different from the use of the term "frailty" in human clinical practice or in colloquial usage. The authors' reply memo does not address any of these problem areas. I stand by my comments about the R1 version: "most of their conclusions based on this questionable index are also supported by other lines of evidence, so little harm is done."

Response: We thank the reviewer for this comment and understand his/her position on this.

Methylation age: this is the only element of R2 that I would ask the authors to reconsider. I had pointed out that the method they used for calculation of methylation age was producing results that called the value and accuracy of the method into question. I wrote (in connection with R1):

“The control mice were 110 weeks old, but the assay suggested that they were about 78 weeks old – not a confidence-builder. The IL11-ra mutants were estimated to be 20 weeks old, i.e. about a quarter of the age of the controls, which is out of line with the lifespan result and most of the other health indices in this paper. The design did not include estimation of the methylation age of young mice, and the data are (according to the helpful rebuttal letter) proprietary and thus available for inspection of confirmation. I would take out this panel, since the other data in the figure make a strong case without raising similar technical reservations.”

In their reply memo, the authors helpfully offer plausible explanations for these problems: the secret proprietary method used by their contract lab was standardized on mice 20 – 70 weeks old, and is considered to be unreliable for mice outside this range. In response to my concerns, they converted the data in the relevant figure panels from “absolute” age in weeks to “relative age,” and delegated the information to extended data figures.

My feeling is that if you know a method is inappropriate – in this case because the test animals lie outside the standard curve – you should not use it. The use of proprietary methods is also a problem, because replication of these results will not be possible except by using the same, secret, recipe. And, importantly, I do not think the authors’ solution of showing only relative ages is an improvement – it does not mitigate the inaccuracy of the method, it just conceals from the reader the information he or she needs to notice the technical problem.

My recommendation would be to remove these figure panels (now Extended Fig 2o and 7j). But if the authors decide to keep them, I would urge them to use the earlier versions, with estimated absolute ages, rather than conceal that information from the reader.

Response: All methylation clock data and reference to these data have been removed.

=====

On the whole, I think the research group should be complimented on an important set of discoveries, which I think are likely to have a strong impact on our concepts of how aging and inflammation are related.

Response: We thank the Reviewer for his/her kind words and for recognizing the impact and importance of our study.